# EASYTUNE: EFFICIENT STEP-AWARE FINE-TUNING FOR DIFFUSION-BASED MOTION GENERATION

**Xiaofeng Tan**[*]  **Wanjiang Weng**[*]  **Haodong Lei**  **Hongsong Wang**[†]
Department of Computer Science and Engineering, Southeast University, Nanjing, China
Key Laboratory of New Generation Artificial Intelligence Technology and Its Interdisciplinary
Applications (Southeast University), Ministry of Education, Nanjing, China
`{xiaofengtan, wjweng, hongsong.wang}@seu.edu.cn`

## ABSTRACT

In recent years, motion generative models have undergone significant advancement, yet pose challenges in aligning with downstream objectives. Recent studies have shown that using differentiable rewards to directly align the preference of diffusion models yields promising results. However, these methods suffer from (1) inefficient and coarse-grained optimization with (2) high memory consumption. In this work, we first theoretically and empirically identify the *key reason* of these limitations: the recursive dependence between different steps in the denoising trajectory. Inspired by this insight, we propose **EasyTune**, which fine-tunes diffusion at each denoising step rather than over the entire trajectory. This decouples the recursive dependence, allowing us to perform (1) a dense and fine-grained, and (2) memory-efficient optimization. Furthermore, the scarcity of preference motion pairs restricts the availability of motion reward model training. To this end, we further introduce a **S**elf-refinement **P**reference **L**earning (**SPL**) mechanism that dynamically identifies preference pairs and conducts preference learning. Extensive experiments demonstrate that EasyTune outperforms DRaFT-50 by 7.7% in alignment (MM-Dist) improvement while requiring only 31.16% of its additional memory overhead and achieving a **7.3×** training speedup. The project page is available at this link.

## 1 INTRODUCTION

Text-to-motion generation aims to synthesize realistic and coherent human motions from natural language (Shen et al., 2025), enabling applications in animation (Azadi et al., 2023), human-computer interaction (Peng et al., 2024), and virtual reality (Tashakori et al., 2025). Recent advances are driven by diffusion models, which capture complex distributions and synthesize high-quality motions from text (Chen et al., 2023). However, their likelihood-based training (Guo et al., 2022a) often misaligns with downstream goals such as semantic understanding (Tan et al., 2025; Su et al., 2026), motion plausibility (Wang et al., 2024), and user preference (Tan et al., 2026).

To bridge this gap, reinforcement learning from human feedback (RLHF) (Kirstain et al., 2023) has been explored to fine-tune diffusion models toward human preferences and task-specific goals. Existing approaches include differentiable reward methods (Clark et al., 2024), reinforcement learning (Black et al., 2023), and direct preference optimization (DPO) (Wallace et al., 2024). Among these, DPO provides a effective way to align models using preference pairs. However, acquiring large-scale, high-quality preference pairs remains challenging due to the cost and difficulty of capturing nuanced semantic and preference signals. A more efficient alternative is to fine-tune models using a reward model that captures semantic alignment and task preference. Reinforcement learning methods, such as DDPO (Black et al., 2023) and DPOK (Fan et al., 2023a), treat the denoising trajectory as a Markov Decision Process, where intermediate motions are states and final motions are evaluated by a reward model. Differentiable reward methods, including DRaFT (Clark et al., 2024) and DRTune (Wu et al., 2025a), directly backpropagate gradients from a differentiable reward $\mathcal{R}(\mathbf{x}^{\theta})$ to optimize the model $\theta$.

---

[*]Equal contribution. [†]Corresponding author.

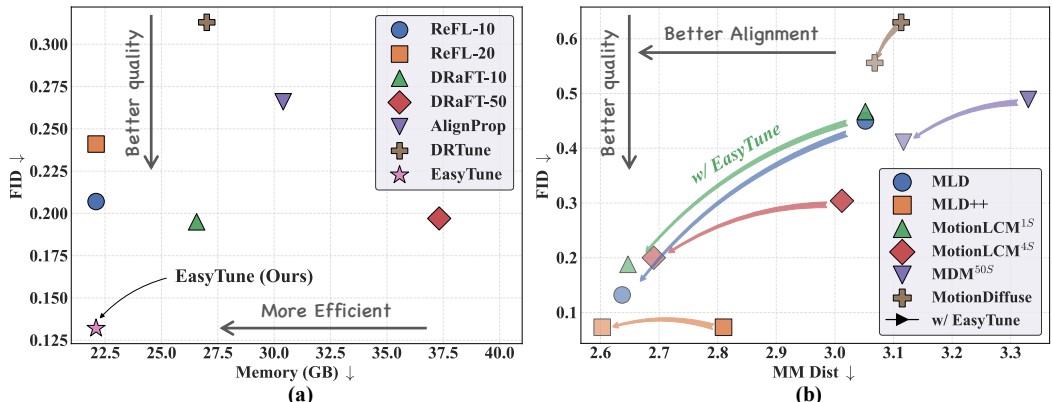

Figure 1: **Comparison of the training costs and generation performance on HumanML3D** (Guo et al., 2022a). (a) Performance comparison of different fine-tuning methods (Clark et al., 2024; Prabhudesai et al., 2023; Wu et al., 2025a). (b) Generalization performance across six pre-trained diffusion-based models (Chen et al., 2023; Dai et al., 2025; 2024; Tevet et al., 2023).

However, these methods still face several primary limitations that hinder their application to diffusion-based motion generation: (1) **Sparse and coarse-grained optimization:** Most approaches only optimize model parameters $\theta$ once after completing a multi-step denoising trajectory, resulting in sparse optimization signals and slowing down convergence. (2) **Excessive memory consumption:** These methods optimize the model $\theta$ by backpropagating the gradients of the reward value $\nabla_\theta \mathcal{R}(\mathbf{x}_\theta)$, which is related to the overall denoising trajectory. Notably, this requires storing a large computation graph of the entire trajectory, leading to excessive memory consumption. Beyond these computational challenges, existing methods rely on intricate designs such as early stopping or partial gradient blocking, increasing implementation complexity and limiting applicability. Moreover, research on motion-specific reward models is limited, so current approaches (Tan et al., 2025) typically use general-purpose retrieval models, which may inadequately capture motion preferences.

**Contributions.** In this work, we first theoretically (Corollary 1) and empirically (Fig. 6) identify a key factor of the significant computational and memory overhead: *the optimization is recursively related to the multi-step denoising trajectory,* causing the reward value of generated motions, $\mathcal{R}(\mathbf{x}^\theta)$, to be recursively depended on each denoised step throughout the overall trajectory. Specifically, each denoised motion $\mathbf{x}_t^\theta$ is generated from the diffusion model, $\mathbf{x}_t^\theta \sim \epsilon_\theta$, and recursively dependent on previous steps $\mathbf{x}_t^\theta \sim \mathbf{x}_{t+1}^\theta$. Thus, computing the gradient $\nabla_\theta \mathbf{x}_t^\theta$ requires solving for that of the prior step, $\nabla_\theta \mathbf{x}_1^\theta$, which in turn depends on those of subsequent steps, $\nabla_\theta \mathbf{x}_2^\theta, \nabla_\theta \mathbf{x}_3^\theta, ..., \nabla_\theta \mathbf{x}_T^\theta$, leading to the large significant computational and memory overhead. Building on this, we then theoretically analyze and empirically validate (Fig.3) the primary limitation of existing methods: *coarse-grained chain optimization leads to vanishing gradients, hindering optimization of early denoising steps.* To address this, we introduce a simple and effective insight: ***perform optimization at each denoising step, thereby decoupling gradients from the full reverse trajectory.*** By decoupling gradients from the denoising trajectory, our *EasyTune* framework facilitates: (1) dense and fine-grained optimization through clearing the computational graph after each denoising step, (2) avoiding storing them until denoising completes, and thus (3) obviation of the need for complex memory-saving techniques.

Nevertheless, two critical challenges remain: the lack of a reliable motion reward model and reward perception for intermediate denoising steps. The first issue stems from limited large-scale, high-quality preference data, making it difficult to train a motion-specific reward model directly. To overcome this, we propose a **S**elf-refinement **P**reference **L**earning (**SPL**), which adapts a pre-trained text-to-motion retrieval model for preference evaluation without human annotations. We dynamically construct preference pairs from the retrieval datasets and fail-retrieved results, and fine-tune this retrieval model to capture implicit preferences. For noisy intermediate steps, we employ single-step prediction rewards for ODE-based models and noise-aware rewards for SDE-based models.

Finally, we evaluate EasyTune on HumanML3D (Guo et al., 2022a) and KIT-ML (Plappert et al., 2016) with six pre-trained diffusion models. As shown in Fig.1, EasyTune achieves SoTA performance (FID = **0.132**, **70.7%** better than the MLD base model), while cutting memory usage to **22.10** GB and achieving a **7.3×** training speedup over DRaFT. In summary, our contributions are as follows:

1. We theoretically and empirically identify the cost and performance limitations of existing differentiable-reward methods, and propose EasyTune, an effective step-aware fine-tuning method.
2. To the best of our knowledge, this work is the first to fine-tune diffusion-based text-to-motion models with a differentiable reward. To achieve this, we introduce SPL to fine-tune a pre-trained retrieval model for preference evaluation without human-annotated preference pairs.
3. Extensive experiments demonstrate that EasyTune significantly outperforms existing methods in terms of performance, optimization efficiency, and storage requirements.

## 2 RELATED WORKS

**Text-to-Motion Generation.** Text-to-motion generation (Chen et al., 2023; Guo et al., 2023) produces human motion sequences from textual descriptions. Among these works, as a powerful generative model, diffusion models (Chen et al., 2023; Tevet et al., 2023) iteratively denoise latent motions under text guidance, offering higher quality and stability. Recent advances include transformer-based diffusion with geometric losses (Tevet et al., 2023), few-step controllable inference (Dai et al., 2024), and hybrid discrete-continuous modeling (Meng et al., 2025). However, these methods primarily target the pretraining stage by aligning to fixed dataset distributions (Guo et al., 2022a), yet they remain misaligned with semantic coherence (Tan et al., 2025) and physical reality.

**Post-training in Motion Generation.** Post-training for motion generation improves semantic coherence (Tan et al., 2025; Pappa et al., 2024) and physical realism (Yuan et al., 2023; Han et al., 2025; Wang et al., 2025a). It commonly uses motion discriminative models as reward signals (Petrovich et al., 2023; Weng et al., 2025a; Wang et al., 2025c;b; Weng et al., 2025b) and applies reinforcement learning to refine behaviors beyond supervised training. Representative methods include SoPo, which optimizes a DPO objective from preference pairs (Tan et al., 2025), MotionCritic, which learns from human preferences with PPO (Wang et al., 2025a), ReinDiffuse with rule-based rewards (Han et al., 2025), and fake motion synthesis for improving motion detection (Tan et al., 2024). However, these methods often optimize within the pretraining reward domain, while the reward model is defined in a separate and opaque space (Janner et al., 2019; Yao et al., 2022; 2024), which can limit feedback quality (Wang et al., 2025a) and increase dependence on preference data (Tan et al., 2025; Pappa et al., 2024). In contrast, we target semantic alignment in text-to-motion generation with a differentiable reward model and a simple reinforcement learning algorithm, treating semantics as the primary objective.

**Differentiable Reward Fine-Tuning for Diffusion Models.** Fine-tuning pre-trained diffusion models with reward signals can be achieved via policy-gradient methods, DPO-based methods, or differentiable reward maximization. Policy-gradient methods (Wu et al., 2025b) such as DDPO and DPOK (Black et al., 2023; Fan et al., 2023a) formulate denoising as an MDP and optimize expected rewards, while DPO-based methods such as Diffusion-DPO (Wallace et al., 2024) and SoPo (Tan et al., 2025) optimize a preference objective from preference pairs. However, policy-gradient updates often rely on exactly computable likelihoods, which can be problematic for diffusion models (Zheng et al., 2025), and DPO introduces explicit dependence on preference data. We therefore adopt differentiable reward-based fine-tuning by directly maximizing reward values (Clark et al., 2024; Prabhudesai et al., 2023; Wu et al., 2025a), which is effective for aligning diffusion models to downstream objectives including semantics (Tan et al., 2025) and safety (Su et al., 2025; Wang et al., 2026). However, as discussed in Sec.3, these approaches can require backpropagation through denoising, leading to sparse gradients, slow convergence, and high memory costs.

## 3 MOTIVATION: RETHINKING DIFFERENTIABLE REWARD-BASED METHODS

**Preliminaries.** As illustrated in Fig. 2, existing methods fine-tune a pre-trained motion diffusion model by maximizing the reward value $\mathcal{R}_\phi(\mathbf{x}_0^\theta, c)$ of the motion $\mathbf{x}_0^\theta$ generated via a $T$-step reverse process. Notably, this generated motion $\mathbf{x}_t^\theta$ requires retaining gradients throughout the entire denoising trajectory $\mathbf{x}_t^\theta$, and thus the model can be optimized via maximizing its reward value $\nabla_\theta \mathcal{R}_\phi(\mathbf{x}_0^\theta, c)$.

Given a pre-trained motion diffusion model parameterized by $\epsilon_\theta$, the optimization objective is to fine-tune $\theta$ to maximize the reward value $\mathcal{R}_\phi(\mathbf{x}_0^\theta, c)$, with the loss defined as:

$$\mathcal{L}(\theta) = -\mathbb{E}_{c \sim \mathcal{D}_\mathrm{T}, \mathbf{x}_0^\theta \sim \pi_\theta(\cdot|c)} \left[ \mathcal{R}_\phi(\mathbf{x}_0^\theta, c) \right], \tag{1}$$

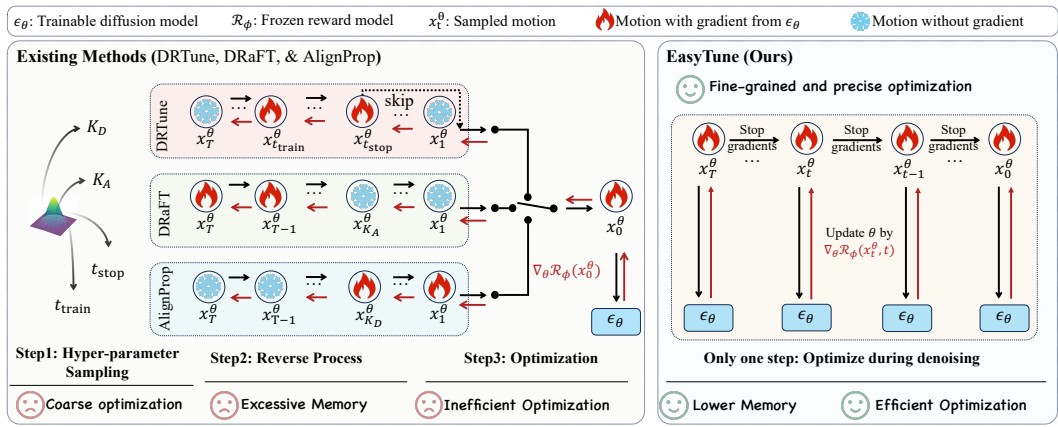

Figure 2: **The framework of existing differentiable reward-based methods (left) and our proposed EasyTune (right).** Existing methods backpropagate the gradients of the reward model through the overall denoising process, resulting in (1) excessive memory, (2) inefficient, and (3) coarse-grained optimization. In contrast, EasyTune optimizes the diffusion model by directly backpropagating the gradients at each denoising step, overcoming these issues.

where $c$ is a text condition from the training set $\mathcal{D}_\mathrm{T}$, and $\mathbf{x}_0^\theta$ is the motion generated from noise $\mathbf{x}_T \sim \mathcal{N}(\mathbf{0}, \mathbf{I})$ via a $T$-step reverse process $\pi_\theta$. The $t$-th step of the reverse process is denoted as:

$$\mathbf{x}_{t-1}^\theta = \pi_\theta(\mathbf{x}_t^\theta, t, c) := \frac{1}{\sqrt{\alpha_t}}\left(\mathbf{x}_t^\theta - \frac{\beta_t}{\sqrt{1-\bar{\alpha}_t}}\epsilon_\theta(\mathbf{x}_t^\theta, t, c)\right), \tag{2}$$

where $\mathbf{x}_{t-1}^\theta$ is the denoised motion at step $t-1$, and $\alpha_t$, $\beta_t$ are noise schedule parameters.

**Gradient Analysis.** To optimize the loss in Eq. (1), we further analyze the gradient computation, where the gradient of $\mathcal{L}(\theta)$ w.r.t. the model parameters $\theta$ is computed via the chain rule:

$$\frac{\partial\mathcal{L}(\theta)}{\partial\theta} = -\mathbb{E}_{c\sim\mathcal{D}_\mathrm{T},\mathbf{x}_0^\theta\sim\pi_\theta(\cdot|c)}\left[\frac{\partial\mathcal{R}_\phi(\mathbf{x}_0^\theta, c)}{\partial\mathbf{x}_0^\theta}\cdot\frac{\partial\mathbf{x}_0^\theta}{\partial\theta}\right]. \tag{3}$$

Here, $\frac{\partial\mathcal{R}_\phi(\mathbf{x}_0^\theta, c)}{\partial\mathbf{x}_0^\theta}$ represents the gradient of the reward model w.r.t. the generated motion, and $\frac{\partial\mathbf{x}_0^\theta}{\partial\theta}$ captures the dependence of the generated motion $\mathbf{x}_t^\theta$ on the model $\theta$ through the reverse trajectory.

Eq. (3) indicates that the gradient of loss function can be divided into two terms: $\partial\mathcal{R}_\phi(\mathbf{x}_0^\theta, c)/\partial\mathbf{x}_0^\theta$, which can be directly computed from the reward model, and $\partial\mathbf{x}_0^\theta/\partial\theta$, which depends on the denoising trajectory $\pi_\theta$. Here, we introduce Corollary 1 to analyze this gradient (See the proof in App. C.1).

**Corollary 1.** *Given the reverse process in Eq. (2), $\mathbf{x}_{t-1}^\theta = \pi_\theta(\mathbf{x}_t^\theta, t, c)$, the gradient w.r.t diffusion model $\theta$, denoted as $\frac{\partial\mathbf{x}_{t-1}^\theta}{\partial\theta}$, can be expressed as:*

$$\frac{\partial\mathbf{x}_{t-1}^\theta}{\partial\theta} = \frac{\partial\pi_\theta(\mathbf{x}_t^\theta, t, c)}{\partial\theta} + \frac{\partial\pi_\theta(\mathbf{x}_t^\theta, t, c)}{\partial\mathbf{x}_t^\theta}\cdot\frac{\partial\mathbf{x}_t^\theta}{\partial\theta}. \tag{4}$$

Corollary 1 shows that the computation involves two parts: (1) a direct term (in blue) from the dependence of the diffusion model $\pi_\theta$ on $\theta$, and (2) an indirect term (in red) that depends on the $t$-th step generated motion $\mathbf{x}_t^\theta$. However, the reverse process in diffusion models is inherently recursive, where the denoised motion $\mathbf{x}_{t-1}$ is relied on $\mathbf{x}_t$, which in turn depends on $\mathbf{x}_{t+1}$, resulting in substantial computational complexity for $T$ time steps intermediate variables.

To compute the full gradient $\partial\mathcal{L}(\theta)/\partial\theta$, we unroll the $\partial\mathbf{x}_0^\theta/\partial\theta$ using Corollary 1 and substitute it into Eq. (3) resulting in (see proof in App. C.3):

$$\frac{\partial\mathcal{L}(\theta)}{\partial\theta} = -\mathbb{E}_{c\sim\mathcal{D}_\mathrm{T},\mathbf{x}_0^\theta\sim\pi_\theta(\cdot|c)}\left[\frac{\partial\mathcal{R}_\phi(\mathbf{x}_0^\theta)}{\partial\mathbf{x}_0^\theta}\cdot\sum_{t=1}^{T}\underbrace{\left(\prod_{s=1}^{t-1}\frac{\partial\pi_\theta(\mathbf{x}_s^\theta, s, c)}{\partial\mathbf{x}_s^\theta}\right)}_{\text{tend to 0 when t is larger}}\underbrace{\left(\frac{\partial\pi_\theta(\mathbf{x}_t^\theta, t, c)}{\partial\theta}\right)}_{\text{optimizing t-th step}}\right]. \tag{5}$$

**Limitations.** Eq. (5) reveals the core optimization mechanism of existing methods: the motions $\mathbf{x}_0^\theta$ are generated via the reverse process $\pi_\theta$, with the full computation graph preserved to enable the maximization of the reward $\mathcal{R}_\phi(\mathbf{x}_0^\theta, c)$. However, as shown in Fig. 2, this optimization incurs severe limitations:

(1) *Memory-intensive and sparse optimization*: Gradient computation over $T$ reverse steps demands storing the entire trajectory $\{\mathbf{x}_t^\theta\}_{t=1}^T$ and corresponding Jacobians, leading to high memory consumption and inefficient, sparse optimization compared to the sampling process.

(2) *Vanishing gradient due to coarse-grained optimization*: Eq. (5) indicates that the optimization of $t$-th noisy step relies on the gradient $\frac{\partial \pi_\theta(\mathbf{x}_t^\theta, t, c)}{\partial \theta}$ with a coefficient $\prod_{s=1}^{t-1} \frac{\partial \pi_\theta(\mathbf{x}_s^\theta, s, c)}{\partial \mathbf{x}_s^\theta}$. However, during optimization, the term $\frac{\partial \pi_\theta(\mathbf{x}_t^\theta, t, c)}{\partial \mathbf{x}_t^\theta}$ tends to converge to 0 (see the blue line in Fig. 3), causing the coefficient $\prod_{s=1}^{t-1} \frac{\partial \pi_\theta(\mathbf{x}_s^\theta, s, c)}{\partial \mathbf{x}_s^\theta}$ to also approach 0 (see the orange line in Fig. 3).

Consequently, the optimization process tends to neglect the contribution of $\frac{\partial \pi_\theta(\mathbf{x}_t^\theta, t, c)}{\partial \theta}$. More importantly, the ignored optimization at these early noise steps may be more crucial than at later steps (Xie & Gong, 2025).

**Motivation.** To address the aforementioned limitations, we argue that the key issue lies in Corollary 1: the computation of $\partial \mathbf{x}_t^\theta / \partial \theta$ recursively depends on $\partial \mathbf{x}_{t+1}^\theta / \partial \theta$, making the computation of $\partial \mathbf{x}_0^\theta / \partial \theta$ reliant on the entire T-step reverse process. This dependency necessitates storing a large computation graph, resulting in substantial memory consumption and delayed optimization. To overcome this, an intuitive insight is introduced: *optimizing the gradient step-by-step during the reverse process.* As illustrated in Fig. 2, step-by-step optimization offers several advantages: *(1) Lower memory consumption and dense optimization*: each update only requires the computation graph of the current step, allowing gradients to be computed and applied immediately instead of waiting until the end of the

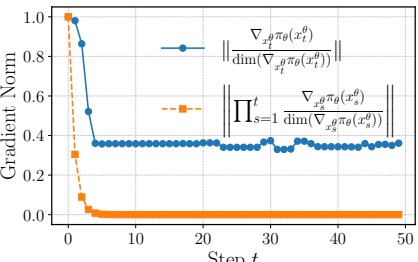

Figure 3: **Gradient norm with respect to denoising steps**. Here, $\dim(\cdot)$ denotes the gradient dimension. Detailed settings are provided in App. B.1.

$T$-step reverse process. *(2) Fine-grained optimization*: each step is optimized independently, so that the update of the $t$-th step does not depend on the vanishing product of coefficients $\prod_{s=1}^{t-1} \frac{\partial \pi_\theta(\mathbf{x}_s^\theta, s, c)}{\partial \mathbf{x}_s^\theta}$.

However, in domains such as image generation, reward are predominantly output-level (Xu et al., 2023) rather than step-aware, since noised states with complex semantics are difficult to interpret. In contrast, motion representations exhibit simpler and more interpretable semantics, thereby making step-aware motion reward viable (Fig. 4; see further details in App. A.8).

Inspired by the above discussion, we propose EasyTune, a step-aware differentiable reward-based fine-tuning framework for diffusion models, introduced in Sec.4.1. Specifically, EasyTune employs a step-aware differentiable reward model designed to

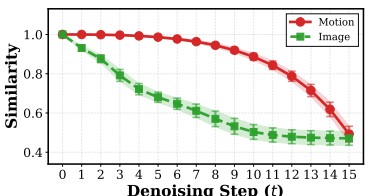

Figure 4: **Similarity between $t$-th step noised and clean motion.**

evaluate noised, rather than clean, motion data, allowing us to perform optimization at each step without storing multi-steps computation graph. Nevertheless, due to the scarcity of human-annotated motion pairs, the primary challenge lies in training such a reward model without any paired data. To address this issue, we present a self-refinement preference learning mechanism, in Sec.4.2, to identify preference data pairs specifically targeting the weaknesses of the pre-trained model, facilitating the acquisition of a reward model.

# 4 METHOD

## 4.1 EFFICIENT STEP-AWARE FINE-TUNING FOR MOTION DIFFUSION

Assuming the reward model for evaluating noisy motion, we aim to propose a step-aware fine-tuning method that reduces the excessive memory usage and performs efficient and fine-grained optimization. As discussed in Sec. 3, limitations of existing methods (Clark et al., 2024; Wu et al., 2025a) stem

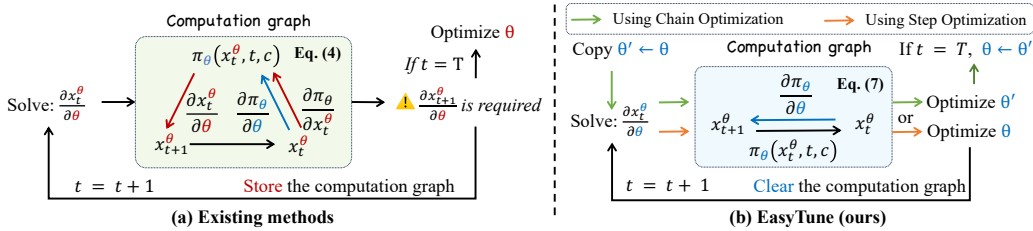

Figure 5: **Core insight of EasyTune.** By replacing the recursive gradient in Eq.(4) with step-level ones in Eq.(7), *EasyTune* removes recursive dependencies, enabling (1) step-wise graph storage, (2) efficiency, and (3) fine-grained optimization. See App. B for pseudocode and discussion.

from the recursive gradients computation. To address these issues, we introduce EasyTune, a simple yet effective method for fine-tuning motion diffusion models. The key idea is to maximize the reward value at each step, allowing the parameter to be optimized at each step without storing the full trajectory, as shown in Fig. 5. Specifically, the training objective function is defined as:

$$\mathcal{L}_{\text{EasyTune}}(\theta) = -\mathbb{E}_{c\sim\mathcal{D}_{\text{T}},\mathbf{x}_t^{\theta}\sim\pi_{\theta}(\cdot|c),t\sim\mathcal{U}(0,T)}\left[\mathcal{R}_{\phi}(\mathbf{x}_t^{\theta},t,c)\right], \tag{6}$$

where $\mathcal{R}_{\phi}(\mathbf{x}_t^{\theta},t,c)$ is the reward value of the *stop gradient* noised motion $\mathbf{x}_t^{\theta}$ at time step $t$, and $\mathcal{U}(1,T)$ is a uniform distribution over the time steps. Here, the stop gradient noised motion $\mathbf{x}_t^{\theta}$ and its gradient w.r.t. the diffusion parameter $\theta$ are represents as:

$$\mathbf{x}_{t-1}^{\theta} = \pi_{\theta}\left(\text{sg}(\mathbf{x}_t^{\theta}),t,c\right) := \frac{1}{\sqrt{\alpha_t}}\left(\text{sg}(\mathbf{x}_t^{\theta}) - \frac{\beta_t}{\sqrt{1-\bar{\alpha}_t}}\epsilon_{\theta}\left(\text{sg}(\mathbf{x}_t^{\theta}),t,c\right)\right), \tag{7}$$

where $\text{sg}(\cdot)$ denotes the stop gradient operations. Eq. (6) and Eq. (7) indicate that EasyTune aims to optimize the diffusion model by maximizing the reward value of the noised motion $\mathbf{x}_t^{\theta}$ at each step $t$.

**Corollary 2.** *Given the reverse process in Eq. (7), the gradient w.r.t. diffusion model $\theta$ is denoted as:*

$$\frac{\partial\mathbf{x}_{t-1}^{\theta}}{\partial\theta} = \frac{\partial\pi_{\theta}\left(\text{sg}(\mathbf{x}_t^{\theta}),t,c\right)}{\partial\theta}. \tag{8}$$

Corollary 2 shows that EasyTune overcomes the recursive gradient issue, enabling efficient, fine-grained updates with substantially reduced memory. As Fig. 6 illustrates, while prior methods incur $\mathcal{O}(T)$ memory by storing the multi-steps trajectory, EasyTune maintains a constant $\mathcal{O}(1)$ memory. Guided by Corollary 2, we optimize the loss function $\mathcal{L}_{\text{EasyTune}}(\theta)$ as follows:

$$\mathcal{L}_{\text{EasyTune}}(\theta) = -\mathbb{E}_{c\sim\mathcal{D}_{\text{T}},\mathbf{x}_t^{\theta}\sim\pi_{\theta}(\cdot|c),t\sim\mathcal{U}(0,T)}\frac{\partial\mathcal{R}_{\phi}(\mathbf{x}_t^{\theta},t,c)}{\partial\mathbf{x}_t^{\theta}} \cdot \frac{\partial\pi_{\theta}\left(\text{sg}(\mathbf{x}_{t+1}^{\theta}),t+1,c\right)}{\partial\theta}. \tag{9}$$

**Discussion of Existing Methods.** Unlike prior methods (Eq. (5)), EasyTune updates the diffusion model $\theta$ using Eq. (6), computing the gradient $\frac{\partial\pi_{\theta}(\text{sg}(\mathbf{x}_t^{\theta}),t,c)}{\partial\theta}$ at each step $t$ without storing the full $\mathcal{O}(T)$-step computation graph. Among related works, the closest is DRTune (Wu et al., 2025a), which also uses stop-gradient operations $\text{sg}(\cdot)$ to solve the limitations of previous methods:

$$\mathbf{x}_{t-1}^{\theta} = \frac{1}{\sqrt{\alpha_t}}\left(\mathbf{x}_t^{\theta} - \frac{\beta_t}{\sqrt{1-\bar{\alpha}_t}}\epsilon_{\theta}\left(\text{sg}(\mathbf{x}_t^{\theta}),t,c\right)\right),$$

$$\frac{\partial\mathbf{x}_{t-1}^{\theta}}{\partial\theta} = \frac{1}{\sqrt{\alpha_t}}\left(\frac{\partial\mathbf{x}_t^{\theta}}{\partial\theta} - \frac{\beta_t}{\sqrt{1-\bar{\alpha}_t}}\frac{\partial\epsilon_{\theta}\left(\text{sg}(\mathbf{x}_t^{\theta}),t,c\right)}{\partial\theta}\right). \tag{10}$$

However, recursive gradient computation remains an issue in existing methods (Eq. (10)). As shown in Fig. 6, their memory usage grows linearly with the number of denoising steps ($\mathcal{O}(T)$), while EasyTune maintains a constant memory footprint ($\mathcal{O}(1)$). These analyses and experiments highlight the efficiency of our method and details discussion are provided in App. B.

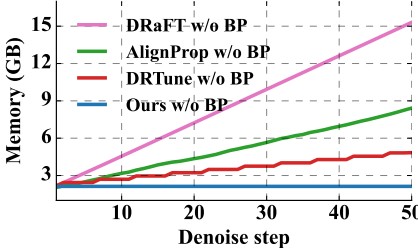

Figure 6: **Memory usage comparison.** Here, "w/o BP" indicates memory measured without backpropagation. Comprehensive analysis are in App. A.7.

## 4.2 Self-Refining Preference Learning for Reward Model

Our goal is to develop a reward model without the requirement of human-labeled data. Existing works (Tan et al., 2025) often repurpose pre-trained text-to-motion retrieval models (Petrovich et al., 2023) to score text-motion alignment. However, this is suboptimal: retrieval models focus on matching positive pairs in a shared embedding space, whereas reward models must distinguish between preferred and non-preferred motions. Given their shared architecture, retrieval models can be fine-tuned for preference learning. The challenge, however, lies in the scarcity of such preference data in the motion domain. To this end, we propose **Self-refining Preference Learning (SPL)**, which leverages a retrieval-based auxiliary task to construct preference pairs for reward learning. SPL involves two steps: (1) *Preference Pair Mining:* retrieve motions for each text; treat the ground-truth as preferred and top incorrect retrieval as non-preferred if it's not retrieved; (2) *Preference Fine-tuning:* updating the encoders to assign higher scores to preferred motions.

**Reward Model.** Given a motion $\mathbf{x}$ and a text description $c$, the reward value is computed based on the similarity between the motion features $\mathbf{x}$ and text features $c$, denoted as:

$$\mathcal{R}_\phi(\mathbf{x}, c) = \mathcal{E}_\mathrm{M}(\mathbf{x}) \cdot \mathcal{E}_\mathrm{T}(c) \cdot \tau, \tag{11}$$

where $\mathcal{E}_\mathrm{M}$ and $\mathcal{E}_\mathrm{T}$ are the motion and text encoders from the pre-trained retrieval model (Weng et al., 2025a), and $\tau$ is a trainable temperature parameter.

Additionally, dealing with noisy motions remains a key challenge in step-level optimization. Current diffusion-based models can be divided into SDE-based (Song et al., 2020) and ODE-based (Lu et al., 2022) models. For ODE-based settings (Dai et al., 2025), thanks to their deterministic sampling, we use the reward value of the coarse clean motion $\hat{\mathbf{x}}_0$ predicted by one-step prediction $\hat{\mathbf{x}}_0 = \pi'_\theta(\mathbf{x}_t, t, c)$ as the final reward value. For both SDE- and ODE-based settings (Tevet et al., 2023), we adopt a noise-aware reward model to accurately calculate their reward values:

$$\mathcal{R}_\phi(\mathbf{x}_t, t, c) = \begin{cases} \mathcal{R}_\phi(\hat{\mathbf{x}}_0, 0, c), & \text{Only for ODE-based settings,} \\ \mathcal{R}_\phi(\mathbf{x}_t, t, c), & \text{For SDE- and ODE-based settings.} \end{cases} \tag{12}$$

**Preference Data Mining.** To identify non-preferred motions often incorrectly retrieved, we retrieve the top-$k$ motions $\mathcal{D}_\mathrm{R}$ from the training set or subset $\mathcal{D}_\mathrm{T}$ given a text condition $c$:

$$\mathcal{D}_\mathrm{R} = \mathrm{top}_k \arg\max_{\mathbf{x} \in \mathcal{D}_\mathrm{T}} \mathcal{R}_\phi(\mathbf{x}, c) = \arg\max_{\mathcal{D}_\mathrm{R} \subset \mathcal{D}_\mathrm{T}, |\mathcal{D}_\mathrm{R}| = k} \sum_{\mathbf{x} \in \mathcal{D}_\mathrm{R}} \mathcal{R}_\phi(\mathbf{x}, c). \tag{13}$$

where $\mathrm{top}_k \arg\max_{\mathbf{x} \in \mathcal{D}_\mathrm{T}} \mathcal{R}_\phi(\mathbf{x}, c)$ denotes the top-$k$ motion with the largest reward value.

Given a ground-truth motion $\mathbf{x}^\mathrm{gt}$ and its text condition $c$, we retrieve the top-$k$ motions $\mathcal{D}_\mathrm{R}$ based on reward scores. If $\mathbf{x}^\mathrm{gt} \notin \mathcal{D}_\mathrm{R}$, we treat it as the preferred motion and the highest-scoring retrieved motion as the non-preferred one; this case provides an informative preference signal. Otherwise, the retrieval model already ranks the ground-truth among the top-$k$, so no informative negative exists; we set $\mathbf{x}^\mathrm{l} = \mathbf{x}^\mathrm{gt}$ and assign $\mathcal{Q} = (0.5, 0.5)$, which yields zero KL gradient and effectively skips optimization for this sample:

$$\mathbf{x}^\mathrm{w} = \mathbf{x}^\mathrm{gt}, \quad \mathbf{x}^\mathrm{l} = \begin{cases} \arg\max_{\mathbf{x} \in \mathcal{D}_\mathrm{R}} \mathcal{R}_\phi(\mathbf{x}, c), & \text{if } \mathbf{x}^\mathrm{gt} \notin \mathcal{D}_\mathrm{R}, \\ \mathbf{x}^\mathrm{gt}, & \text{otherwise.} \end{cases} \tag{14}$$

**Preference Fine-tuning.** Given a preference pair consisting of a preferred motion $\mathbf{x}^\mathrm{w}$ and a non-preferred motion $\mathbf{x}^\mathrm{l}$, we compute their reward scores and convert them into softmax probabilities:

$$\mathcal{P} = \left( \mathcal{P}(\mathbf{x}^\mathrm{w}, c), \mathcal{P}(\mathbf{x}^\mathrm{l}, c) \right) = \mathrm{Softmax} \left( \mathcal{R}_\phi(\mathbf{x}^\mathrm{w}, c), \mathcal{R}_\phi(\mathbf{x}^\mathrm{l}, c) \right). \tag{15}$$

Following Pick-a-pic (Kirstain et al., 2023), we optimize model by aligning the predicted softmax distribution $\mathcal{P}$ with a target distribution $\mathcal{Q}$, which reflects the ground-true preference between $\mathbf{x}^\mathrm{w}$ and $\mathbf{x}^\mathrm{l}$, defined as:

$$\mathcal{Q} = \begin{cases} (1.0, 0.0), & \text{if } \mathbf{x}^\mathrm{w} \text{ is preferred over } \mathbf{x}^\mathrm{l}, \\ (0.5, 0.5), & \text{if } \mathbf{x}^\mathrm{w} = \mathbf{x}^\mathrm{l}. \end{cases} \tag{16}$$

Formally, the target distribution $\mathcal{Q}$ encodes the preference between a preferred motion $\mathbf{x}^\mathrm{w}$ and a non-preferred motion $\mathbf{x}^\mathrm{l}$. If $\mathbf{x}^\mathrm{w}$ is preferred, we set $\mathcal{Q} = (1.0, 0.0)$, encouraging $\mathcal{P}(\mathbf{x}^\mathrm{w}, c) \to 1$ and $\mathcal{P}(\mathbf{x}^\mathrm{l}, c) \to 0$. If the two are identical ($\mathbf{x}^\mathrm{w} = \mathbf{x}^\mathrm{l}$), we set $\mathcal{Q} = (0.5, 0.5)$, indicating no preference.

To optimize the reward model $\phi$, we minimize the KL divergence between them:

$$\mathcal{L}_{\mathrm{SPL}}(\phi) = \mathrm{D}_{\mathrm{KL}}(\mathcal{Q} \parallel \mathcal{P}) = \sum_{\mathbf{x} \in \{\mathbf{x}^w, \mathbf{x}^l\}} \mathcal{Q}(\mathbf{x}, c) \log \frac{\mathcal{Q}(\mathbf{x}, c)}{\mathcal{P}(\mathbf{x}, c)}. \tag{17}$$

where $\mathcal{Q}$ and $\mathcal{P}$ are the target and reward distribution. By mining preference motion pairs by *preference data mining*, we can fine-tune pre-trained retrieval models by Eq. (17), to obtain reward modesl without human-annotated data. Detailed process are described in App. B.4.

## 5 EXPERIMENT

### 5.1 EXPERIMENTAL SETUP

**Datasets & Evaluation.** We conduct experiments on HumanML3D (Guo et al., 2022a) and KIT-ML (Plappert et al., 2016). Following standard practice (Guo et al., 2023; Li et al., 2025), we report R-Precision@$k$, Fréchet Inception Distance (FID), Multi-Modal Distance (MM Dist), and Diversity. We also measure peak memory usage to assess efficiency. Additionally, our SPL mechanism is evaluated using R-Precision@$k$ under the previous setup (Li et al., 2025).

**Implementation.** Our method consists of two components: fine-tuning the diffusion model with EasyTune and fine-tuning a pretrained retrieval model to obtain the reward model. For EasyTune, we evaluate pretrained backbones—MLD (Chen et al., 2023), MLD++ (Dai et al., 2025), MotionLCM (Dai et al., 2024), and MDM (Tevet et al., 2023)—with hyperparameters: a learning rate of $1 \times 10^{-5}$ and a batch size of 256. EasyTune is benchmarked against differentiable reward-based baselines with their official hyperparameters, detailed in App.A.1. We fine-tune the reward model initialized with ReAlign (Weng et al., 2025a) using our SPL with top-$K$ samples ($K = 10$). Then, the reward model is frozen and provides supervision for optimizing the diffusion model. The experimental results for the ODE-based model, using both reward computation methods from Eq. (12), are provided. Results corresponding to the first and second terms are presented in Tab. 1 and 2, and Tab. 3, respectively. Experiments are conducted on a single NVIDIA RTX A6000 GPU with 48GB memory, detailed overhead is provided in App. A.7.

### 5.2 EVALUATION ON MOTION DIFFUSION FINE-TUNING

**Comparison with SoTA Fine-Tuning Methods.** To assess the effectiveness and efficiency of Easy-Tune, we compare it with recent state-of-the-art fine-tuning methods, including DRaFT, AlignProp, and DRTune, as shown in Tab. 1. EasyTune consistently achieves the best overall performance across key metrics, including R-Precision, FID (0.132, +70.7%), MM Dist (2.637, +13.6%), and Diversity, while also requiring the least GPU memory (22.10 GB). We attribute these gains to two core designs: optimizing rewards at each denoising step for finer supervision, and discarding redundant computation graphs to reduce memory usage.

**Efficiency of the Optimization.** To assess convergence efficiency, we compare optimization curves of fine-tuning methods in Fig. S1 (in App. A). EasyTune converges faster and achieves consistently

Table 1: **Comparison of fine-tuning methods on HumanML3D.** Arrows $\uparrow$, $\downarrow$, and $\rightarrow$ indicate that higher, lower, and closer to real values are better. **Bold** and underline denote the best and second-best results. All methods adopt the noise-aware reward $\mathcal{R}_\phi(\mathbf{x}_t, t, c)$.

| Method | R Precision $\uparrow$ | | | FID $\downarrow$ | MM Dist $\downarrow$ | Diversity $\rightarrow$ | Memory (GB) $\downarrow$ |
|---|---|---|---|---|---|---|---|
| | Top 1 | Top 2 | Top 3 | | | | |
| Real | 0.511 | 0.703 | 0.797 | 0.002 | 2.974 | 9.503 | - |
| MLD (Base Model) | $0.504_{\pm.002}$ | $0.698_{\pm.003}$ | $0.796_{\pm.002}$ | $0.450_{\pm.011}$ | $3.052_{\pm.009}$ | $9.634_{\pm.064}$ | 15.21 |
| w/ ReFL-10 (Clark et al., 2024) | $0.533_{+5.8\%}$ | $0.720_{+3.2\%}$ | $0.821_{+3.1\%}$ | $0.207_{+54.0\%}$ | $2.852_{+6.6\%}$ | $10.129_{-0.495}$ | $\mathbf{22.10}_{+6.89}$ |
| w/ ReFL-20 (Clark et al., 2024) | $0.528_{+4.8\%}$ | $0.718_{+2.9\%}$ | $0.813_{+2.1\%}$ | $0.241_{+46.4\%}$ | $2.883_{+5.5\%}$ | $10.189_{-0.555}$ | $\mathbf{22.10}_{+6.89}$ |
| w/ DRaFT-10 (Clark et al., 2024) | $0.565_{+12.1\%}$ | $0.757_{+8.5\%}$ | $0.846_{+6.3\%}$ | $0.195_{+56.7\%}$ | $2.703_{+11.4\%}$ | $9.851_{-0.217}$ | $26.56_{+11.35}$ |
| w/ DRaFT-50 (Clark et al., 2024) | $0.528_{+4.8\%}$ | $0.724_{+3.7\%}$ | $0.819_{+2.9\%}$ | $0.197_{+56.2\%}$ | $2.872_{+5.9\%}$ | $\underline{9.641}_{-0.007}$ | $37.32_{+22.11}$ |
| w/ AlignProp (Prabhudesai et al., 2023) | $0.560_{+11.1\%}$ | $0.753_{+7.9\%}$ | $0.841_{+5.7\%}$ | $0.266_{+40.9\%}$ | $2.739_{+10.3\%}$ | $9.877_{-0.243}$ | $30.40_{+15.19}$ |
| w/ DRTune (Wu et al., 2025a) | $0.549_{+8.9\%}$ | $0.746_{+6.9\%}$ | $0.836_{+5.0\%}$ | $0.313_{+30.4\%}$ | $2.795_{+8.4\%}$ | $9.930_{-0.296}$ | $27.01_{+11.80}$ |
| w/ EasyTune (Ours, Step Optimization) | $\mathbf{0.581}_{+15.3\%}$ | $\mathbf{0.769}_{+10.2\%}$ | $\mathbf{0.855}_{+7.4\%}$ | $\mathbf{0.132}_{+70.7\%}$ | $\underline{2.637}_{+13.6\%}$ | $\mathbf{9.465}_{+0.093}$ | $\mathbf{22.10}_{+6.89}$ |
| w/ EasyTune (Ours, Chain Optimization) | $\underline{0.574}_{+13.9\%}$ | $\underline{0.766}_{+9.7\%}$ | $\underline{0.854}_{+7.3\%}$ | $\underline{0.172}_{+61.8\%}$ | $\mathbf{2.614}_{+14.3\%}$ | $9.348_{-0.024}$ | $24.21_{+9.00}$ |

Table 2: **Comparison of text-to-motion generation performance on the HumanML3D dataset.**

| Method | R Precision ↑ | | | FID ↓ | MM Dist ↓ | Diversity → |
|---|---|---|---|---|---|---|
| | Top 1 | Top 2 | Top 3 | | | |
| Real | 0.511 | 0.703 | 0.797 | 0.002 | 2.974 | 9.503 |
| TM2T (Guo et al., 2022b) | $0.424^{\pm0.003}$ | $0.618^{\pm0.003}$ | $0.729^{\pm0.002}$ | $1.501^{\pm0.017}$ | $3.467^{\pm0.011}$ | $8.589^{\pm0.076}$ |
| T2M (Guo et al., 2022a) | $0.455^{\pm0.002}$ | $0.636^{\pm0.003}$ | $0.736^{\pm0.003}$ | $1.087^{\pm0.002}$ | $3.347^{\pm0.008}$ | $9.175^{\pm0.002}$ |
| MDM (Tevet et al., 2023) | $0.455^{\pm0.006}$ | $0.645^{\pm0.007}$ | $0.749^{\pm0.006}$ | $0.489^{\pm0.047}$ | $3.330^{\pm0.025}$ | $9.920^{\pm0.083}$ |
| T2M-GPT (Zhang et al., 2023a) | $0.492^{\pm0.003}$ | $0.679^{\pm0.002}$ | $0.775^{\pm0.002}$ | $0.141^{\pm0.005}$ | $3.121^{\pm0.009}$ | $9.722^{\pm0.082}$ |
| ReMoDiffuse (Zhang et al., 2023b) | $0.510^{\pm0.005}$ | $0.698^{\pm0.006}$ | $0.795^{\pm0.004}$ | $0.103^{\pm0.004}$ | $2.974^{\pm0.016}$ | $9.018^{\pm0.075}$ |
| AttT2M (Zhong et al., 2023) | $0.499^{\pm0.003}$ | $0.690^{\pm0.002}$ | $0.786^{\pm0.002}$ | $0.112^{\pm0.006}$ | $3.038^{\pm0.007}$ | $9.700^{\pm0.090}$ |
| MotionDiffuse (Zhang et al., 2024a) | $0.491^{\pm0.001}$ | $0.681^{\pm0.001}$ | $0.775^{\pm0.001}$ | $0.630^{\pm0.001}$ | $3.113^{\pm0.001}$ | $9.410^{\pm0.049}$ |
| MotionLCM (Dai et al., 2024) | $0.502^{\pm0.003}$ | $0.698^{\pm0.002}$ | $0.798^{\pm0.002}$ | $0.304^{\pm0.012}$ | $3.012^{\pm0.007}$ | $9.607^{\pm0.066}$ |
| MotionMamba (Zhang et al., 2024b) | $0.502^{\pm0.003}$ | $0.693^{\pm0.002}$ | $0.792^{\pm0.002}$ | $0.281^{\pm0.011}$ | $3.060^{\pm0.000}$ | $9.871^{\pm0.084}$ |
| CoMo (Huang et al., 2024) | $0.502^{\pm0.002}$ | $0.692^{\pm0.007}$ | $0.790^{\pm0.002}$ | $0.262^{\pm0.004}$ | $3.032^{\pm0.015}$ | $9.936^{\pm0.066}$ |
| ParCo (Zou et al., 2024) | $0.515^{\pm0.003}$ | $0.706^{\pm0.003}$ | $0.801^{\pm0.002}$ | $0.109^{\pm0.005}$ | $2.927^{\pm0.008}$ | $9.576^{\pm0.088}$ |
| SoPo (Tan et al., 2025) | $0.528^{\pm0.005}$ | $0.722^{\pm0.004}$ | $0.827^{\pm0.004}$ | $0.174^{\pm0.005}$ | $2.939^{\pm0.011}$ | $9.584^{\pm0.074}$ |
| MLD (Chen et al., 2023) (Base Model) | $0.504^{\pm0.002}$ | $0.698^{\pm0.002}$ | $0.796^{\pm0.002}$ | $0.450^{\pm0.011}$ | $3.052^{\pm0.009}$ | $9.634^{\pm0.064}$ |
| w/ EasyTune (Ours) | $0.581^{\pm0.003}_{+15.3\%}$ | $0.769^{\pm0.002}_{+10.2\%}$ | $0.855^{\pm0.002}_{+7.4\%}$ | $0.132^{\pm0.005}_{+70.7\%}$ | $2.637^{\pm0.007}_{+13.6\%}$ | $\mathbf{9.465}^{\pm0.075}_{+0.09}$ |
| MLD++ (Dai et al., 2025) (Base Model) | $0.548^{\pm0.003}$ | $0.738^{\pm0.003}$ | $0.829^{\pm0.002}$ | $0.073^{\pm0.003}$ | $2.810^{\pm0.008}$ | $9.658^{\pm0.089}$ |
| w/ EasyTune (Ours) | $\mathbf{0.591}^{\pm0.004}_{+7.8\%}$ | $\mathbf{0.777}^{\pm0.002}_{+5.3\%}$ | $\mathbf{0.859}^{\pm0.002}_{+3.6\%}$ | $\mathbf{0.069}^{\pm0.003}_{+6.8\%}$ | $\mathbf{2.592}^{\pm0.008}_{+7.8\%}$ | $9.705^{\pm0.086}_{-0.06}$ |

lower loss, suggesting better local optima with higher reward values. This improvement stems from its fine-grained, step-wise optimization, in contrast to the sparser, trajectory-level updates used in prior work (Clark et al., 2024), enabling more precise gradient signals and accelerated training. Furthermore, EasyTune achieves a **7.3×** training speedup over DRaFT to reach the same reward level, detailed computational overhead analysis is provided in App. A.7.

## 5.3 EVALUATION ON TEXT-TO-MOTION GENERATION

**Comparison with SoTA Text-to-Motion Methods.** We evaluate EasyTune on text-to-motion generation using MLD (Chen et al., 2023) and MLD++ (Dai et al., 2025) as base models, comparing with state-of-the-art methods on the HumanML3D (Guo et al., 2022a) and KIT-ML (Plappert et al., 2016) datasets, as shown in Tab. 2 and S3 (in App. A.3). On HumanML3D, EasyTune improves the R-P@1 of MLD from 0.504 to 0.581 and MLD++ from 0.548 to 0.591, surpassing baselines like ParCo (Zou et al., 2024) (0.515) and ReMoDiffuse (Zhang et al., 2023b) (0.510). It also achieves the best MM Dist (2.637 and 2.592) and competitive FID (0.132 and 0.069).

**Generalization across Different Pretrained Models.** To evaluate the generalization of EasyTune across pretrained text-to-motion models, we applied it to MLD (Chen et al., 2023), MLD++ (Dai et al., 2025), $\text{MLCM}^{1S}$ (Dai et al., 2024), and $\text{MDM}^{50S}$ (Tevet et al., 2023). As shown in Tab. 3, EasyTune consistently improved performance. For instance, MLD saw a 12.7% increase in R-P@1 (0.568) and a 56.9% reduction in FID (0.194). MLD++ achieved a 6.0% gain in R-Precision@1 (0.581)

Table 3: **Performance enhancement of diffusion-based motion generation methods.** For ODE samplings (MLD, MLD++, MLCM), we adopt the one-step prediction reward.

| Method | R Precision ↑ | | | FID ↓ | MM Dist ↓ | Diversity → |
|---|---|---|---|---|---|---|
| | Top 1 | Top 2 | Top 3 | | | |
| Real | 0.511 | 0.703 | 0.797 | 0.002 | 2.974 | 9.503 |
| MLD | 0.504 | 0.698 | 0.796 | 0.450 | 3.052 | 9.634 |
| w/ EasyTune | $0.568_{+12.7\%}$ | $0.754_{+8.0\%}$ | $0.846_{+6.3\%}$ | $0.194_{+56.9\%}$ | $2.672_{+12.5\%}$ | $9.368_{-0.00}$ |
| MLD++ (Dai et al., 2025) | 0.548 | 0.738 | 0.829 | 0.073 | 2.810 | 9.658 |
| w/ EasyTune | $0.581_{+6.0\%}$ | $0.762_{+3.3\%}$ | $0.849_{+2.4\%}$ | $0.073_{+0.0\%}$ | $2.603_{+7.4\%}$ | $9.719_{-0.06}$ |
| $\text{MLCM}^{1S}$ (Dai et al., 2024) | 0.502 | 0.701 | 0.803 | 0.467 | 3.052 | 9.631 |
| w/ EasyTune | $0.571_{+13.7\%}$ | $0.766_{+9.3\%}$ | $0.854_{+6.4\%}$ | $0.188_{+59.7\%}$ | $2.647_{+13.3\%}$ | $9.692_{-0.06}$ |
| $\text{MLCM}^{4S}$ (Dai et al., 2024) | 0.502 | 0.698 | 0.798 | 0.304 | 3.012 | 9.607 |
| w/ EasyTune | $0.565_{+12.5\%}$ | $0.760_{+8.9\%}$ | $0.848_{+6.3\%}$ | $0.200_{+34.2\%}$ | $2.691_{+10.7\%}$ | $9.812_{-0.21}$ |
| $\text{MDM}^{50S}$ (Tevet et al., 2023) | 0.455 | 0.645 | 0.749 | 0.489 | 3.330 | 9.920 |
| w/ EasyTune | $0.472_{+3.7\%}$ | $0.679_{+5.3\%}$ | $0.787_{+5.1\%}$ | $0.411_{+16.0\%}$ | $3.117_{+6.4\%}$ | $9.239_{+0.15}$ |
| Mo.Diffuse (Zhang et al., 2024a) | 0.491 | 0.681 | 0.775 | 0.630 | 3.113 | 9.410 |
| w/ EasyTune | $0.488_{-0.6\%}$ | $0.686_{+0.7\%}$ | $0.788_{+1.7\%}$ | $0.556_{+11.7\%}$ | $3.068_{+1.4\%}$ | $9.215_{-0.20}$ |

and a 7.4% improvement in MM Dist (2.603). $\text{MLCM}^{1S}$ and $\text{MDM}^{50S}$ also showed significant FID reductions of 59.7% and 16.0%, respectively. These results highlight the generalization of EasyTune across various diffusion-based architectures.

## 5.4 ABLATION STUDY, USER STUDY & VISUALIZATION

**Self-refinement Preference Learning for Reward Model Training.** We evaluate SPL on text-motion retrieval, comparing it with ReAlign and state-of-the-art methods. As shown in Tab. 4, SPL boosts ReAlign (Weng et al., 2025a) on HumanML3D (R@1: 69.31, +2.5%; R@3: 88.66, +1.4%; motion-to-text R@1: 70.23, +1.9%), outperforming LaMP (Li et al., 2025) and TMR (Petrovich

Table 4: **Evaluation on text-motion retrieval benchmark, HumanML3D and KIT-ML.** The column "Noise" indicates whether the method can handle noisy motion from the denoised process.

| | Methods | Noise | Text-Motion Retrieval↑ | | | | | Motion-Text Retrieval↑ | | | | |
|---|---|---|---|---|---|---|---|---|---|---|---|---|
| | | | R@1 | R@2 | R@3 | R@5 | R@10 | R@1 | R@2 | R@3 | R@5 | R@10 |
| HumanML3D | TEMOS (Petrovich et al., 2022) | ✗ | 40.49 | 53.52 | 61.14 | 70.96 | 84.15 | 39.96 | 53.49 | 61.79 | 72.40 | 85.89 |
| | T2M (Guo et al., 2022a) | ✗ | 52.48 | 71.05 | 80.65 | 89.66 | 96.58 | 52.00 | 71.21 | 81.11 | 89.87 | 96.78 |
| | TMR (Petrovich et al., 2023) | ✗ | 67.16 | 81.32 | 86.81 | 91.43 | 95.36 | 67.97 | 81.20 | 86.35 | 91.70 | 95.27 |
| | LaMP (Li et al., 2025) | ✗ | 67.18 | 81.90 | 87.04 | 92.00 | 95.73 | 68.02 | 82.10 | 87.50 | 92.20 | 96.90 |
| | ReAlign (Weng et al., 2025a) (Base Model) | ✓ | 67.59 | 82.24 | 87.44 | 91.97 | 96.28 | 68.94 | 82.86 | 87.95 | 92.44 | 96.28 |
| | w/ SPL(Ours) | ✓ | **69.31** | **83.71** | **88.66** | **92.81** | **96.75** | **70.23** | **83.41** | **88.72** | **93.07** | **97.04** |
| KIT-ML | T2MOS (Petrovich et al., 2022) | ✗ | 43.88 | 58.25 | 67.00 | 74.00 | 84.75 | 41.88 | 55.88 | 65.62 | 75.25 | 85.75 |
| | T2M (Guo et al., 2022a) | ✗ | 42.25 | 62.62 | 75.12 | 87.50 | 96.12 | 39.75 | 62.75 | 73.62 | 86.88 | 95.88 |
| | TMR (Petrovich et al., 2023) | ✗ | 49.25 | 69.75 | 78.25 | 87.88 | 95.00 | 50.12 | 67.12 | 76.88 | 88.88 | 94.75 |
| | LaMP (Li et al., 2025) | ✗ | 52.50 | **74.80** | **84.70** | 92.70 | 97.60 | 54.00 | 75.30 | 84.40 | 92.20 | **97.60** |
| | ReAlign (Weng et al., 2025a) (Base Model) | ✓ | 52.84 | 71.66 | 82.96 | 91.19 | 97.59 | 52.98 | 72.87 | 84.38 | 92.61 | 96.87 |
| | w/ SPL(Ours) | ✓ | **53.27** | 73.58 | 84.52 | **93.18** | **97.73** | **55.11** | 75.28 | **86.36** | **93.18** | 97.44 |

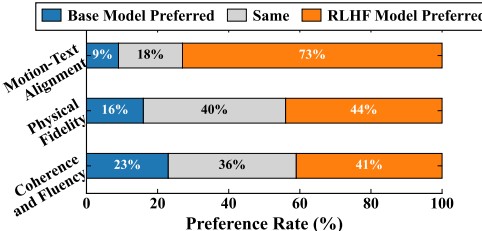

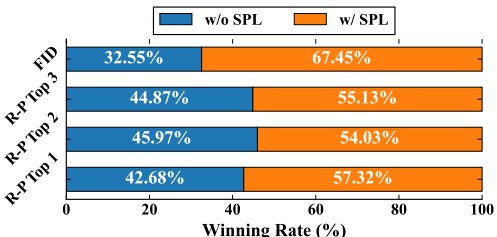

Figure 7: **User study on HumanML3D test set.** We use MLD model as base model.

Figure 8: **Comparison of models fine-tuned with and without SPL.**

et al., 2023). On KIT-ML, SPL achieves R@5 of 93.18 (+2.2%), and motion-to-text R@3 of 86.36 (+2.3%), consistently surpassing baselines.

**Self-refinement Preference Learning for Fine-Tuning.** To evaluate the effect of SPL, we fine-tune MLD (Chen et al., 2023) using reward models trained with and without SPL, and compare their win rates across epochs, as shown in Fig. 8. The model with SPL consistently outperforms the baseline, achieving win rates of 57.32%, 54.03%, 55.13%, and 67.45% on R-P Top 1, Top 2, Top 3, and FID, respectively. This shows that SPL can improve motion generation by enhancing the reward model.

**Human Evaluation.** To assess whether our fine-tuned model exhibits reward hacking, we conducted a user study and visualized the corresponding motions. These visualizations are presented in Fig. 7, with additional visualization in Fig. S3 (App. A.4). Results shows that our method enhance the alignment, fidelity, and coherence of generated motions. A more detailed discussion and further experimental results are provided in App. A.2.

# 6 CONCLUSION

In this work, we theoretically identify recursive dependence in denoising trajectories as the key limitation in aligning motion generative models. Our proposed **EasyTune** method decouples this dependence, enabling denser, more memory-efficient, and fine-grained optimization. Combined with the **SPL** mechanism to dynamically generate preference pairs, experimental results demonstrate that EasyTune significantly outperforms existing methods while requiring less memory overhead.

**Limitation.** In this work, we focus on improving semantic alignment in text-to-motion generation. Due to limited preference data, SPL relies on retrieval-based mining, which can introduce noisy or ambiguous pairs and provides limited physical grounding in reward design. Encouragingly, we find that the semantic reward still implicitly distinguishes real motions from generated ones (App. A.9). Building a unified reward model that explicitly captures both physical plausibility and semantic alignment remains an important direction for future work.

## ACKNOWLEDGMENTS

This research is supported by the National Natural Science Foundation of China (Nos.52441503 and 62302093), the Natural Science Foundation of Jiangsu Province (Nos.BK20230833), and the Big Data Computing Center of Southeast University.

## ETHICS STATEMENT

Our work on EasyTune, a method for fine-tuning motion generative models, introduces several ethical considerations that warrant careful discussion. As our method is designed to align existing generative diffusion models, it inherits the potential biases and limitations of these foundational models. The large-scale motion datasets used to train these base models may contain demographic biases (e.g., representation of age, gender, or physical ability) or may underrepresent certain types of human movement.

Furthermore, like other generative technologies, motion generation models could be misused by malicious actors. The ability to create realistic human motions could be exploited to generate convincing deepfakes or synthetic media for the purpose of disinformation, harassment, or creating non-consensual content. As EasyTune makes the process of aligning models to specific objectives more efficient, it could inadvertently lower the barrier for adapting these models to generate harmful or undesirable motions.

Finally, the advancement of motion generation technology may have a significant socio-economic impact. On one hand, such tools could automate tasks traditionally performed by animators, choreographers, and motion capture actors, potentially displacing jobs in creative industries. On the other hand, EasyTune could also serve as a powerful creative tool, democratizing animation and enabling new forms of artistic expression for independent creators and small studios. It also holds potential for positive applications in fields like robotics, virtual reality, and physical rehabilitation. We believe that continued research and community dialogue are essential to mitigate the risks while harnessing the benefits of this technology.

## REPRODUCIBILITY STATEMENT

We commit to releasing all *code, model weights, and baseline implementations* upon acceptance. To ensure the reproducibility of our experiments, we put the key parts in Appendix A.1. For datasets, we use open source datasets described in Sec. 5.1. For generated results, we upload generated videos to the supplementary material.

## LARGE LANGUAGE MODELS USAGE STATEMENT

We used Large Language Models (LLMs) as auxiliary tools during the preparation of this manuscript. In particular, LLMs were employed to polish the language, improve grammar, and enhance readability of the text. All conceptual ideas, technical contributions, analyses, and conclusions presented in this work are entirely our own and were developed independently of LLM assistance. The models were not used to generate novel scientific content, perform data analysis, or contribute to the design of experiments. We have carefully verified all statements and ensured that the final version of the manuscript accurately reflects our intended meaning and contributions.

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

# EasyTune: Efficient Step-Aware Fine-Tuning for Diffusion-Based Motion Generation

## Supplementary Material

### Supplementary Contents

This supplementary document provides additional experimental results, technical discussions, and theoretical analysis. It is organized as follows: Sec. A presents extended experimental results, including baseline settings, reward hacking analysis, text-to-motion generation performance on KIT-ML, qualitative visualizations, ablation studies on step-level reweighting and sensitivity analyses, efficiency analyses, cross-domain noise-perception comparison, and failure case analysis. Sec. B offers in-depth discussions on gradient analysis, connections to policy gradient methods, existing fine-tuning methods, SPL details, and noise-aware reward strategies. Sec. C contains theoretical proofs, including Corollary 1, convergence analysis, and the derivation of Eq. (5).

## A  More Experimental Results

### A.1  Experimental Setting for Baseline

The hyperparameter configurations for our EasyTune method and the baseline are provided, with the majority of the settings following the official settings, as presented in Tab. S1.

Table S1: **Hyperparameters for EasyTune and baseline methods.**

| Hyperparameter | MLD | AlignProp | ReFL-10 | ReFL-20 | DRaFT-10 | DRaFT-50 | DRTune | EasyTune (Ours) | Description |
|---|---|---|---|---|---|---|---|---|---|
| Random Seed | 1234 | 1234 | 1234 | 1234 | 1234 | 1234 | 1234 | 1234 | Seed for reproducibility |
| Batch Size | 256 | 256 | 256 | 256 | 256 | 256 | 256 | 256 | Training examples per step |
| Gradient Clip Range | – | 1e0 | 1e0 | 1e0 | 1e0 | 1e0 | 1e0 | 1e0 | Maximum gradient norm |
| Checkpoint Interval | – | 1000 | 1000 | 1000 | 1000 | 1000 | 1000 | 1000 | Steps between checkpoints |
| Reward Weight | – | 1e-1 | 1e-1 | 1e-1 | 1e-1 | 1e-1 | 1e-1 | 1e-1 | Weight for reward loss |
| Learning Rate | 1e-5 | 1e-5 | 1e-5 | 1e-5 | 1e-5 | 1e-5 | 1e-5 | 1e-5 | Optimizer learning rate |
| Timestep Range $K$ | – | – | 10 | 20 | [40,50] | [0,50] | [40,50] | [0,50] | Denoising range for fine-tuning |
| Total Steps $T$ | 50 | 50 | 50 | 50 | 50 | 50 | 50 | 50 | Denoising scheduler steps |
| Early-Stop Range $M$ | – | – | – | – | – | – | [40,50] | – | Early-stop timestep range |
| TBPTT Length $P$ | – | 25 | – | – | – | – | – | – | Truncated BPTT window size |

## A.2 EXPERIMENTAL RESULTS AND DISCUSSION ABOUT REWARD HACKING

**Discussion about Reward Hacking.** Previous studies have discussed reward-based differentiable approaches to mitigating reward hacking. Specifically, the proposed strategies include early stopping on a validation set, fine-tuning with LoRA instead of full-parameter tuning, and incorporating KL-divergence regularization. Importantly, our method can be seamlessly combined with these strategies. In our implementation, we provide support for early stopping on a validation set as well as fine-tuning using LoRA.

Additionally, EasyTune is compatible with established techniques like KL regularization and multi-aspect rewards, which effectively mitigate overfitting, as shown in prior work (D-RaFT, AlignProp, DTune). Following these methods, we omitted KL regularization in our main loss:

$$\mathcal{L}_{\text{EasyTune}}^{\text{KL}}(\theta) = -\mathbb{E}_{c \sim \mathcal{D}_{\text{T}}, \mathbf{x}_t^\theta \sim \pi_\theta(\cdot|c), t \sim \mathcal{U}(1,T)} \left[ \mathcal{R}_\phi(\mathbf{x}_t^\theta, t, c) + \mathbb{D}_{\text{KL}}(\mathbf{x}_t^\theta | \mathbf{x}_t^{\theta'}) \right], \tag{S1}$$

**Experimental Results about Reward Hacking.** Quantitative analysis (Fig.S3) confirms that our method is free from reward hacking. To further investigate this behavior, we conducted a user study adn compare the baseline method with a variant of our approach that incorporates KL-divergence regularization. Specifically, we generated motions for the first 100 prompts in the HumanML3D test set using both the pre- and post-finetuned models. Each generated motion was independently evaluated by five participants. Using MDM and MLD as base models, the results—presented in Fig. S2 and Fig. 7, respectively—show that our method consistently outperforms the baseline models in human evaluations without exhibiting signs of reward hacking.

Additionally, Tab. S2 provides a detailed comparison between the baseline and the KL-regularized variants. Results shows that KL regularization helps mitigate overfitting and improves diversity, although at a slight cost in generation quality.

## A.3 TEXT-TO-MOTION GENERATION EVALUATION ON KIT-ML DATASET

Tab. S3 presents the quantitative performance of text-to-motion generation models on the KIT-ML dataset, evaluated across multiple metrics: R-Precision (Top-1, Top-2, Top-3) for text-motion alignment, Frechet Inception Distance (FID) for motion quality, Multi-Modal Distance (MM Dist) for semantic relevance, and Diversity for motion variety. The analysis compares models enhanced with EasyTune against a comprehensive set of baselines.

The results reveal significant improvements in models enhanced with EasyTune. For MDM (Chen et al., 2023), Top-1 R-Precision increases from 0.403 to 0.442 (a 9.7% gain), Top-2 R-Precision from 0.606 to 0.655 (8.1% gain), and Top-3 R-Precision from 0.731 to 0.773 (5.7% gain). FID of MDM decreases substantially from 0.497 to 0.284 (42.9% improvement), indicating enhanced motion quality. MM Dist of MDM improves from 3.096 to 2.755 (11.0% reduction), reflecting

Table S2: **Performance comparison between EasyTune with and without KL-regularized.**

| Method | R@P1 | R@P2 | R@P3 | FID ↓ | MM-Dist ↓ | Div. ↑ | Memory ↓ |
|---|---|---|---|---|---|---|---|
| EasyTune | **0.581** | **0.769** | **0.855** | **0.132** | **2.637** | 9.465 | **22.10** |
| EasyTune+KL | 0.575 | 0.763 | 0.846 | 0.172 | 2.674 | **9.482** | 29.70 |

Table S3: **Comparison of text-to-motion generation performance on the KIT-ML dataset.**

| Method | R Precision ↑ | | | FID ↓ | MM Dist ↓ | Diversity → |
|---|---|---|---|---|---|---|
| | Top 1 | Top 2 | Top 3 | | | |
| Real | 0.424 | 0.649 | 0.779 | 0.031 | 2.788 | 11.08 |
| TM2T (Guo et al., 2022b) | 0.280 | 0.463 | 0.587 | 3.599 | 4.591 | 9.473 |
| T2M (Guo et al., 2022a) | 0.361 | 0.559 | 0.681 | 3.022 | 2.052 | 10.72 |
| M2DM (Kong et al., 2023) | 0.416 | 0.628 | 0.743 | 0.515 | 3.015 | 11.42 |
| T2M-GPT (Zhang et al., 2023a) | 0.416 | 0.627 | 0.745 | 0.514 | 3.007 | 10.86 |
| Fg-T2M (Wang et al., 2023) | 0.418 | 0.626 | 0.745 | 0.571 | 3.114 | 10.93 |
| AttT2M (Zhong et al., 2023) | 0.413 | 0.632 | 0.751 | 0.870 | 3.039 | 10.96 |
| MotionMamba (Zhang et al., 2024b) | 0.419 | 0.645 | 0.765 | 0.307 | 3.021 | 11.02 |
| CoMo (Huang et al., 2024) | 0.422 | 0.638 | 0.765 | 0.332 | 2.873 | 10.95 |
| ParCo (Zou et al., 2024) | 0.430 | 0.649 | 0.772 | 0.453 | 2.820 | 10.95 |
| SiT (Meng et al., 2025) | 0.387 | 0.610 | 0.749 | **0.242** | - | - |
| MDM (Chen et al., 2023) | 0.403 | 0.606 | 0.731 | 0.497 | 3.096 | 10.76 |
| w/ EasyTune (ours) | **0.442**+9.7% | **0.655**+8.1% | 0.773+5.7% | 0.284+42.9% | **2.755**+11.0% | 11.27+0.13 |
| MoDiffuse (Zhang et al., 2024a) | 0.417 | 0.621 | 0.739 | 1.954 | 2.958 | **11.10** |
| w/ EasyTune (ours) | 0.438+5.0% | 0.649+4.5% | **0.777**+5.1% | 1.719+12.0% | 2.892+2.2% | 10.63-0.43 |

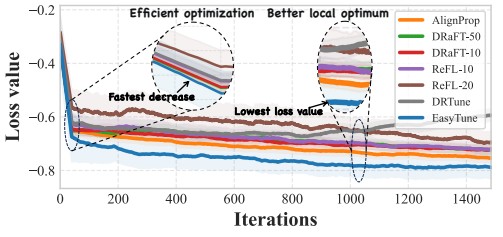

Figure S1: **Loss curves for EasyTune and existing fine-tuning methods.** Here, the x-axis represents the number of generated motion batches.

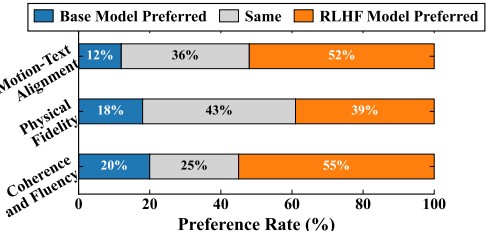

Figure S2: **User study on HumanML3D test set.** We use MDM model as base model.

stronger semantic alignment, while Diversity of MDM slightly rises from 10.76 to 11.27, suggesting maintained motion variety. Similarly, for MoDiffuse (Zhang et al., 2024a), Top-1 R-Precision improves from 0.417 to 0.438 (5.0% gain), Top-2 R-Precision from 0.621 to 0.649 (4.5% gain), and Top-3 R-Precision from 0.739 to 0.777 (5.1% gain). FID of MoDiffuse decreases from 1.954 to 1.719 (12.0% improvement), and MM Dist of MoDiffuse reduces from 2.958 to 2.892 (2.2% improvement). However, Diversity of MoDiffuse slightly declines from 11.10 to 10.63, indicating a minor trade-off in variety for improved alignment and quality. Compared to baselines, EasyTune-enhanced models achieve superior performance. Top-1 R-Precision of MDM with EasyTune (0.442) surpasses that of ParCo (0.430) and MotionMamba (0.419), while FID of MDM with EasyTune (0.284) is competitive with SiT (0.242). MM Dist of MDM with EasyTune (2.755) outperforms most baselines, approaching the real data's 2.788. These results establish EasyTune-enhanced models as new state-of-the-art on KIT-ML, highlighting their ability to improve text-motion alignment, motion quality, and semantic relevance.

## A.4 VISUALIZATIONS

We visualize motions generated by the original MLD (Chen et al., 2023) and by MLD fine-tuned with our EasyTune, as shown in Fig. S3. Our proposed EasyTune substantially improves the capacity of text-to-motion models to comprehend textual semantics. For example, in Fig. S3(j), the model fine-tuned with our proposed EasyTune effectively generates a motion that accurately reflects the semantic intent of the description "The man is marching like a soldier," whereas the original model fails to capture this nuanced behavior.

## A.5 QUANTITATIVE RESULTS ON STEP-LEVEL REWARD REWEIGHTING

Prior work has demonstrated that early denoising steps exert a substantial influence on the final generation quality (Xie & Gong, 2025). We further observe that existing trajectory-level optimization methods frequently under-optimize these early steps, as discussed in Sec. 3. To systematically

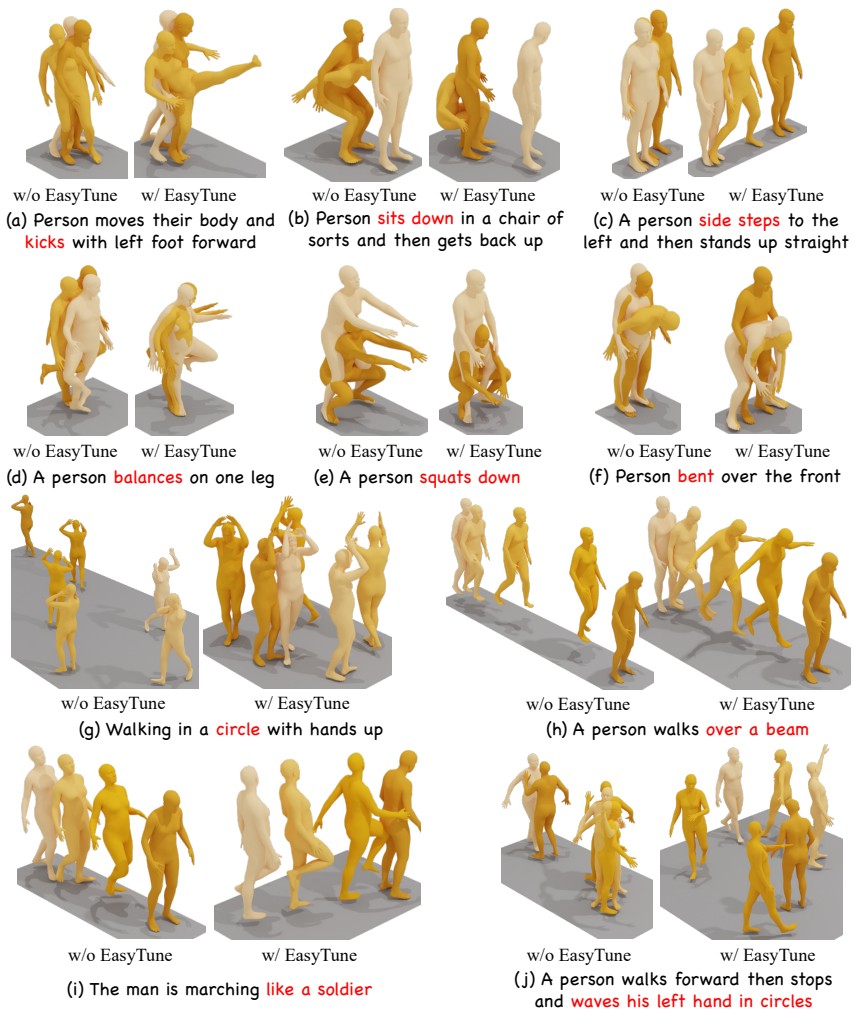

Figure S3: **Visual results on HumanML3D dataset.** "w/o EasyTune" refers to motions generated by the original MLD model (Chen et al., 2023), while "w/ EasyTune" indicates motions generated by the MLD model fine-tuned using our proposed EasyTune.

examine the effect of fine-tuning different subsets of steps, we evaluate four alternative reweighting strategies: (1) optimizing only the final 20 steps; (2) optimizing only the initial 20 steps; (3) linear increasing reweighting with $w_t = \frac{T-t}{T} + 0.5$ at each step; and (4) linear decreasing reweighting with $w_t = -\frac{T-t}{T} + 1.5$ at each step. Results are summarized in Tab. S4.

The experimental results in Tab. S4 highlight the critical role of appropriately weighting different denoising steps. Strategy (1), which optimizes only the final 20 steps, yields clearly inferior performance (R@1: 0.546, FID: 0.184) compared to EasyTune (Full), indicating that restricting

Table S4: **Ablation study on step-level reward reweighting strategies for EasyTune.** The baseline is MLD.

| Method | R Precision ↑ | | | FID ↓ | MM Dist ↓ | Diversity → | Memory (GB) ↓ |
|---|---|---|---|---|---|---|---|
| | Top 1 | Top 2 | Top 3 | | | | |
| Real | 0.511 | 0.703 | 0.797 | 0.002 | 2.974 | 9.503 | - |
| w/ DRaFT-50 (Clark et al., 2024) | 0.528 | 0.724 | 0.819 | 0.197 | 2.872 | 9.641 | 37.32 |
| w/ EasyTune (Full) | 0.581 | 0.769 | 0.855 | 0.132 | 2.637 | 9.465 | 22.10 |
| EasyTune + (1) | 0.546 | 0.735 | 0.804 | 0.184 | 2.815 | 9.682 | 22.10 |
| EasyTune + (2) | 0.567 | 0.759 | 0.842 | 0.158 | 2.673 | 9.430 | 22.10 |
| EasyTune + (3) | 0.556 | 0.748 | 0.838 | 0.147 | 2.652 | 9.421 | 22.10 |
| EasyTune + (4) | 0.584 | 0.773 | 0.859 | 0.136 | 2.631 | 9.521 | 22.10 |

Table S5: **Performance of SPL mechanism for fine-tuning TMR.**

| Methods | Text-Motion Retrieval↑ | | | | | Motion-Text Retrieval↑ | | | | |
|---|---|---|---|---|---|---|---|---|---|---|
| | R@1 | R@2 | R@3 | R@5 | R@10 | R@1 | R@2 | R@3 | R@5 | R@10 |
| TMR | 67.16 | 81.32 | 86.81 | 91.43 | 95.36 | 67.97 | 81.20 | 86.35 | 91.70 | 95.27 |
| +SPL | 68.76 | 82.36 | 87.99 | 92.06 | 96.47 | 69.03 | 82.87 | 87.84 | 92.56 | 96.45 |

optimization to late steps is insufficient to fully exploit the potential of the diffusion process. This result is also comparable to that of DRaFT-50 (R@1: 0.528, FID: 0.197), suggesting that DRaFT-50's suboptimal performance likely stems from gradient vanishing, which effectively causes the method to under-optimize early denoising steps. Strategy (2), which optimizes only the initial 20 steps, achieves somewhat better performance (R@1: 0.567, FID: 0.158) than Strategy (1), but still falls short of EasyTune (Full), showing that early-step optimization alone is also not sufficient.

For the linearly increasing reweighting scheme (Strategy (3), $w_t = \frac{T-t}{T} + 0.5$), which places larger weights on later steps. In contrast, the linearly decreasing reweighting scheme (Strategy (4), $w_t = -\frac{T-t}{T} + 1.5$), which emphasizes earlier steps, achieves the best overall performance among the reweighted variants (R@1: 0.584, FID: 0.136, MM Dist: 2.631). Notably, its performance is on par with, or slightly better than, EasyTune (Full) in terms of alignment (R@1: 0.584 vs. 0.581) while maintaining comparable generation quality. Taken together, these results provide strong empirical evidence that properly optimizing early denoising steps is crucial for downstream performance.

## A.6 SENSITIVITY ANALYSIS

### A.6.1 SENSITIVITY ANALYSIS OF RETRIEVAL MODEL SELECTION

**Effect on Settings.** We investigate the sensitivity of our SPL mechanism to different reward models and their impact on final fine-tuned generation results. A key observation is that weaker reward models, when trained using hyperparameter settings optimized for stronger models, can suffer from training collapse. This occurs because the core principle of SPL is to mine motion pairs and maximize the gap between preferred and non-preferred motions. In other words, the model must learn to produce preferred motions while forgetting non-preferred ones. However, forgetting is inherently simpler than learning, and weaker retrieval models are more prone to mining incorrect pairs during online sampling. To address this, we employ more relaxed candidate number $K$ to increase the probability of successful pair mining, thereby strengthening learning signals and reducing erroneous unlearning.

**Effect on Text-Motion Retrieval Task.** We demonstrate this using TMR, a moderately weaker retrieval model, where SPL consistently improves performance. As shown in Tab. S5, the motion-text retrieval R@1 improves from 67.16% to 68.76%, and motion-text retrieval R@1 improves from 67.97% to 69.03%, confirming that our method generalizes effectively to weaker reward models with appropriate hyperparameter adjustments.

**Effect on Text-to-Motion Generation Task.** We further explore whether step-aware fine-tuning generalizes to weaker pre-trained reward models. As shown in Tab. S6, even when using TMR, a less discriminative retrieval model compared to SPL, our step-aware optimization approach consistently improves generation quality. Specifically, when combined with EasyTune using TMR as the reward model, both the step-level and chain-of-thought variants achieve substantial gains: the step variant reaches R@1 of 0.573 (vs. baseline 0.504), and the chain variant reaches 0.567. They still represent significant improvements over the baseline. This finding suggests that the effectiveness of step-aware optimization is not solely dependent on using the strongest available reward model. Rather, the key insight is that even weaker but still discriminative reward models can provide effective supervision signals, as their ranking capability, though inferior to stronger models, still exceeds that of the base generation model itself (Tan et al., 2025). This robustness to reward model choice broadens the applicability of our approach.

### A.6.2 ANALYSIS OF CANDIDATE NUMBER $K$ SELECTION & RETRIEVAL POOL

**Mechanism.** As discussed above, the core mechanism of our SPL is to mine preference motion pairs online and to maximize the learning signal by enlarging the gap between preferred and non-preferred pairs. In essence, SPL is designed to forget incorrectly generated motions while retaining correct ones. However, this task is inherently asymmetric: forgetting is much easier than remembering, making it crucial to carefully control the frequency at which negative samples are forgotten. In our implementation, both the candidate number $K$ and the retrieval pool configuration substantially affect this behavior. Specifically, when retrieval fails, SPL jointly learns to memorize correct samples and forget incorrect ones; otherwise, the model simply optimizes the original pre-training objective. A larger $K$ and a smaller retrieval pool generally increase the retrieval success rate. **Intuitively, we should therefore choose $K$ and the retrieval pool such that successful retrievals occur substantially more often than failures.**

**Experimental Settings.** Fortunately, Petrovich et al. (2023) has discussed similar retrieval settings. Specifically, four retrieval pool settings are considered: (a) *All*: Using the entire test set without modification, though this can be problematic due to repetitive or subtly different text descriptions (e.g., "person" vs. "human", "walk" vs. "walking"). (b) *All with threshold*: Searching over the entire test set but accepting a retrieval as correct only if the text similarity exceeds a threshold (set to 0.95 on a $[0, 1]$ scale). This approach is more principled, distinguishing between genuine matches and superficially similar pairs like "A human walks forward" vs. "Someone is walking forward". (c) *Dissimilar subset*: Sampling 100 motion-text pairs with maximally dissimilar texts (via quadratic knapsack approximation). This provides a cleaner but easier evaluation setting. (d) *Small batches*: Randomly sampling batches of 32 motion-text pairs and reporting average performance, providing a more realistic in-the-wild scenario.

Among these four configurations, settings (a) and (b) yield low retrieval success rates and are computationally expensive. Their success rates are often even lower than the failure rates, which substantially hinders the practical deployment of SPL. By contrast, configurations (c) and (d) achieve much higher retrieval success rates. In our main experiments, we therefore adopt setting (d). With $K = 10$, configuration (d) attains a retrieval failure ratio of approximately 1:20. This ratio empirically leads to stable optimization. In comparison, configuration (c) exhibits a failure ratio of about 1:6 at $K = 10$, which tends to result in less stable performance due to the higher frequency of retrieval failures.

**Results & Discussion.** Based on configurations (c) and (d), we systematically study how different choices of $K$ and retrieval pool settings influence generation performance. Specifically, we evaluate $K \in \{10, 15, 20\}$ for both settings, and report the results in Tab. S7.

In practice, we recommend conducting a similar sensitivity analysis under limited computational resources (e.g., 10 minutes on a single GPU) to determine appropriate values of $K$ and the retrieval pool for a given application. A retrieval failure ratio of approximately 1:20 typically leads to stable and robust optimization across different scenarios.

### A.6.3 AVAILABILITY OF NOISE-AWARE REWARD

**Experimental Setting.** In this experiment, we assess the noise-aware availability of our reward model, i.e., how well the underlying retrieval models can operate under noisy motion inputs. Specificallly, based on the HumanML3D dataset, we construct noisy test samples $\mathbf{x}_t$ by running the forward diffusion process of MLD. For noise-aware reward models, we directly evaluate on $\mathbf{x}_t$. For output-aware models (TMR (Petrovich et al., 2023)), we instead apply an ODE-based denoising process to $\mathbf{x}_t$ and use the resulting predicted clean samples $\hat{\mathbf{x}}_0$ as inputs.

Table S6: **Fine-tuning performance using TMR as reward model.** The baseline is MLD.

| Method | R Precision ↑ | | | FID ↓ | MM Dist ↓ | Diversity → |
|---|---|---|---|---|---|---|
| | Top 1 | Top 2 | Top 3 | | | |
| Real | 0.511 | 0.703 | 0.797 | 0.002 | 2.974 | 9.503 |
| +TMR (w/ SPL, Step) | 0.573 | 0.760 | 0.843 | 0.173 | 2.682 | 9.942 |
| +TMR (w/ SPL, Chain) | 0.567 | 0.753 | 0.836 | 0.158 | 2.698 | 9.874 |

Table S7: **Sensitivity analysis of the number of candidate motions $K$ and retrieval pool settings.** The baseline is MLD. Numbers in the Method column denote the value of $K$.

| Method | R Precision ↑ | | | FID ↓ | MM Dist ↓ | Diversity → |
|---|---|---|---|---|---|---|
| | Top 1 | Top 2 | Top 3 | | | |
| Real | 0.511 | 0.703 | 0.797 | 0.002 | 2.974 | 9.503 |
| $K = 10 + (d)$ | 0.581 | 0.769 | 0.855 | 0.132 | 2.637 | 9.465 |
| $K = 15 + (d)$ | 0.571 | 0.758 | 0.843 | 0.142 | 2.668 | 9.486 |
| $K = 20 + (d)$ | 0.564 | 0.747 | 0.830 | 0.184 | 2.704 | 9.629 |
| $K = 10 + (c)$ | - | - | - | - | - | - |
| $K = 15 + (c)$ | 0.585 | 0.773 | 0.859 | 0.155 | 2.626 | 9.428 |
| $K = 20 + (c)$ | 0.574 | 0.759 | 0.844 | 0.149 | 2.653 | 9.495 |

Table S8: **Experimental results of Noise-Aware Text-Motion Retrieval.**

| Methods | Input Data | Text-Motion Retrieval↑ | | | | | Motion-Text Retrieval↑ | | | | |
|---|---|---|---|---|---|---|---|---|---|---|---|
| | | R@1 | R@2 | R@3 | R@5 | R@10 | R@1 | R@2 | R@3 | R@5 | R@10 |
| ReAlign | Clean Data | 67.59 | 82.24 | 87.44 | 91.97 | 96.28 | 68.94 | 82.86 | 87.95 | 92.44 | 96.28 |
| ReAlign | Noisy Data | 67.20 | 81.46 | 87.11 | 91.39 | 95.67 | 68.02 | 81.84 | 87.56 | 91.39 | 95.69 |
| ReAlign | Predicted Clean Data | 67.80 | 82.38 | 87.86 | 92.27 | 96.61 | 68.04 | 82.47 | 87.97 | 92.10 | 96.34 |
| SPL | Clean Data | 69.31 | 83.71 | 88.66 | 92.81 | 96.75 | 70.23 | 83.41 | 88.72 | 93.07 | 97.04 |
| SPL | Noisy Data | 69.36 | 83.63 | 88.53 | 92.83 | 96.76 | 70.34 | 83.41 | 88.66 | 93.04 | 96.93 |
| SPL | Predicted Clean Data | 68.39 | 83.31 | 88.59 | 93.11 | 96.73 | 68.60 | 82.35 | 88.06 | 92.79 | 96.53 |
| TMR | Clean Data | 67.16 | 81.32 | 86.81 | 91.43 | 95.36 | 67.97 | 81.20 | 86.35 | 91.70 | 95.27 |
| TMR | Predicted Clean Data | 66.98 | 81.04 | 87.09 | 92.11 | 95.74 | 68.32 | 80.69 | 86.43 | 92.13 | 95.84 |

**Results & Discussion.** As shown in Tab. S8, both noise-aware models (ReAlign and SPL) demonstrate remarkable stability across conditions, with SPL achieving virtually identical performance on noisy data (69.36% R@1) versus clean data (69.31% R@1), while output-aware models like TMR can be effectively adapted via ODE-based denoising (66.98% recovery from clean baseline 67.16%), collectively validating that modern retrieval models reliably perceive intermediate denoising states and can serve as robust reward models for diffusion-based optimization.

### A.6.4 SENSITIVITY ANALYSIS ON LEARNING RATE

To further assess the robustness of our approach, we examine the sensitivity of EasyTune to the learning rate hyperparameter. Across a reasonable range of learning rates (from $10^{-5}$ to $2 \times 10^{-4}$), our method exhibits only minor performance variation, indicating strong robustness to this critical hyperparameter. The corresponding results are shown in Fig. S4.

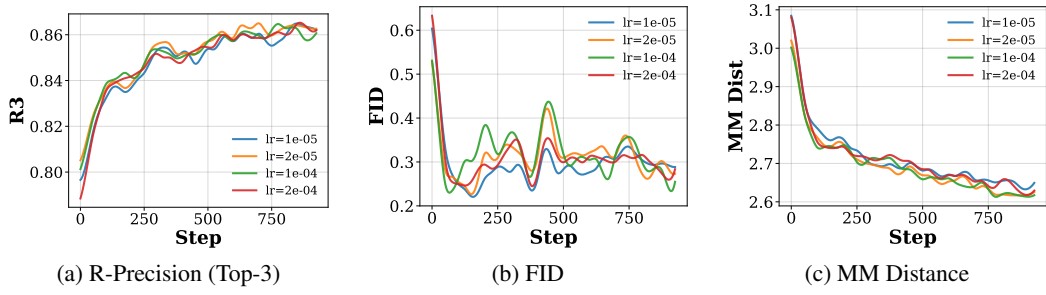

(a) R-Precision (Top-3)  (b) FID  (c) MM Distance

Figure S4: **Learning rate sensitivity analysis on validation set.** Performance metrics remain stable across the learning rate range (spanning from $2 \times 10^{-4}$ to $10^{-5}$), demonstrating the robustness of EasyTune to this hyperparameter. (a) R-Precision at Top-3; (b) Frechet Inception Distance; (c) Multi-Modal Distance.

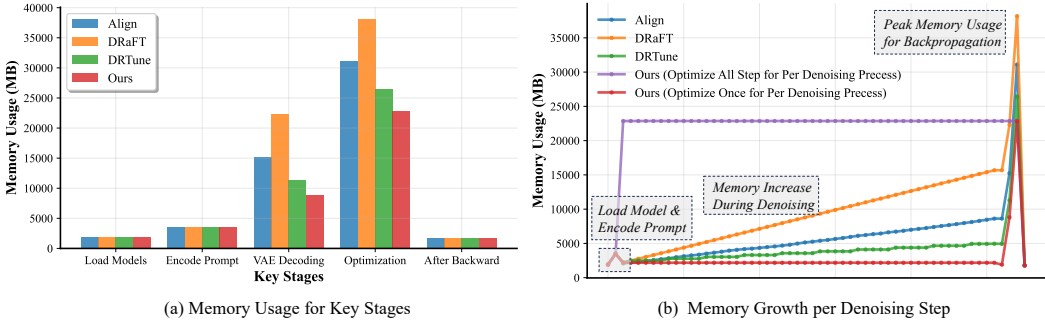

(a) Memory Usage for Key Stages          (b) Memory Growth per Denoising Step

Figure S5: **Comprehensive memory analysis of EasyTune and existing fine-tuning methods.** We report the memory usage of key stages (model loading, prompt encoding, denoising, VAE-based motion decoding, and reward computation with backpropagation), as well as the full memory trajectory during optimization. EasyTune achieves lower peak memory while maintaining high utilization, benefiting from the $\mathcal{O}(1)$ memory growth of the denoising process.

## A.7 COMPREHENSIVE STATISTICS ON OVERHEAD

### A.7.1 MEMORY OVERHEAD

In this part, we provide a detailed comparison of the memory consumption of existing fine-tuning methods and our EasyTune framework. Our pipeline consists of several key stages, including loading the model, encoding text prompts, denoising in the latent space, decoding motions via the VAE, and computing rewards followed by backpropagation. As illustrated in Fig. S5, we report both (a) the memory usage of each individual stage and (b) the overall memory trajectory throughout the full optimization process. All experiments are conducted on a single NVIDIA RTX A6000 GPU, with Intel(R) Xeon(R) Silver 4316 CPU @ 2.30GHz.

It is worth noting that our method performs multiple optimization steps within a single denoising trajectory, which leads to relatively high average memory utilization. However, the peak memory consumption of EasyTune is significantly lower than that of existing methods. In practice, higher utilization indicates more efficient use of available GPU resources, while a lower peak memory footprint reflects reduced hardware requirements. The results further demonstrate that the memory savings of our method mainly stem from the $\mathcal{O}(1)$ memory growth of the denoising process.

### A.7.2 COMPUTATIONAL OVERHEAD

In this part, we benchmark the training-time and computational overhead of EasyTune against existing fine-tuning methods. Following the setup used in our main experiments, we measure the training time and total TFLOPs required to reach convergence. Additionally, all methods share the same sampling procedure as their corresponding base diffusion models, and thus incur no additional overhead during inference.

As shown in Tab. S9, EasyTune is consistently more training-efficient than prior differentiable reward-based methods. It achieves the lowest per-step optimization cost (1.47 seconds per update vs roughly 4.7–5.6 seconds for other methods), and to reach a reward score of 0.70 it needs only 263.36 seconds and 10191 TFLOPs, compared to 466.27 seconds and 18044 TFLOPs for DRaFT. For a reward score of 0.75, EasyTune converges in 358.17 seconds (13861 TFLOPs), while DRaFT requires 2616.54 seconds (101260 TFLOPs), yielding a **7.3x speedup** under a much smaller compute budget. Moreover, EasyTune is the only method that successfully reaches reward scores of 0.80 and 0.85 within the given budget, highlighting its stronger optimization capacity. Together with the memory efficiency analysis in Fig. S5, these results show that EasyTune offers a substantially more efficient and practical fine-tuning strategy than existing differentiable reward-based approaches.

Table S9: **Computational overhead comparison.** We report the training time and TFLOPs required to reach different reward scores. Total time is measured in seconds on a single NVIDIA RTX A6000 GPU. "-" indicates the method could not reach that reward level within a reasonable training budget.

|  | DRaFT | AlignProp | DRTune | ReFL | **EasyTune (Ours)** |
|---|---|---|---|---|---|
| Time per Opt. (s) | 5.61 | 5.17 | 4.90 | 4.72 | **1.47** |
| *Reward Score = 0.70* | | | | | |
| Time (s) | 466.27 | 271.99 | 554.77 | 820.29 | **263.36** |
| TFLOPs | 18044 | 10526 | 21469 | 31745 | **10191** |
| *Reward Score = 0.75* | | | | | |
| Time (s) | 2616.54 | 971.55 | 2009.59 | - | **358.17** |
| TFLOPs | 101260 | 37599 | 77771 | - | **13861** |
| *Reward Score = 0.80* | | | | | |
| Time (s) | - | - | - | - | **452.53** |
| TFLOPs | - | - | - | - | **17513** |
| *Reward Score = 0.85* | | | | | |
| Time (s) | - | - | - | - | **1025.17** |
| TFLOPs | - | - | - | - | **39674** |

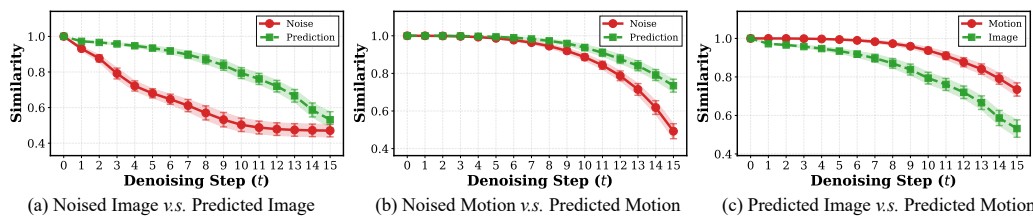

(a) Noised Image *v.s.* Predicted Image  (b) Noised Motion *v.s.* Predicted Motion  (c) Predicted Image *v.s.* Predicted Motion

Figure S6: **Noise-perception comparison between images and motions.** We report the cosine similarity between noisy states and their ODE-based predictions etween ODE-based predictions and trajectory-level rewards, across denoising steps for both the image (FLUX.1 dev) and motion (MLD) domains.

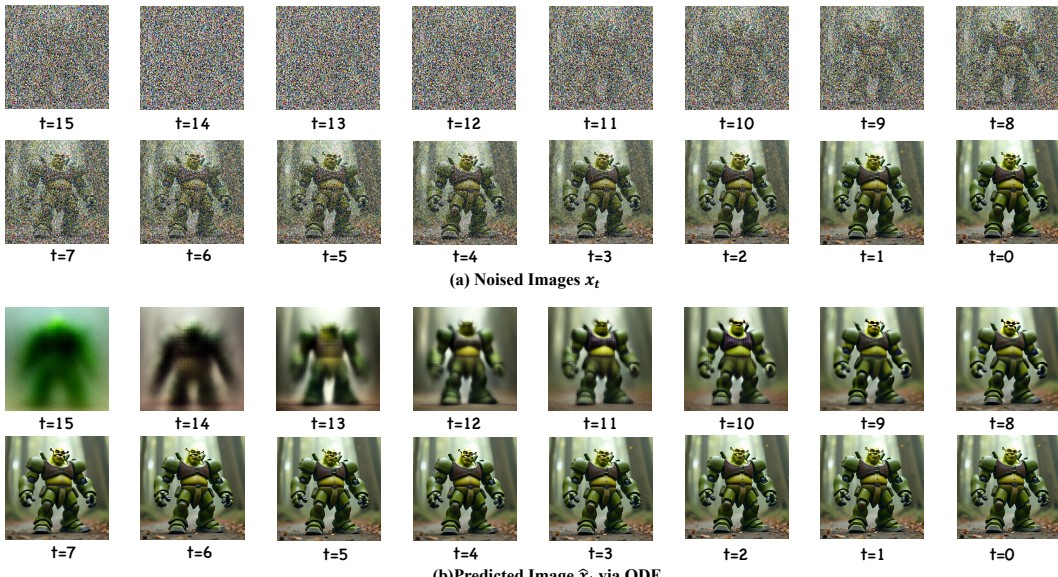

Figure S7: **Noisy states and ODE-based predictions for images across denoising steps.** Visualization of intermediate noisy images and their corresponding ODE-based predictions at different steps of the denoising process using FLUX.1 dev.

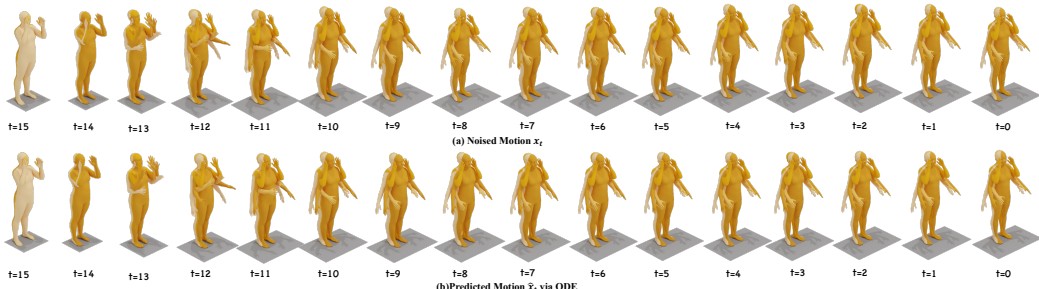

Figure S8: **Noisy states and ODE-based predictions for motions across denoising steps.** Visualization of intermediate noisy motions and their corresponding ODE-based predictions at different steps of the denoising process using MLD.

## A.8 Noise-Perception Analysis of Image and Motion

**Experimental Settings.** To quantify the perceptibility and sensitivity of noisy states, we perform a unified study on both image and motion generation. For the image domain, we select 50 prompts from HPDv2 (Wu et al., 2023) and use FLUX.1 dev (Labs, 2024) as the base model. For each prompt, we generate 12 images with 16 denoising steps, a guidance scale of 3.5, and a resolution of $720 \times 720$. At each of the 16 steps, we store both the noisy image and its ODE-based prediction, and compute their cosine similarity using CLIP ViT-L/14 features (Radford et al., 2021). To ensure statistical significance and robustness, all experiments are repeated 50 times per prompt over 50 prompts, and we report the mean.

For the motion domain, we follow a similar protocol using MLD as the base model and TMR as the feature extractor. We randomly sample 50 text prompts from HumanML3D and, for each prompt, generate 12 motion samples along the full denoising trajectory. At each step, we compute the cosine similarity between the noisy motion and its ODE-based prediction in the TMR feature space.

**Results and Discussion.** As summarized in Fig. S6, both image and motion models exhibit increasing similarity between noisy states and ODE-based predictions as the denoising process proceeds. However, the image final results consistently shows weaker similarity with noised state, especially at high-noise steps. This indicates that image generation results have a poorer perception of early noise states compared to motion generation, making it harder for directly fine-tuning models by step-level optimization. Fortunately, their ODE-based prediction results show comparable performance, indicating that their step-aware optimization can still be performed by ODE-based prediction. Additionally, visual examples of noisy states and their ODE-based predictions are presented in Fig. S7 and Fig. S8, showing the perceptual differences across denoising steps for both domains.

## A.9 Evaluation on Physical Perception Ability of Reward Model.

To investigate the physical perception capabilities of our reward model, we conducted an experiment to assess its ability to distinguish between real and generated motions. We selected 50 prompts and their corresponding ground-truth motions ($\mathbf{x}_{gt}$) from the HumanML3D test set. For each prompt, we also generated a motion ($\mathbf{x}_{gen}$) using the pretrained MLD model. We then evaluated both the ground-truth and generated motions using our reward model with an empty text condition ($c = $ ''), obtaining reward scores $r(\mathbf{x}_{gt}, c)$ and $r(\mathbf{x}_{gen}, c)$.

Table S10: **Physical perception evaluation of the reward model.**

| Comparison Result | Count |
|---|---|
| $r(\mathbf{x}_{gt}, c) > r(\mathbf{x}_{gen}, c)$ | 48 |
| $r(\mathbf{x}_{gt}, c) \leq r(\mathbf{x}_{gen}, c)$ | 2 |
| Total | 50 |

Our findings are summarized in Tab. S10. The reward for the ground-truth motion was higher than for the generated motion in 48 out of 50 cases (96%). **This result strongly suggests that the reward model possesses a significant degree of physical perception.** This capability likely arises because the reward model is trained on real-world motion data, treating it as in-distribution, while viewing synthetically generated data as out-of-distribution. Such a distinction enables the model to

*Prompt: A person stands up from the ground, lifts their right foot, and sets it back down.*

*Prompt: A person squats down, then stands up and moves forward.*

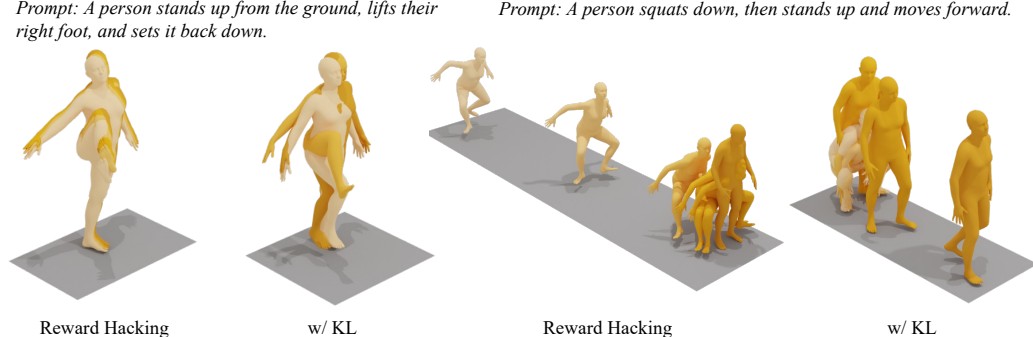

Reward Hacking      w/ KL      Reward Hacking      w/ KL

Figure S9: **Illustration of reward hacking in motion generation.** Examples demonstrating that over-fitting to reward signals may lead to semantically aligned but physically unrealistic motions. For better visualization, corresponding videos are provided in the supplementary materials.

develop a sensitivity to physical plausibility, a phenomenon similarly observed in anomaly detection literature (Flaborea et al., 2023).

### A.10 FAILURE CASE ANALYSIS: REWARD HACKING.

As a well-known challenge in reinforcement learning (Clark et al., 2024), reward hacking can emerge during model fine-tuning, where continued optimization after convergence may degrade generation quality.

As illustrated in Fig. S9, models may over-fit to semantic alignment while neglecting realistic motion dynamics. For example, given the prompt "A person stands up from the ground, lifts their right foot, and sets it back down," a model suffering from reward hacking might generate a person *continuously* lifting their foot to over-fit to the "lifts their right foot" action. Similarly, for "A person squats down, *then stands up and moves forward*," the model might misinterpret this as "A person squats down *while moving forward*." Fortunately, our method combined with KL-divergence regularization (as discussed in Sec. A.2) can effectively mitigate this phenomenon. We recommend early stopping after initial convergence, with the stopping point determined by validation set performance, which provides an effective strategy to alleviate reward hacking issues.

## B MORE TECHNICAL DISCUSSIONS

### B.1 GRADIENT ANALYSIS OF DIFFERENTIABLE REWARD METHODS

Given a diffusion model $\epsilon_\theta$, we first sample a latent variable $\mathbf{x}_T \sim \mathcal{N}(0, \mathbf{I})$. Starting from $\mathbf{x}_T$, we apply $T-t$ denoising steps to obtain $\mathbf{x}_t$, while retaining the computational graph for gradient analysis. At step $t$, we consider the Jacobian of the diffusion prediction with respect to its input,

$$\frac{\partial \pi_\theta(\mathbf{x}_t^\theta)}{\partial \mathbf{x}_t^\theta}, \tag{S2}$$

and its sequential product across denoising steps,

$$\prod_{s=1}^{t} \frac{\partial \pi_\theta(\mathbf{x}_s^\theta)}{\partial \mathbf{x}_s^\theta}. \tag{S3}$$

This quantity can be interpreted as the gradient of the noisy sample $\mathbf{x}_t^\theta$ with respect to the predicted clean sample $\mathbf{x}_0^\theta$, i.e., the effective coefficient governing optimization at step $t$. Consequently, the gradient of the training objective takes the form

$$\frac{\partial \mathcal{L}(\theta)}{\partial \theta} = -\mathbb{E}_{c \sim \mathcal{D}_{\mathrm{T}}, \mathbf{x}_0^\theta \sim \pi_\theta(\cdot|c)} \left[ \frac{\partial \mathcal{R}_\phi(\mathbf{x}_0^\theta)}{\partial \mathbf{x}_0^\theta} \cdot \sum_{t=1}^{T} \underbrace{\left( \prod_{s=1}^{t-1} \frac{\partial \pi_\theta(\mathbf{x}_s^\theta, s, c)}{\partial \mathbf{x}_s^\theta} \right)}_{\text{tends to 0 as t increases}} \underbrace{\left( \frac{\partial \pi_\theta(\mathbf{x}_t^\theta, t, c)}{\partial \theta} \right)}_{\text{optimizing t-th step}} \right]. \tag{S4}$$

Importantly, during optimization, especially when $t$ is large, these coefficients rapidly decay toward 0 (as shown in Fig. 3), i.e.,

$$\prod_{s=1}^{t} \frac{\partial \pi_\theta(x_s^\theta)}{\partial x_s^\theta} \to 0. \tag{S5}$$

Thus, for large noise steps, which often determine the final generation quality (Xie & Gong, 2025), this vanishing effect implies that existing methods tend to under-optimize such steps, thereby leading to coarse or suboptimal results.

## B.2 DISCUSSION ON STEP OPTIMIZATION AND POLICY GRADIENT METHODS

In the field of diffusion posting-training, beyond mentions differentiable-reward approaches, another major line of work is policy-gradient methods, such as GRPO (Xue et al., 2025; Liu et al., 2025), PPO (Ren et al., 2024), and DDPO (Black et al., 2023). In this section, we point out that, **both our approach and these policy-gradient methods truncate the gradient along the diffusion chain, and in practice, they have not been observed to suffer from global inconsistency issues.** As a result, our method inherits a widely accepted and principled design, and does not suffer from undesirable behaviors caused by breaking the differentiable chain structure. Here, we briefly discuss the connections between our method and existing policy-gradient methods.

**GRPO.** DanceGRPO (Xue et al., 2025) can be viewed as instantiating GRPO (Guo et al., 2025) in the diffusion setting, where each denoising trajectory sampled under a condition $c$ is treated as a rollout generated by the policy $p_\theta$. For every prompt $c$, a group $\mathcal{G} = \{\mathbf{x}^{1:K}\}$ of $K$ trajectories is drawn from the current policy, and a group-wise relative advantage is computed to compare their rewards. By estimating advantages $\mathcal{A}^k$ from this group of $K$ rollouts, GRPO (Xue et al., 2025) optimizes a PPO-style clipped surrogate objective:

$$\mathcal{J}_{\text{GRPO}}(\theta, \mathcal{G}) = \mathbb{E}_{c\sim\mathcal{D},\, \mathbf{x}^{1:K},\, t\sim\mathcal{U}(0,T),\, k\sim\mathcal{U}(1,K)}\left[\min\big(\rho_\theta^{k,t}\,\mathcal{A}^k,\ \text{clip}(\rho_\theta^{k,t},\, 1-\varepsilon,\, 1+\varepsilon)\,\mathcal{A}^k\big)\right], \tag{S6}$$

where advantages $\mathcal{A}^k$ and probability ratios $\rho_\theta^{k,t}$ are computed as follows:

$$\mathcal{A}^k = \frac{r(\mathbf{x}_0^k, c) - \mu(r(\mathbf{x}^{1:K}, c))}{\sigma(r(\mathbf{x}^{1:K}, c))},\ \rho_\theta^{k,t} = \frac{p_\theta(\mathbf{x}_t^k \mid \mathbf{x}_{t-1}^k, c)}{p_{\theta_{\text{old}}}(\mathbf{x}_t^k \mid \mathbf{x}_{t-1}^k, c)}, \tag{S7}$$

where $r(\mathbf{x}_0^k, c)$ is the scalar reward assigned to the final sample in the $k$-th rollout, and $\mu(r(\mathbf{x}^{1:K}, c))$ and $\sigma(r(\mathbf{x}^{1:K}, c))$ denote the mean and standard deviation of rewards within the group $\mathbf{x}^{1:K}$. This group-wise normalization makes $\mathcal{A}^k$ a relative score that measures how much better (or worse) a sample is compared to its peers under the same condition $c$, which stabilizes training and mitigates scale mismatch across prompts.

**DDPO.** As a classical RL method, DDPO (Black et al., 2023; Fan et al., 2023b) optimizes the generative diffusion model via policy gradients using $K$ rollouts. Its objective is written as:

$$\nabla_\theta \mathcal{J}_{\text{DDPO}}(\theta, \mathcal{G}) = \mathbb{E}_{c\sim\mathcal{D},\, \mathbf{x}^{1:K},\, t\sim\mathcal{U}(0,T),\, k\sim\mathcal{U}(1,K)}\left[r(\mathbf{x}_0^k, c) \cdot \nabla_\theta \log p_\theta(\mathbf{x}_t^k \mid \mathbf{x}_{t-1}^k, c)\right]. \tag{S8}$$

Here $r(\mathbf{x}_0^k, c)$ again denotes the scalar reward assigned to the final denoised sample in the $k$-th rollout, while $\log p_\theta(\mathbf{x}_t^k \mid \mathbf{x}_{t-1}^k, c)$ is the log-probability at step $t$.

**Discussion.** By examining Eq. (S7) and Eq. (S8), we identify a key observation: both GRPO and DDPO treat this optimization as multiple pre-step optimization (in the form of $\log p_\theta(\mathbf{x}_t^k \mid \mathbf{x}_{t-1}^k, c)$), rather than optimizing the entire denoising process in a single optimization pass. Specifically, GRPO and DDPO first sample $K$ trajectories of length $T$ under each condition. Subsequently, they assign a scalar reward to each trajectory and use this reward as a weight to progressively imitate these trajectories step by step.

However, we observe a fundamental inconsistency in existing differentiable-reward methods: **differentiable-reward methods recursively decompose the optimization process as the optimization of multiple sub-chains, rather than as the optimization of independent steps.** This design choice introduces the challenges discussed in the Sec. 3, particularly the vanishing gradient and memory problem illustrated in Fig. 3 and 6. In contrast, our EasyTune solve this issue by step-aware optimization, and its objective is:

$$\mathcal{L}_{\text{EasyTune}}(\theta) = -\mathbb{E}_{c\sim\mathcal{D}_{\text{T}},\, \mathbf{x}_t^\theta\sim\pi_\theta(\cdot|c),\, t\sim\mathcal{U}(1,T)}\left[\mathcal{R}_\phi(\mathbf{x}_t^\theta, t, c)\right], \tag{S9}$$

---

**Algorithm 1** EasyTune: Efficient Step-Aware Fine-Tuning

---

**Input:** Pre-trained diffusion model $\epsilon_\theta$, reward model $R_\phi$.
**Output:** Fine-tuned diffusion model $\epsilon_\theta$.
 1: **for** each text condition $c \in \mathcal{D}_{\mathrm{T}}$ **and** not converged **do**
 2: $\quad$ $\mathbf{x}_T \sim \mathcal{N}(0, \mathbf{I})$
 3: $\quad$ **if** *Chain Optimization* **then**
 4: $\quad\quad$ Copy $\theta' \leftarrow \theta$
 5: $\quad$ **end if**
 6: $\quad$ **for** $t = T, ..., 1$ **do**
 7: $\quad\quad$ Denoise by $\theta$: $\mathbf{x}_{t-1}^\theta = \pi_\theta(\mathbf{x}_t^\theta)$ by $\mathbf{x}_{t-1}^\theta = \pi_\theta(\mathbf{x}_t^\theta, t, c)$, Eq. (7)
 8: $\quad\quad$ **if** *Chain Optimization* **then**
 9: $\quad\quad\quad$ Optimize: update diffusion model $\epsilon_{\theta'}$ by gradient from $\epsilon_\theta$: $\frac{\partial \mathcal{L}_{\mathrm{EasyTune}}(\theta)}{\partial \theta}$ in Eq. (6)
10: $\quad\quad$ **else**
11: $\quad\quad\quad$ Optimize: update diffusion model $\epsilon_\theta$ by $\frac{\partial \mathcal{L}_{\mathrm{EasyTune}}(\theta)}{\partial \theta}$ in Eq. (6)
12: $\quad\quad$ **end if**
13: $\quad\quad$ Stop Gradient: $\mathbf{x}_{t-1}^\theta = \mathrm{sg}(\mathbf{x}_{t-1}^\theta)$
14: $\quad$ **end for**
15: $\quad$ **if** *Chain Optimization* **then**
16: $\quad\quad$ Assign $\theta \leftarrow \theta'$
17: $\quad$ **end if**
18: **end for**

---

where $\mathcal{R}_\phi(\mathbf{x}_t^\theta, t, c)$ denotes the reward directly assigned to the intermediate denoised sample at step $t$, rather than a single trajectory-level scalar. Crucially, our approach evaluates and optimizes at each timestep independently, with $t$ uniformly sampled from $\{0, \ldots, T\}$. Therefore, compared to the chain optimization as shown in Eq. (S4), our method is more aligned with the principled design of policy-gradient methods, while maintaining a step-wise optimization structure that avoids long-chain backpropagation and achieves better results.

## B.3 DISCUSSIONS ON EXISTING FINE-TUNING METHODS

We compare **EasyTune** with existing direct reward fine-tuning methods, including DRaFT-K (Clark et al., 2024), AlignProp (Prabhudesai et al., 2023), ReFL (Clark et al., 2024), and DRTune (Wu et al., 2025a). The pseudocode of EasyTune is provided in Algorithm 1. We highlight four key advantages:

**(1) Higher Optimization Efficiency**: Existing methods update model parameters only after completing $T$ or $T - t_{\mathrm{stop}}$ reverse steps, yielding one update per trajectory. EasyTune optimizes at every denoising step, significantly increasing update frequency and enabling faster convergence.

**(2) Lower Storage Requirements**: DRaFT-K, AlignProp, ReFL, and DRTune rely on recursive gradient computations, requiring storage of intermediate states across multiple timesteps (Eq. (4) and Eq. (10)). EasyTune computes gradients solely for the current timestep (Eq. (7)), eliminating recursive state storage and substantially reducing memory usage.

**(3) Fine-grained Optimization**: Existing methods optimize over coarse timestep ranges or rely on early stopping, limiting their ability to capture step-specific dynamics. EasyTune performs optimization at each denoising step (Algorithm 1), enabling precise per-step adjustments to better align with motion generation objectives.

**(4) Simpler Pipeline**: Existing methods introduce complex designs such as variable timestep sampling or early stopping to mitigate optimization and storage challenges. EasyTune simplifies the process via step-wise optimization (Algorithm 1), making it more straightforward and broadly applicable.

## B.4 DETAILS ON SELF-REFINING PREFERENCE LEARNING

The **Self-Refining Preference Learning (SPL)** mechanism constructs preference pairs for reward model fine-tuning without human annotations, using a retrieval-based auxiliary task. Algorithm 2

---

**Algorithm 2** Self-Refining Preference Learning

---

**Input:** Training subset $\mathcal{D}_\text{T}$, text/motion encoders $\mathcal{E}_\text{M}/\mathcal{E}_\text{T}$, temperature parameter $\tau$, retrieval number $k$.

**Output:** Fine-tuned reward model $\mathcal{E}_\text{M}$, $\mathcal{E}_\text{T}$, and $\tau$.

 1: **Initialize:** Parameters $\phi \leftarrow \{\mathcal{E}_\text{M}, \mathcal{E}_\text{T}, \tau\}$
 2: **for** each data pair $(\mathbf{x}^\text{gt}, c) \in \mathcal{D}_\text{T}$ and not converged **do**
 3:  ▶ **Step 1: Preference Data Mining**
 4:  Compute reward scores for all $\mathbf{x} \in \mathcal{D}_\text{T}$ using Eq. (11)
 5:  Retrieve top-$k$ motions $\mathcal{D}_\text{R}$ using Eq. (13)
 6:  Set winning $\mathbf{x}^\text{w}$ and losing motions $\mathbf{x}^\text{l}$ using Eq. (14)
 7:  ▶ **Step 2: Preference Fine-tuning**
 8:  Compute softmax probabilities $\mathcal{P}$ using Eq. (15)
 9:  Define target distribution $\mathcal{Q}$ using Eq. (16)
10:  Compute loss $\mathcal{L}_\text{SPL}(\phi)$ by $\mathcal{Q}$ and $\mathcal{P}$ using Eq. (17)
11:  Update parameters $\phi$ by $\nabla_\phi \mathcal{L}_\text{SPL}(\phi)$
12: **end for**

---

outlines the process, which iterates over a training subset $\mathcal{D}_\text{T}$ of motion-text pairs to refine text and motion encoders $\mathcal{E}_\text{T}$, $\mathcal{E}_\text{M}$, and a temperature parameter $\tau$, collectively parameterized as $\phi$.

Algorithm 2 formalizes the process of SPL, which operates on a training subset $\mathcal{D}_\text{T}$ containing motion-text pairs $(\mathbf{x}^\text{gt}, c)$, utilizing pre-trained text and motion encoders $\mathcal{E}_\text{T}$, $\mathcal{E}_\text{M}$, and a temperature parameter $\tau$, collectively parameterized as $\phi$. Overall, at each optimization iteration, SPL attempts to mine a preference pair consisting of a winning motion $\mathbf{x}^\text{w}$ and a losing motion $\mathbf{x}^\text{l}$. If such a pair is found (i.e., when retrieval fails), the model is optimized based on this pair; otherwise (i.e., when retrieval succeeds), it falls back to the pretraining objective to reinforce the correct knowledge. The algorithm executes two core steps: Preference Data Identification and Preference Fine-tuning. In the first step (Lines 3–6), for each text condition $c$, reward scores are computed for all motions in $\mathcal{D}_\text{T}$ based on the similarity between motion and text features scaled by $\tau$. The top-$k$ motions are retrieved, and the ground-truth motion $\mathbf{x}^\text{gt}$ is designated as the preferred motion $\mathbf{x}^\text{w}$. If $\mathbf{x}^\text{gt}$ is not among the retrieved motions, the highest-scoring retrieved motion is set as the non-preferred motion $\mathbf{x}^\text{l}$; otherwise, $\mathbf{x}^\text{l}$ is set to $\mathbf{x}^\text{gt}$, and optimization is skipped to avoid trivial updates. This retrieval-based approach effectively mines preference pairs by identifying motions that are incorrectly favored by the current model, thus providing a robust signal for refinement. In the second step (Lines 7–10), the reward scores of the preference pair $(\mathbf{x}^\text{w}, \mathbf{x}^\text{l})$ are converted into softmax probabilities $\mathcal{P}$, representing the model's predicted preference distribution. These are aligned with a target distribution $\mathcal{Q}$, which assigns a probability of 1.0 to $\mathbf{x}^\text{w}$ and 0.0 to $\mathbf{x}^\text{l}$ when a preference exists, or 0.5 to both when they are identical. The model is optimized by minimizing the KL divergence between $\mathcal{Q}$ and $\mathcal{P}$, with the resulting loss used to update $\phi$ via gradient descent. This fine-tuning process iteratively refines the encoders to assign higher scores to preferred motions, enhancing the reward model's ability to capture fine-grained preferences. The iteration continues until convergence, yielding a reward model tailored for motion generation tasks.

## B.5 Discussion on Noise-Aware and One-Step Reward

In Sec. 4.1, we introduced both the Noise-Aware reward for ODE and SDE sampling and the One-Step reward specifically for ODE sampling. Here, we provide recommendations for selecting between these strategies and briefly compare their performance.

**Perceptual Difference Between Noisy and Predicted Data.** As analyzed in App. A.8, the predictability of noisy data in the motion domain is relatively strong compared to the image domain (see Fig. S6). Fig. S8 and Fig. S7 demonstrates that ODE-based strategy further enhances this predictability. *Consequently, both reward strategies can effectively perceive noisy data in motion generation.* For image generation, where noisy data is harder to interpret, we recommend the One-Step reward strategy for more accurate perception.

**Quantitative Analysis of Retrieval Results on Noisy Data.** In App. A.6.3, we quantitatively analyze the performance difference between the two strategies on retrieval tasks using noisy data.

The results in Tab. S8 demonstrate that *both strategies achieve robust performance on noisy data retrieval, comparable to results on clean data.*

**Quantitative Comparison of Generation Results.** For ODE-based models, both strategies are applicable. In Tab. 2, we provided performance metrics for MLD and MLD++ under both strategies. We revisit and consolidate those results in Tab. S11. The results indicate that the Noise-Aware reward generally yields better performance. *Therefore, we recommend using the Noise-Aware strategy if the reward model possesses noise-perception capabilities. Otherwise, the One-Step reward can achieve comparable results.*

Table S11: **Comparison of Noise-Aware and One-Step Rewards on ODE-based models.**

| Model | Strategy | R-Precision ↑ | | | FID ↓ | MM-Dist ↓ |
| | | Top 1 | Top 2 | Top 3 | | |
|---|---|---|---|---|---|---|
| MLD | One-Step | 0.568 | 0.754 | 0.846 | 0.194 | 2.672 |
| | Noise-Aware | **0.581** | **0.769** | **0.855** | **0.132** | **2.637** |
| MLD++ | One-Step | 0.581 | 0.762 | 0.849 | 0.073 | 2.603 |
| | Noise-Aware | **0.591** | **0.777** | **0.859** | **0.069** | **2.592** |

## C   PROOF

### C.1   PROOF OF COROLLARY 1

Recall the Corollary 1.

**Corollary.** *Given the reverse process in Eq.* (2), $\mathbf{x}_{t-1}^{\theta} = \pi_{\theta}(\mathbf{x}_t^{\theta}, t, c)$, *the gradient w.r.t diffusion model $\theta$, denoted as $\frac{\partial \mathbf{x}_{t-1}^{\theta}}{\partial \theta}$, can be expressed as:*

$$\frac{\partial \mathbf{x}_{t-1}^{\theta}}{\partial \theta} = \frac{\partial \pi_{\theta}(\mathbf{x}_t^{\theta}, t, c)}{\partial \theta} + \frac{\partial \pi_{\theta}(\mathbf{x}_t^{\theta}, t, c)}{\partial \mathbf{x}_t^{\theta}} \cdot \frac{\partial \mathbf{x}_t^{\theta}}{\partial \theta}. \tag{S10}$$

*Proof.* Let $u(\theta) = \mathbf{x}_t^{\theta}$, $v(\theta) = \theta$, and define $\tilde{F}(\theta) = F\big(u(\theta), v(\theta)\big) = \pi_{v(\theta)}\big(u(\theta), t, c\big)$. By the multivariate chain rule, the derivative of $\tilde{F}$ w.r.t. $\theta$ is:

$$\frac{\partial \tilde{F}(\theta)}{\partial \theta} = \frac{\partial F}{\partial v} \cdot \frac{\partial v}{\partial \theta} + \frac{\partial F}{\partial u} \cdot \frac{\partial u}{\partial \theta}. \tag{S11}$$

The first term $\frac{\partial v}{\partial \theta}$ can be expressed as:

$$\frac{\partial v}{\partial \theta} = \frac{\partial \theta}{\partial \theta} = I, \tag{S12}$$

and the second term $\frac{\partial u}{\partial \theta}$ can be expressed as:

$$\frac{\partial u}{\partial \theta} = \frac{\partial \mathbf{x}_t^{\theta}}{\partial \theta}. \tag{S13}$$

Hence, we can rewrite the equation as:

$$\frac{\partial \tilde{F}(\theta)}{\partial \theta} = \frac{\partial F}{\partial v} + \frac{\partial F}{\partial u} \cdot \frac{\partial \mathbf{x}_t^{\theta}}{\partial \theta}. \tag{S14}$$

Furthermore, we substitute $F(u, v)$ with $\pi_{\theta}(\mathbf{x}_t^{\theta}, t, c)$, and thus the relationship described in Eq. (S10) holds:

$$\frac{\partial \mathbf{x}_{t-1}^{\theta}}{\partial \theta} = \frac{\partial \pi_{\theta}(\mathbf{x}_t^{\theta}, t, c)}{\partial \theta} = \frac{\partial \pi_{\theta}(\mathbf{x}_t^{\theta}, t, c)}{\partial \theta} + \frac{\partial \pi_{\theta}(\mathbf{x}_t^{\theta}, t, c)}{\partial \mathbf{x}_t^{\theta}} \cdot \frac{\partial \mathbf{x}_t^{\theta}}{\partial \theta}. \tag{S15}$$

The proof is completed. ☐

## C.2 Convergence Analysis

We now provide a convergence guarantee for EasyTune. For clarity, we write its update rule in the generic stochastic-gradient form

$$\theta_{k+1} = \theta_k - \eta_k g_k, \tag{S16}$$

where $g_k$ is the stochastic gradient computed from a minibatch of noisy motions at a randomly sampled denoising step, following the EasyTune objective in Eq. (6). Let $\mathcal{L}(\theta)$ denote the corresponding expected training objective.

We make the following assumptions on the EasyTune update:

(A1) (Lower bounded and smooth objective) $\mathcal{L}(\theta)$ is lower bounded by some $\mathcal{L}_{\inf} > -\infty$ and has $L$-Lipschitz continuous gradient (i.e., $L$-smooth), meaning $\|\nabla\mathcal{L}(\theta) - \nabla\mathcal{L}(\theta')\| \le L\|\theta - \theta'\|$ for all $\theta, \theta'$.

(A2) (Bounded second moment of stochastic gradient) The stochastic gradient $g_k$ satisfies $\mathbb{E}[\|g_k\|^2 \mid \theta_k] \le G^2$ for some constant $G > 0$.

(A3) (Controlled bias from stop-gradient) The bias induced by the stop-gradient operation is uniformly bounded and proportional to the step size, i.e., $\|\mathbb{E}[g_k \mid \theta_k] - \nabla\mathcal{L}(\theta_k)\| \le b\,\eta_k$ for some constant $b \ge 0$.

These assumptions are standard in the analysis of non-convex stochastic gradient methods and, in our diffusion-based motion tuning setting, (A2) captures the bounded variance of the minibatch gradient obtained by sampling noisy motions and timesteps, while (A3) models the $\mathcal{O}(\eta_k)$ bias introduced by the stop-gradient design.

**Theorem S1** (Convergence of EasyTune). *Under the above conditions, the sequence $\{\theta_k\}$ generated by EasyTune satisfies the following properties:*

$$\mathbb{E}[\mathcal{L}(\theta_{k+1})] \le \mathbb{E}[\mathcal{L}(\theta_k)] - c_1\,\eta_k\,\mathbb{E}\big[\|\nabla\mathcal{L}(\theta_k)\|^2\big] + c_2\,\eta_k^2, \tag{S17}$$

*Proof.* The proof follows the standard template for non-convex stochastic gradient descent, adapted to the EasyTune update.

By $L$-smoothness of $\mathcal{L}$ *(Assumption (A1))*, for any $k$ we have

$$\mathcal{L}(\theta_{k+1}) \le \mathcal{L}(\theta_k) + \nabla\mathcal{L}(\theta_k)^T(\theta_{k+1} - \theta_k) + \tfrac{L}{2}\|\theta_{k+1} - \theta_k\|^2. \tag{S18}$$

Substituting $\theta_{k+1} - \theta_k = -\eta_k g_k$ gives

$$\mathcal{L}(\theta_{k+1}) \le \mathcal{L}(\theta_k) - \eta_k\,\nabla\mathcal{L}(\theta_k)^T g_k + \tfrac{L}{2}\eta_k^2\|g_k\|^2. \tag{S19}$$

Taking conditional expectation given $\theta_k$ and using the tower property of expectation, we obtain

$$\mathbb{E}[\mathcal{L}(\theta_{k+1}) \mid \theta_k] \le \mathcal{L}(\theta_k) - \eta_k\,\nabla\mathcal{L}(\theta_k)^T\,\mathbb{E}[g_k \mid \theta_k] + \tfrac{L}{2}\eta_k^2\,\mathbb{E}[\|g_k\|^2 \mid \theta_k]. \tag{S20}$$

By *Assumption (A2)*, $\mathbb{E}[\|g_k\|^2 \mid \theta_k] \le G^2$, so

$$\mathbb{E}[\mathcal{L}(\theta_{k+1}) \mid \theta_k] \le \mathcal{L}(\theta_k) - \eta_k\,\nabla\mathcal{L}(\theta_k)^T\,\mathbb{E}[g_k \mid \theta_k] + \tfrac{L}{2}\eta_k^2\,G^2. \tag{S21}$$

Next we control the inner product term using *Assumption (A3)*. Let $m_k := \mathbb{E}[g_k \mid \theta_k]$ and write

$$\nabla\mathcal{L}(\theta_k)^T m_k = \nabla\mathcal{L}(\theta_k)^T\nabla\mathcal{L}(\theta_k) + \nabla\mathcal{L}(\theta_k)^T(m_k - \nabla\mathcal{L}(\theta_k)) \tag{S22}$$

$$\ge \|\nabla\mathcal{L}(\theta_k)\|^2 - \|\nabla\mathcal{L}(\theta_k)\|\,\|m_k - \nabla\mathcal{L}(\theta_k)\| \tag{S23}$$

$$\ge \|\nabla\mathcal{L}(\theta_k)\|^2 - b\,\eta_k\,\|\nabla\mathcal{L}(\theta_k)\|, \tag{S24}$$

where we used Cauchy–Schwarz and (A3) in the last inequality. Hence

$$-\eta_k\,\nabla\mathcal{L}(\theta_k)^T m_k \le -\eta_k\,\|\nabla\mathcal{L}(\theta_k)\|^2 + b\,\eta_k^2\,\|\nabla\mathcal{L}(\theta_k)\|. \tag{S25}$$

Applying Young's inequality $2ab \le a^2 + b^2$ with $a = \sqrt{\eta_k}\,\|\nabla\mathcal{L}(\theta_k)\|$ and $b = b\,\eta_k^{3/2}$, we get

$$b\,\eta_k^2\,\|\nabla\mathcal{L}(\theta_k)\| \le \tfrac{1}{2}\,\eta_k\,\|\nabla\mathcal{L}(\theta_k)\|^2 + \tfrac{1}{2}b^2\,\eta_k^3. \tag{S26}$$

Therefore

$$-\eta_k \nabla \mathcal{L}(\theta_k)^T m_k \le -\tfrac{1}{2}\eta_k \|\nabla \mathcal{L}(\theta_k)\|^2 + \tfrac{1}{2}b^2 \eta_k^3. \tag{S27}$$

Combining the above bounds yields

$$\mathbb{E}[\mathcal{L}(\theta_{k+1}) \mid \theta_k] \le \mathcal{L}(\theta_k) - c_1 \eta_k \|\nabla \mathcal{L}(\theta_k)\|^2 + C_1 \eta_k^2, \tag{S28}$$

for some positive constants $c_1, C_1$ depending only on $L$, $G$, and $b$ (we absorb the $\mathcal{O}(\eta_k^3)$ term into the $\mathcal{O}(\eta_k^2)$ term). Taking full expectation over $\theta_k$ then gives

$$\mathbb{E}[\mathcal{L}(\theta_{k+1})] \le \mathbb{E}[\mathcal{L}(\theta_k)] - c_1 \eta_k \mathbb{E}\big[\|\nabla \mathcal{L}(\theta_k)\|^2\big] + c_2 \eta_k^2, \tag{S29}$$

where we set $c_2 := C_1$. This proves the claimed one-step descent inequality. $\square$

Building on this classical descent inequality and following standard non-convex SGD theory, we can derive a global convergence consequence for EasyTune.

**Corollary S1** (Asymptotic stationarity of EasyTune). *Suppose Assumptions* (A1)–(A3) *hold and that the step sizes satisfy* $\eta_k > 0$, $\sum_{k=0}^{\infty} \eta_k = \infty$ *and* $\sum_{k=0}^{\infty} \eta_k^2 < \infty$. *Then the EasyTune iterates satisfy*

$$\liminf_{K \to \infty} \frac{\sum_{k=0}^{K-1} \eta_k \mathbb{E}\big[\|\nabla \mathcal{L}(\theta_k)\|^2\big]}{\sum_{k=0}^{K-1} \eta_k} = 0, \tag{S30}$$

*and in particular*

$$\liminf_{k \to \infty} \mathbb{E}\big[\|\nabla \mathcal{L}(\theta_k)\|^2\big] = 0. \tag{S31}$$

*That is, EasyTune converges to first-order critical points in the standard non-convex sense.*

*Proof.* From Theorem S1 we have, for all $k \ge 0$,

$$\mathbb{E}[\mathcal{L}(\theta_{k+1})] \le \mathbb{E}[\mathcal{L}(\theta_k)] - c_1 \eta_k \mathbb{E}\big[\|\nabla \mathcal{L}(\theta_k)\|^2\big] + c_2 \eta_k^2. \tag{S32}$$

Summing this inequality over $k = 0, \ldots, K-1$ and using telescoping of the left-hand side gives

$$\mathbb{E}[\mathcal{L}(\theta_K)] \le \mathbb{E}[\mathcal{L}(\theta_0)] - c_1 \sum_{k=0}^{K-1} \eta_k \mathbb{E}\big[\|\nabla \mathcal{L}(\theta_k)\|^2\big] + c_2 \sum_{k=0}^{K-1} \eta_k^2. \tag{S33}$$

Rearranging the previous inequality to move the gradient term to the left-hand side, and then applying that $\mathcal{L}$ is bounded below by $\mathcal{L}_{\inf}$ (Assumption (A1)) together with $\mathbb{E}[\mathcal{L}(\theta_K)] \ge \mathcal{L}_{\inf}$ and the monotonicity $\sum_{k=0}^{K-1} \eta_k^2 \le \sum_{k=0}^{\infty} \eta_k^2$, we obtain

$$c_1 \sum_{k=0}^{K-1} \eta_k \mathbb{E}\big[\|\nabla \mathcal{L}(\theta_k)\|^2\big] \le \mathbb{E}[\mathcal{L}(\theta_0)] - \mathbb{E}[\mathcal{L}(\theta_K)] + c_2 \sum_{k=0}^{K-1} \eta_k^2 \le \mathbb{E}[\mathcal{L}(\theta_0)] - \mathcal{L}_{\inf} + c_2 \sum_{k=0}^{\infty} \eta_k^2. \tag{S34}$$

The right-hand side is finite by the assumptions on $\{\eta_k\}$, so letting

$$C_0 := \frac{\mathbb{E}[\mathcal{L}(\theta_0)] - \mathcal{L}_{\inf}}{c_1} + \frac{c_2}{c_1} \sum_{k=0}^{\infty} \eta_k^2 < \infty, \tag{S35}$$

we deduce

$$\sum_{k=0}^{\infty} \eta_k \mathbb{E}\big[\|\nabla \mathcal{L}(\theta_k)\|^2\big] \le C_0. \tag{S36}$$

Dividing both sides by $\sum_{k=0}^{K-1} \eta_k$ and letting $K \to \infty$ yields

$$0 \le \frac{\sum_{k=0}^{K-1} \eta_k \mathbb{E}\big[\|\nabla \mathcal{L}(\theta_k)\|^2\big]}{\sum_{k=0}^{K-1} \eta_k} \le \frac{C_0}{\sum_{k=0}^{K-1} \eta_k} \xrightarrow[K \to \infty]{} 0, \tag{S37}$$

where we used $\sum_k \eta_k = \infty$ in the last step. This proves the weighted-average statement. The liminf statement then follows: if there existed an $\varepsilon > 0$ and $K_0$ such that $\mathbb{E}[\|\nabla \mathcal{L}(\theta_k)\|^2] \ge \varepsilon$ for all $k \ge K_0$, the weighted average would be bounded below by $\varepsilon$, contradicting the previous limit. Hence $\liminf_{k \to \infty} \mathbb{E}[\|\nabla \mathcal{L}(\theta_k)\|^2] = 0$. $\square$

This corollary is a direct application of classical convergence theory for non-convex stochastic gradient methods; we include the standard argument above for completeness.

**Discussion.** Theorem S1 provides a one-step descent inequality showing that each EasyTune update decreases the expected training objective up to a small quadratic term in the step size. The corollary then instantiates the standard non-convex SGD theory in our diffusion-based motion tuning setting, proving that, under mild step-size conditions, the EasyTune iterates converge to first-order critical points in expectation. In other words, despite the stop-gradient design and step-aware sampling in Eq. (S9), EasyTune enjoys the same asymptotic convergence guarantees as classical stochastic gradient methods.

### C.3 PROOF OF EQ. (5)

*Proof.* Given a diffusion model $\epsilon_\theta$, and a reward model $\mathcal{R}_\phi$, the diffusion model is fine-tuned by maximizing the differentiable reward value:

$$\frac{\partial \mathcal{L}(\theta)}{\partial \theta} = -\mathbb{E}_{c \sim \mathcal{D}_\mathrm{T}, \mathbf{x}_0^\theta \sim \pi_\theta(\cdot|c)} \left[ \frac{\partial \mathcal{R}_\phi(\mathbf{x}_0^\theta)}{\partial \mathbf{x}_0^\theta} \cdot \frac{\partial \mathbf{x}_0^\theta}{\partial \theta} \right]. \tag{S38}$$

where $\pi_\theta$ denotes the reverse process defined in Eq. (2).

Here, we introduce Theorem 1 to compute $\frac{\partial \mathbf{x}_0^\theta}{\partial \theta}$, and thus we have:

$$\begin{aligned}
\frac{\partial \mathcal{L}(\theta)}{\partial \theta} &= -\mathbb{E}_{c \sim \mathcal{D}_\mathrm{T}, \mathbf{x}_0^\theta \sim \pi_\theta(\cdot|c)} \frac{\partial \mathcal{R}_\phi(\mathbf{x}_0^\theta)}{\partial \mathbf{x}_0^\theta} \cdot \frac{\partial \mathbf{x}_0^\theta}{\partial \theta} \\
&= -\mathbb{E}_{c \sim \mathcal{D}_\mathrm{T}, \mathbf{x}_0^\theta \sim \pi_\theta(\cdot|c)} \frac{\partial \mathcal{R}_\phi(\mathbf{x}_0^\theta)}{\partial \mathbf{x}_0^\theta} \cdot \left( \frac{\partial \pi_\theta(\mathbf{x}_1^\theta)}{\partial \theta} + \frac{\partial \pi_\theta(\mathbf{x}_1^\theta)}{\partial \mathbf{x}_1^\theta} \cdot \frac{\partial \mathbf{x}_1^\theta}{\partial \theta} \right) \\
&= -\mathbb{E}_{c \sim \mathcal{D}_\mathrm{T}, \mathbf{x}_0^\theta \sim \pi_\theta(\cdot|c)} \frac{\partial \mathcal{R}_\phi(\mathbf{x}_0^\theta)}{\partial \mathbf{x}_0^\theta} \cdot \left( \frac{\partial \pi_\theta(\mathbf{x}_1^\theta)}{\partial \theta} + \frac{\partial \pi_\theta(\mathbf{x}_1^\theta)}{\partial \mathbf{x}_1^\theta} \cdot \frac{\partial \pi_\theta(\mathbf{x}_2^\theta)}{\partial \theta} + \frac{\partial \pi_\theta(\mathbf{x}_1^\theta)}{\partial \mathbf{x}_1^\theta} \cdot \frac{\partial \pi_\theta(\mathbf{x}_2^\theta)}{\partial \mathbf{x}_2^\theta} \cdot \frac{\partial \mathbf{x}_2^\theta}{\partial \theta} \right) \\
&= \cdots \\
&= -\mathbb{E}_{c \sim \mathcal{D}_\mathrm{T}, \mathbf{x}_0^\theta \sim \pi_\theta(\cdot|c)} \frac{\partial \mathcal{R}_\phi(\mathbf{x}_0^\theta)}{\partial \mathbf{x}_0^\theta} \cdot \left( \sum_{t=1}^T \left( \prod_{s=1}^{t-1} \frac{\partial \pi_\theta(\mathbf{x}_s^\theta)}{\partial \mathbf{x}_s^\theta} \right) \cdot \frac{\partial \pi_\theta(\mathbf{x}_t^\theta)}{\partial \theta} \right).
\end{aligned}$$

$$\tag{S39}$$

The proof is completed. $\qquad\square$

