# OpenReview forum: "EasyTune: Efficient Step-Aware Fine-Tuning for Diffusion-Based Motion Generation"
_ICLR.cc/2026/Conference — ICLR 2026 Poster_

### Official Review · Reviewer_UAQn · 2025-10-25

**Soundness:** 3
**Presentation:** 2
**Contribution:** 2
**Rating:** 4
**Confidence:** 4

**Summary:**

The paper proposes EasyTune, a step-aware fine-tuning scheme for diffusion-based motion generation that optimizes at each denoising step using stop-gradient to break recursive dependencies; this yields fine-grained updates and O(1) memory rather than O(T). It trains a reward model without human labels via Self-refinement Preference Learning (SPL), mining preference pairs from retrieval results and fine-tuning a retrieval backbone as a differentiable reward.

**Strengths:**

- Clear motivation and theory; step-wise objective avoids vanishing gradients and heavy graph storage.
- Strong empirical results on alignment metrics (R-Precision/MM-Dist) with reduced memory footprint.
- Practical reward learning via SPL removes dependence on expensive human preference data.

**Weaknesses:**

- Limited discussion on when to use noise-aware vs. one-step (ODE) rewards and their sensitivity.
- Reward model is retrieval-based, which may under-capture physics/realism and encourage metric chasing. (Authors note limited physical grounding.)

**Questions:**

- Are the metrics giving a distorted signal? R-Precision exceeds the “Real” upper bound (0.581 > 0.511), yet FID is still notably higher than Real (0.132 vs 0.002). How do you interpret this divergence, and does it indicate semantic alignment without true distributional realism?
- Why must rewards be differentiable here? In RLHF for diffusion, rewards need not be differentiable. For motion, why is a differentiable reward essential—especially when non-differentiable motion metrics (e.g., foot-skate or slip measures) could be used as rewards?
- What exactly is being aligned? In EasyTune + SPL, is the alignment primarily to text–motion retrieval similarity, to human preference proxies, or to motion-quality attributes (e.g., fluency/physicality)? Please specify the concrete targets the gradients optimize toward.
- Missing citations ( PhysDiff, ReinDiffuse, Aligning Human Motion Generation with Human Perceptions ).
- I'm glad to raise the score if the author can address my concerns.

---

> ### Author Response · Authors · 2025-11-25
> **Response to Q1**
>
> > **[Q1]Are the metrics giving a distorted signal? R-Precision exceeds the “Real” upper bound (0.581 > 0.511), yet FID is still notably higher than Real (0.132 vs 0.002). How do you interpret this divergence, and does it indicate semantic alignment without true distributional realism?**
>
> **Response:** We thank the reviewer for this insightful question. This is indeed a crucial point to explain. We will break down our response into three parts:
> - **Why is it not a contradiction?** We will explain why R-Precision can exceed the “Real” benchmark while FID cannot.
> - **Is this phenomenon common?** We will show that this is a common and reasonable observation in recent literature and discuss what it implies about the state of the field.
> - **How does our method improve realism?** We will clarify how our method improves physical realism while simultaneously targeting semantic alignment.
>
> We discuss each of these points in detail below.
>
> ---
>
> > **[A-1] The divergence is not a contradiction but an expected result of how the two metrics are defined.**
>
> The core difference lies in their calculation:
>
> - **R-Precision** evaluates retrieval accuracy by measuring if the ground-truth text ranks in the top-k matches among 32 candidates (1 correct + 31 random negatives) for each generated motion.
>
> - **Frechet Inception Distance (FID)** evaluates distributional similarity. It compares the feature distribution of all generated motions against the distribution of all real motions in the test set. It measures the overall realism and variety of the generated results as a whole.
>
> Therefore, by definition, a model can generate motions that are an even better semantic match for their text descriptions than the ground-truth pairs in the dataset, thus **exceeding the "Real" R-Precision**. However, the theoretical best FID is 0 (a perfect match to the real data distribution), so it **cannot be surpassed**. (The observed 0.002 FID for "Real" data is due to minor computational noise).
>
> > **[A-2] This result is common in recent state-of-the-art methods and indicates that the field has prioritized and achieved a high level of semantic alignment.**
>
> Several concurrent works [4-8] also report R-Precision scores that surpass the "Real" benchmark while their FID scores remain higher. This pattern suggests:
> - The text-to-motion field has made significant progress on **semantic alignment**, with models now capable of generating motions that are highly faithful to the input text.
> - Achieving true **distributional realism** (as measured by FID) is a distinct, and in some ways harder, challenge that the community is still actively working on. Our method makes significant strides here, substantially improving FID over the baseline, which indicates a much more realistic generation distribution.
>
> We guess the R-Precision score can exceed the real data benchmark for two main reasons: (1) The ground-truth annotations in HumanML3D are imperfect, with some ambiguity across the three captions provided for each motion. (2) Research has heavily focused on improving text-motion semantic alignment, leading to rapid progress.
>
> > **[A-3] Our method improves physical realism by leveraging a reward model that has an implicit understanding of physical plausibility.**
>
> As discussed in our response to **[W2]**, while our retrieval-based reward model is explicitly trained on semantic similarity, it implicitly learns to distinguish in-distribution (real, physically plausible) motions from out-of-distribution (generated) ones. This provides a grounding signal that encourages more realistic motions. We provide a detailed quantitative analysis of this effect in **Appendix A.11** and further discussion in our response to **[Q3]**.
>
> ---
>
> We thank the reviewer for this excellent question. Finally, we acknowledge that the current standard metrics have limitations, and some contemporary works [1,2] have begun to explore alternatives. However, as there is no community consensus on a new standard, we have adhered to the established metrics to **ensure a fair comparison with prior work**. We agree that refining evaluation is a critical direction for future research.
>
> ---
>
> **References**
>
> [1] Go to Zero: Towards Zero-shot Motion Generation with Million-scale Data. ICCV 2025
>
> [2] MotionStreamer: Streaming Motion Generation via Diffusion-based Autoregressive Model in Causal Latent Space. ICCV 2025
>
> [3] Generating diverse and natural 3d human motions from text. ICCV 2022.
>
> [4] SoPo: Text-to-Motion Generation Using Semi-Online Preference Optimization. NeurIPS 2025
>
> [5] MaskControl: Spatio-Temporal Control for Masked Motion Synthesis. ICCV 2025
>
> [6] SnapMoGen: Human Motion Generation from Expressive Texts. NeurIPS 2025
>
> [7] FlashMo: Geometric Interpolants and Frequency-Aware Sparsity for Scalable Efficient Motion Generation. NeurIPS 2025
>
> [8] Autoregressive Motion Generation with Gaussian Mixture-Guided Latent Sampling. NeurIPS 2025

---

> ### Author Response · Authors · 2025-11-25
> **Response to Q2&Q4 (1/2)**
>
> > **[Q2] Why must rewards be differentiable here? In RLHF for diffusion, rewards need not be differentiable. For motion, why is a differentiable reward essential—especially when non-differentiable motion metrics (e.g., foot-skate or slip measures) could be used as rewards?**
>
> > **[Q4] Missing citations (PhysDiff, ReinDiffuse, Aligning Human Motion Generation with Human Perceptions).**
>
> **Response:** We thank the reviewer for these insightful questions. We understand the concern regarding the applicability of our method to non-differentiable rewards, particularly in the **physical-aware domain**. We also appreciate the reviewer pointing out the **missing citations**, which we have now discussed in the Section **Related Work**.
>
>
> In our paper, we do not claim that rewards for text-to-motion must be differentiable. The majority of reward models in recent **text-to-motion research** [1-3] are, in fact, **differentiable**. Our work aims to build upon this foundation to address the overhead and vanishing gradient problems associated with RL using such rewards. For the **physical-aware domain**, while non-differentiable models like ReinDiffuse (2024) [14] exist, recent studies also employ **differentiable physical-aware models** or **human perceptual models**, such as MotionCritic (2025) [7], and PP-Motion (2025) [8].
>
>
> To address your questions comprehensively, our response is structured as follows:
> - A discussion on the trade-offs between differentiable and non-differentiable reward models.
> - A review of reward model development in the **text-to-motion domain**.
> - A review of reward model development in the **physical-aware motion domain**.
> - A summary of EasyTune's applicability in both domains.
> ---
>
> > **[A-1] Differentiable vs. Non-Differentiable Reward Models**
>
> From optimization and application perspectives, they have distinct trade-offs:
>
> - From an **optimization perspective**, a differentiable reward allows the model to be optimized using gradient information, not just the scalar reward value. This provides **a richer, more direct signal about the direction** for improvement. The main drawback, of course, is the **dependency on the reward model's differentiability**.
>
> - From an **application perspective**, differentiable models (typically neural networks) offer flexibility in modeling **abstract concepts like human preferences or visual semantics**. Non-differentiable models (e.g., physics simulators, rule-based systems) can provide precise guidance for specific, well-defined goals. However, **perfectly satisfying these goals does not always guarantee physical plausibility**. For instance, as noted in MotionCritic [7], foot-ground contact metrics fail to penalize twisting arm motions that violate bio-mechanical constraints. It is also infeasible to manually define all the human motion rules in a handcrafted manner.
>
> ---
>
> > **[A-2] Reward Models in Text-to-Motion Generation**
>
> In the text-to-motion domain, recent post-training research has primarily focused on enhancing semantic consistency. Methods like **SoPo [2]**, **MoDiPO [1]**, and **MotionFlux [3]** optimize models on preference pairs. Since semantic correctness is inherently difficult to define with rules or simulators, **the reward models in this domain** are typically motion-text alignment models, which are implemented as neural networks and are thus **differentiable**.

---

> ### Author Response · Authors · 2025-11-25
> **Response to Q2&Q4 (2/2)**
>
> > **[A-3] Reward Models in Physical-Aware Motion Generation**
>
> We **agree with the reviewer** that **non-differentiable rewards are highly valuabl**e for physical-aware motion generation. However, there is also a growing body of research advocating for **differentiable rewards in this physical-aware domain**.
>
> A key line of research focuses on improving physical realism through learned reward models. Works like **MotionCritic [7]** and **PP-Motion [8]** model human perceptual preferences with neural networks to enhance physical plausibility. It has been noted [4-6] that methods using non-differentiable rewards often optimize in the pre-trained motion space while the reward operates in a separate, black-box feature space, potentially leading to limited feedback. Furthermore, several differentiable physics simulators now exist [12], such as the **XLA branch of MuJoCo [11]**, **PyBullet's Tiny Differentiable Simulator [9]**, and **Genesis [10]**, making gradient-based optimization in physical settings increasingly feasible.
>
>
> > **[A-4] Applicability of EasyTune**
>
>
> In summary, for the **text-to-motion** task, where the vast majority of reward models are differentiable, our method is directly applicable. For the **physical-aware motion** domain, where both approaches coexist, EasyTune can be readily applied to settings that use differentiable reward models, such as those in [7-11].
>
> Our use of a differentiable reward for the text-to-motion task aligns with the predominant approach in prior work. **The relative merits of each reward type are still an open topic of discussion in the community.**
>
> ---
>
> Thank you for your valuable suggestions. We have updated the discussion in **Related Work.** We hope this addresses your concerns.
>
> ---
>
> **References:**
>
> [1] MoDiPO: text-to-motion alignment via AI-feedback-driven Direct Preference Optimization, ArXiv 2024
>
> [2] SoPo: Text-to-Motion Generation Using Semi-Online Preference Optimization, NeurIPS 2025
>
> [3] MotionFlux: Efficient Text-Guided Motion Generation through Rectified Flow Matching and Preference Alignment, ArXiv 2025
>
> [4] When to trust your model: Model-based policy optimization, NeurIPS 2019.
>
> [5] Controlvae: Model-based learning of generative controllers for physics-based characters, SIGGRAPH 2022.
>
> [6] Moconvq: Unified physics-based motion control via scalable discrete representations, SIGGRAPH 2023.
>
> [7] Aligning Human Motion Generation with Human Perceptions, ICLR 2025
>
> [8] PP-Motion: Physical-Perceptual Fidelity Evaluation for Human Motion Generation, ACM MM 2025
>
> [9] Pybullet, a python module for physics simulation for games, robotics and machine learning, 2016-2019
>
> [10] Genesis: A universal and generative physics engine for robotics and beyond, 2024
>
> [11] Mujoco: A physics engine for model-based control, 2011
>
> [12] A Survey: Learning Embodied Intelligence from Physical Simulators and World Models, 2025
>
> [13] PhysDiff: Physics-Guided Human Motion Diffusion Model, ICCV 2023
>
> [14] ReinDiffuse: Crafting Physically Plausible Motions with Reinforced Diffusion Model, WACV 2025
>
> [15] Aligning Human Motion Generation with Human Perceptions, ICLR 2025

---

> ### Author Response · Authors · 2025-11-25
> **Response to Q3**
>
> > **[Q3] What exactly is being aligned? In EasyTune + SPL, is the alignment primarily to text–motion retrieval similarity, to human preference proxies, or to motion-quality attributes (e.g., fluency/physicality)? Please specify the concrete targets the gradients optimize toward.**
>
> **Response:** Thank you for this critical question. To be precise, our method **explicitly aligns the generated motion with the text semantics**. We have revised the manuscript to state this clearly in **Lines 131-133** and **Line 527**.
>
> We believe the core of the reviewer's question may be: *why does explicitly aligning for text-motion semantics also lead to an improvement in physical realism?* We attribute this to a phenomenon observed in recent works [1-3], where explicitly optimizing for an in-domain reward (in our case, semantic similarity) can implicitly improve out-of-domain performance (such as physical realism).
>
> Further, we think this phenomenon arises from the following two reasons:
> 1.  As discussed in our response to **[W2]**, the reward model, while trained for semantics, possesses an **implicit physical perception capability**. It has learned to distinguish in-distribution (real) data from out-of-distribution (generated) data, which allows it to improve overall motion quality.
> 2.  Our method aligns the generated motion embedding to the text embedding within the reward model's feature space. Since the reward model was pre-trained by aligning *real* motion embeddings to the same text embeddings, these two alignment processes share a common text embedding space. Therefore, by explicitly aligning the generated motion to the text, we are **implicitly aligning it to the distribution of real motions**.
>
> ---
>
> We thank the reviewer for this insightful question, which has helped us clarify the underlying mechanics of our approach.
>
> ---
>
> **References:**
>
> [1] Mixgrpo: Unlocking flow-based grpo efficiency with mixed ode-sde
>
> [2] DanceGRPO: Unleashing GRPO on Visual Generation
>
> [3] Branchgrpo: Stable and efficient grpo with structured branching in diffusion models

---

> ### Author Response · Authors · 2025-11-25
> **Response to W1**
>
> > **[W1] Limited discussion on when to use noise-aware vs. one-step (ODE) rewards and their sensitivity.**
>
> **Response**: We thank the reviewer for this excellent question. We agree that providing a clear analysis of the sensitivity of our reward mechanisms and guidance on when to use the **noise-aware** versus the **one-step (ODE)** strategy is crucial.
>
> In response, we have added extensive quantitative and qualitative analyses to the revised manuscript to address this point directly. Our response is structured as follows:
> - We first provide a new sensitivity analysis of the noised states perceptibility for both strategies across both image and motion domains in **Appendix A.6**.
> - We then demonstrate the robustness of our reward models on noisy data through a retrieval task analysis in **Appendix A.10.3**.
> - Finally, we have made the direct comparison in our main results (Table 2) more explicit and added a new summary and discussion in **Appendix B.5** to provide clear guidance.
>
> ---
>
> > **[A-1] Sensitivity and Perceptibility Analysis (Appendix A.6)**
>
> To analyze the perceptibility and sensitivity of noisy states, we conducted an experimental to analyze the perceptibility and sensitivity of noisy states in both image and motion generation. The full analysis is in **Appendix A.6**.
>
> **Summary of Findings**: As shown in **Fig. S4, S5, and S6**, the predictability of noisy data is high in the motion domain, meaning the reward model can reliably interpret noisy intermediate states. This makes the **noise-aware** strategy highly effective. For images, predictability is weaker in high-noise steps, making the **one-step** reward (which uses the denoised prediction) a more reliable choice. This confirms that both strategies are viable, and the optimal choice depends on the reward model's noise perception capability in the given domain.
>
> > **[A-2] Robustness Analysis on Noisy Data (Appendix A.10.3)**
>
> To further demonstrate the robustness of our approach, we evaluated the performance of the underlying retrieval models on noisy and denoised motion inputs. The results are now in **Appendix A.10.3**.
>
> **Summary of Findings**: The results show that both the noise-aware models (ReAlign and SPL) are remarkably stable when given noisy inputs. The one-step strategy (TMR on predicted clean data) also recovers its performance effectively. This demonstrates that our reward signals are robust and reliable.
>
> | Method | Input Data | T2M R@1 | T2M R@2 | T2M R@3 | T2M R@5 | T2M R@10 | M2T R@1 | M2T R@2 | M2T R@3 | M2T R@5 | M2T R@10 |
> | :--- | :--- | :--- | :--- | :--- | :--- | :--- | :--- | :--- | :--- | :--- | :--- |
> | ReAlign &nbsp; &nbsp; | Clean Data | 67.59 | 82.24 | 87.44 | 91.97 | 96.28 | 68.94 | 82.86 | 87.95 | 92.44 | 96.28 |
> | ReAlign | Noisy Data | 67.20 | 81.46 | 87.11 | 91.39 | 95.67 | 68.02 | 81.84 | 87.56 | 91.39 | 95.69 |
> | ReAlign | Pred. Clean &nbsp; &nbsp; | 67.80 | 82.38 | 87.86 | 92.27 | 96.61 | 68.04 | 82.47 | 87.97 | 92.10 | 96.34 |
> | SPL | Clean Data | 69.31 | 83.71 | 88.66 | 92.81 | 96.75 | 70.23 | 83.41 | 88.72 | 93.07 | 97.04 |
> | SPL | Noisy Data | 69.36 | 83.63 | 88.53 | 92.83 | 96.76 | 70.34 | 83.41 | 88.66 | 93.04 | 96.93 |
> | SPL | Pred. Clean | 68.39 | 83.31 | 88.59 | 93.11 | 96.73 | 68.60 | 82.35 | 88.06 | 92.79 | 96.53 |
> | TMR | Clean Data | 67.16 | 81.32 | 86.81 | 91.43 | 95.36 | 67.97 | 81.20 | 86.35 | 91.70 | 95.27 |
> | TMR | Pred. Clean | 66.98 | 81.04 | 87.09 | 92.11 | 95.74 | 68.32 | 80.69 | 86.43 | 92.13 | 95.84 |
>
> > **[A-3] Direct Comparison and Guidance (Appendix B.5)**
>
> We acknowledge that the comparison between the two strategies in the original manuscript could have been clearer. While the results were included in **Tables 2 and 3**, we have now made this comparison more explicit in **Table 2** and added a dedicated discussion and summary in **Appendix B.5**.
>
> **Summary of Discussion**: The results, consolidated below, show that the **Noise-Aware** reward consistently achieves slightly better performance across all metrics for both MLD and MLD++ models. Therefore, we recommend using the **Noise-Aware** strategy if the reward model has sufficient noise-perception capability. Otherwise, the **One-Step** reward provides a strong and comparable alternative.
>
> | Model | Strategy | R-Prec@1 &nbsp; &nbsp; | R-Prec@2 &nbsp; &nbsp; | R-Prec@3 &nbsp; &nbsp; | FID ↓ | MM-Dist ↓ |
> | :--- | :--- | :--- | :--- | :--- | :--- | :--- |
> | MLD | One-Step | 0.568 | 0.754 | 0.846 | 0.194 | 2.672 |
> | MLD | Noise-Aware &nbsp; &nbsp; | **0.581** | **0.769** | **0.855** | **0.132** | **2.637** |
> | MLD++ &nbsp; &nbsp; | One-Step | 0.581 | 0.762 | 0.849 | 0.073 | 2.603 |
> | MLD++ | Noise-Aware | **0.591** | **0.777** | **0.859** | **0.069** | **2.592** |
>
> ---
>
> We thank the reviewer for prompting these important clarifications. We have added these new analyses to **Appendices A.6, A.10.3, and B.5** and updated our main paper to make the comparisons and guidance more explicit.

---

> ### Author Response · Authors · 2025-11-25
> **Response to W2**
>
> > **[W2] Reward model is retrieval-based, which may under-capture physics/realism and encourage metric chasing. (Authors note limited physical grounding.)**
>
> **Response**: We thank the reviewer for this critical question. We agree on the concern regarding the lack of an explicit physics simulation in our reward model. Our response is two-fold:
>
>
> - While our retrieval-based reward model does not explicitly model physics, **it can still implicitly judge physical realism**. We demonstrate this with a new quantitative analysis in **Appendix A.11**.
>
>
> - We acknowledge that existing reward models typically **focus on either semantics [1-5] or physics [6-9]**, and that developing a unified model would be valuable to explore. However, our work focuses on the **text-to-motion generation task**, where the community [1-5] has prioritized semantic alignment. We agree that **physical motion generation** [6-9] is an important direction and have added discussion on this to our **related work** and **limitations** sections.
>
>
> To verify that our reward model captures physical realism, we conducted a direct evaluation of its physical perception ability, with the full analysis now in **Appendix A.11**.
>
> ---
>
> > **[A] Physical Perception Analysis**
>
> **Experiment:** To investigate the physical perception capabilities of our reward model, we conducted an experiment to assess its ability to distinguish between real and generated motions. We selected 50 prompts and their corresponding ground-truth motions ($\mathbf{x}\_\text{gt}$) from the HumanML3D test set. For each prompt, we also generated a motion ($\mathbf{x}\_\text{fake}$) using the MLD model. We then evaluated both motions using our reward model with an empty text condition ($c=\text{`'}$), obtaining reward scores $r(\mathbf{x}\_\text{gt}, c)$ and $r(\mathbf{x}\_\text{fake}, c)$.
>
> **Result:** The model assigned a higher reward to the real, ground-truth motion in **96% of cases (48 out of 50)**, demonstrating a strong preference for physically plausible motions.
>
> | Comparison | Count |
> | :--- | :--- |
> | $r(\mathbf{x}\_\text{gt}, c)$ > $r(\mathbf{x}\_\text{fake}, c)$ | 48 |
> | $r(\mathbf{x}\_\text{gt}, c)$ < $r(\mathbf{x}\_\text{fake}, c)$ | 2 |
>
> **This result suggests the reward model has an implicit understanding of physical realism**, likely by learning to distinguish in-distribution (real) data from out-of-distribution (generated) data. This capability serves as a grounding mechanism against pure metric chasing.
>
> As a further piece of evidence, our method, along with other RL baselines, **not only improves semantic alignment but also enhances distributional consistency with real data (e.g., FID score)**, as shown in **Table 1**. Further discussion for this phenomenon is provided in **[Q3]**.
>
> ---
>
> We thank the reviewer for prompting this analysis, which we believe strengthens our claims. The corresponding analysis and discussions have been added to **Appendix A.11**, the **related work**, and the **limitations** sections of the revised paper.
>
> **References:**
>
> [1] MoDiPO: text-to-motion alignment via AI-feedback-driven Direct Preference Optimization, ArXiv 2024
>
> [2] SoPo: Text-to-Motion Generation Using Semi-Online Preference Optimization, NeurIPS 2025
>
> [3] MotionFlux: Efficient Text-Guided Motion Generation through Rectified Flow Matching and Preference Alignment, ArXiv 2025
>
> [4] ReMoDiffuse: Retrieval-Augmented Motion Diffusion Model, NeurIPS 2023
>
> [5] AToM: Aligning Text-to-Motion Model at Event-Level with GPT-4Vision Reward, CVPR 2025
>
> [6] Aligning Human Motion Generation with Human Perceptions
>
> [7] Controlvae: Model-based learning of generative controllers for physics-based characters.
>
> [8] ReinDiffuse: Crafting Physically Plausible Motions with Reinforced Diffusion Model
>
> [9] PP-Motion: Physical-Perceptual Fidelity Evaluation for Human Motion Generation

---

> ### Comment · Reviewer_UAQn · 2025-11-28
>
> Well rebuttal! I will raise my score ASAP!

---

> > ### Author Response · Authors · 2025-12-01
> >
> > Thank you very much for your positive feedback. We truly appreciate the time and effort you put into reviewing our work.

---

### Official Review · Reviewer_pyGX · 2025-10-26

**Soundness:** 3
**Presentation:** 3
**Contribution:** 3
**Rating:** 6
**Confidence:** 4

**Summary:**

# Summary
The paper aims at aligning motion diffusion models with downstream objectives using differentiable rewards. The authors first diagnose a root cause behind prior methods’ inefficiency and memory use: recursive gradient dependence across denoising steps. They propose EasyTune, which fine‑tunes at each denoising step by stopping gradients through the current sample and backpropagating only through the local denoiser. This decouples steps, enabling dense, fine‑grained updates with O(1) step memory. To address the scarcity of motion preference data, they introduce Self‑refinement Preference Learning (SPL) that adapts a retrieval model into a reward model without human‑labeled pairs.
Experiments on HumanML3D and KIT‑ML across multiple backbones (MLD/MLD++, MotionLCM, MDM, MotionDiffuse) report consistent improvements and lower peak memory than trajectory‑level fine‑tuning baselines.

**Strengths:**

- The paper cleanly shows how chain‑rule recursion creates sparse/vanishing gradients and large memory graphs and contrasts chain vs. step optimization. The motivation is clear and reasonable.
- The proposed method seems to be a simple, general, and effective idea: The per‑step objective with `sg(·)` is easy to implement on standard motion diffusers.
- Results span two datasets and several backbones; on HumanML3D, EasyTune improves R‑Precision/FID/MM‑Dist vs. DRaFT/AlignProp/DRTune while using less peak memory than those baselines. EasyTune also achieves faster convergence curves for optimization.
- The authors promised to release the code and models.
- Most details are provided in the paper and appendix. Video result (website) is well-made.

**Weaknesses:**

I am a bit worried about the technical novelty (but it seems ok: The fine-tuning input x_t can be obtained by denoising sampling - seems new). But Theorem 1 is a direct chain‑rule decomposition (can it be called a Theorem?); Theorem 2 is the local derivative after inserting `sg(·)`. Both are too straightforward to be called Theorem...  Besides, SPL improvements are modest... On HumanML3D retrieval, SPL improves ReAlign by R@1 +2.5% / R@3 +1.4%; on KIT‑ML, R@5 +2.2%. In the Limitation section, the authors note SPL’s reliance on retrieval mining may introduce noisy/ambiguous pairs.

Minor:
- The Implementation section states LR=2×10⁻⁴, batch=128, whereas Tab.S1 shows LR=1×10⁻⁵, batch=256 for all methods including EasyTune?
- Please also end‑to‑end costs, EasyTune’s total compute/time…Fig.5’s “w/o BP” curves illustrate the theoretical O(1) vs. O(T) difference without backprop; training, however, always with backpropagates.

**Questions:**

Besides the questions in the above section:

The objective uses uniform \(t \sim U(0, T)\), yet the paper argues early, high‑noise steps are critical and under‑optimized by chain methods. Would non‑uniform sampling or up‑weighting early steps help?

In SPL, the authors set K=10. How do different K and different retrieval pools (subset vs. full) impact reward quality and final generation?

Any failure cases (I mean figures/videos)?

**Details Of Ethics Concerns:**

/

---

> ### Author Response · Authors · 2025-11-25
> **Response to Q1**
>
> > **[W1] The objective uses uniform $t \sim U(0, T)$, yet the paper argues early, high‑noise steps are critical and under‑optimized by chain methods. Would non‑uniform sampling or up‑weighting early steps help?**
>
> **Response**: We thank the reviewer for this insightful question. This is an excellent point. While our main experiments use a uniform sampling schedule for simplicity, we agree that exploring non-uniform weighting is a valuable direction.
>
> To investigate the effect of different weighting strategies, we conducted an ablation study on step-level reward reweighting. We evaluated four alternative schemes and compared them against our standard uniform approach ('EasyTune (Full)'). The strategies, summarized in **Appendix A.8**, are: (1) optimizing only the **final 20 steps**; (2) optimizing only the **initial 20 steps**; (3) using linearly increasing weights to **emphasize later steps**; and (4) using linearly decreasing weights to **emphasize earlier steps**.
>
> The results are summarized in the table below.
>
> | Method | R@1 | R@2 | R@3 | FID ↓ | MM Dist ↓ | Diversity → | Memory (GB) ↓ |
> | :--- | :--- | :--- | :--- | :--- | :--- | :--- | :--- |
> | Real | 0.511 | 0.703 | 0.797 | 0.002 | 2.974 | 9.503 | - |
> | MLD (Baseline) | 0.481 | 0.673 | 0.772 | 0.473 | 3.196 | 9.724 | 15.21 |
> | w/ DRaFT-50 | 0.528 | 0.724 | 0.819 | 0.197 | 2.872 | 9.641 | 37.32 |
> | w/ EasyTune (Full) | 0.581 | 0.769 | 0.855 | 0.132 | 2.637 | 9.465 | 22.10 |
> | EasyTune + (1) | 0.546 | 0.735 | 0.804 | 0.184 | 2.815 | 9.682 | 22.10 |
> | EasyTune + (2) | 0.567 | 0.759 | 0.842 | 0.158 | 2.673 | 9.430 | 22.10 |
> | EasyTune + (3) | 0.556 | 0.748 | 0.838 | 0.147 | 2.652 | 9.421 | 22.10 |
> | EasyTune + (4) | 0.584 | 0.773 | 0.859 | 0.136 | 2.631 | 9.521 | 22.10 |
>
> Our findings confirm the reviewer's intuition. While focusing exclusively on initial or final steps (strategies 1 & 2) degrades performance, reweighting the rewards is beneficial. Notably, **linearly decreasing weights to emphasize earlier steps (strategy 4) performs best**, even slightly outperforming our uniform baseline. This result strongly supports the hypothesis that accurately optimizing the high-noise, early diffusion steps is critical for achieving high-quality generation.
>
> ---
>
> We sincerely thank the reviewer for this valuable suggestion, which has led to a deeper analysis of the optimization schedule. We have added this ablation study and discussion to **Appendix A.8** in our revised manuscript.

---

> ### Author Response · Authors · 2025-11-25
> **Response to Q2**
>
> > **[Q2] In SPL, the authors set K=10. How do different K and different retrieval pools (subset vs. full) impact reward quality and final generation?**
>
> **Response**: We thank the reviewer for this excellent question. We agree that analyzing the sensitivity of our Self-refinement Preference Learning (SPL) to the candidate number $K$ and the retrieval pool is crucial for understanding the method's stability and performance.
>
> In response, we have conducted a thorough sensitivity analysis and added the results to **Appendix A.10.2** of the revised manuscript. We summarize the key findings below.
>
> ---
>
> > **[A] Analysis on Candidate Size $K$ and Retrieval Pools**
> >
> **Mechanism :** The stability of our online SPL hinges on the retrieval success rate. This rate is controlled by the candidate number ($K$) and the retrieval pool design. Our empirical finding is that a retrieval failure ratio of approximately 1:20 yields stable and effective optimization.
>
> Pool Setting. We consider four retrieval pool settings in TMR: (a) the entire test set, (b) the entire test set with a high text-similarity threshold, (c) a “dissimilar subset” of 100 motion–text pairs, and (d) “small batches” of 32 randomly sampled pairs. Settings (a) and (b) are computationally expensive and yield low retrieval success rates, so they are impractical for online mining. Settings (c) and (d) give much higher success rates: (c) provides a cleaner but easier pool, while (d) better matches realistic online mining where negatives come from small random batches. We therefore adopt setting (d) in our main experiments: with (K=10) it achieves a failure ratio of about (1{:}20) and stable optimization, whereas setting (c) gives a higher (\sim 1{:}6) failure ratio and less stable performance.
>
>
> **Results:** We studied the impact of $K \in \{10, 15, 20\}$ for both pool settings (c) and (d). The results are summarized in the table below.
>
> | Method | R-Prec@1 | R-Prec@2 | R-Prec@3 | FID ↓ | MM Dist ↓ | Diversity → |
> | :--- | :--- | :--- | :--- | :--- | :--- | :--- |
> | Real | 0.511 | 0.703 | 0.797 | 0.002 | 2.974 | 9.503 |
> | MLD (Baseline) | 0.481 | 0.673 | 0.772 | 0.473 | 3.196 | 9.724 |
> | $K=10$ + (d) | 0.581 | 0.769 | 0.855 | 0.132 | 2.637 | 9.465 |
> | $K=15$ + (d) | 0.571 | 0.758 | 0.843 | 0.142 | 2.668 | 9.486 |
> | $K=20$ + (d) | 0.564 | 0.747 | 0.830 | 0.184 | 2.704 | 9.629 |
> | $K=10$ + (c) | - | - | - | - | - | - |
> | $K=15$ + (c) | 0.585 | 0.773 | 0.859 | 0.155 | 2.626 | 9.428 |
> | $K=20$ + (c) | 0.574 | 0.759 | 0.844 | 0.149 | 2.653 | 9.495 |
>
> The results show that SPL consistently improves over the baseline across all valid settings. Our chosen setting of $K=10$ with pool (d) provides a strong and stable performance boost. In practice, we recommend a brief sensitivity analysis to find a configuration that yields a ~1:20 failure ratio for robust optimization.
>
> ---
>
> We sincerely thank the reviewer for this valuable question. Following your advice, we have added this detailed sensitivity analysis to **Appendix A.10.2** of the revised manuscript.

---

> ### Author Response · Authors · 2025-11-25
> **Response to Q3**
>
> > **[Q3] Any failure cases (I mean figures/videos)?**
>
> **Response**: We thank the reviewer for this important question. We agree that analyzing failure cases is crucial for a complete evaluation.
>
> In response, we have added a failure case analysis focused on **reward hacking** to **Appendix A.7** of our revised manuscript. We provide both qualitative examples (figures and videos) and discuss potential mitigation strategies. We summarize the key points below.
>
> ---
>
> > **[A] Failure Case Analysis: Reward Hacking**
>
> A well-known challenge in RL lies in reward hacking, where the model over-optimizes the reward model. In our case, we observe this can happen when fine-tuning continues past the point of convergence.
>
> When reward hacking arises,  the model may generate motions that are semantically over-aligned with the text but are physically unrealistic or unnatural. For example:
> > - Given the prompt, ``"A person stands up from the ground, lifts their right foot, and sets it back down,"``  a hacked model might generate a person **continuously** lifting their foot to maximize the `"lifts their right foot"` signal.
> > - For `"A person squats down, then stands up and moves forward,"` the model might misinterpret this as `"A person squats down *while* moving forward."`
>
> We provide visualizations of these failure cases in **Figure S7** in the **Appendix A.7** and include corresponding **videos in the supplementary material** for better visualization.
>
> Fortunately, as discussed in **Appendix A.3**, we have found that this issue can be effectively mitigated by combining our method with some common engineering techniques, such as **KL-divergence regularization** and early stopping strategies.
>
> ---
>
> We thank the reviewer for prompting this discussion. We have included this failure analysis, along with figures and a reference to supplementary videos, in **Appendix A.7** of the revised manuscript.

---

> ### Author Response · Authors · 2025-11-25
> **Response to W1**
>
> > **[W1] The Implementation section states LR = $2\times 10^{-4}$, batch = $256$, whereas Tab.~S1 shows LR = $1\times 10^{-5}$, batch = $256$ for all methods including EasyTune?**
>
> **Response**: We apologize for the oversight. The correct settings are LR = $1\times 10^{-5}$ and batch size = $256$. We have updated these values in **Sec. 5.1**. Additionally, our ablation study on the learning rate showed that results are not sensitive to this hyperparameter (see **Appendix A.10.4**). To ensure full **reproducibility**, we **commit to releasing all code and model weights** upon acceptance.

---

> ### Author Response · Authors · 2025-11-25
> **Response to W2**
>
> > **[W2] Please also end‑to‑end costs, EasyTune’s total compute/time…Fig.5’s “w/o BP” curves illustrate the theoretical O(1) vs. O(T) difference without backprop; training, however, always with backpropagates.**
>
> **Response**: We thank the reviewer for this valuable suggestion. We agree that a practical, end-to-end cost analysis, including backpropagation, is crucial, and add a comprehensive end-to-end cost analysis in **Appendix A.9.1**.
>
> > **[A-1] Comprehensive Memory overhead**
>
> To directly address this, we have included a comprehensive end-to-end cost analysis in the revised manuscript. We provide a detailed memory overhead breakdown in **Appendix A.9.1**, reporting the the end-to-end GPU memory usage of EasyTune and existing fine-tuning methods (**Figure S8**).
>
> > **[A-2] Computational overhead (training time and TFLOPs)**
>
> Beyond memory overhead, we also provide a comprehensive analysis of both per-step optimization time and the total training time and TFLOPs required to reach different reward thresholds. All experiments were conducted on a single NVIDIA RTX A6000 GPU. The key results, which account for the full training process, are summarized in the table below (where “–” indicates a method failed to reach the target reward within the training budget).
>
>
> | | DRaFT | AlignProp | DRTune | ReFL | EasyTune (Ours) |
> | :--- | :--- | :--- | :--- | :--- | :--- |
> | Time per Opt. (s) | 5.61 | 5.17 | 4.90 | 4.72 | **1.47** |
> | **Reward = 0.70** Time (s) | 466.27 | 271.99 | 554.77 | 820.29 | **263.36** |
> | **Reward = 0.70** TFLOPs | 18044 | 10526 | 21469 | 31745 | **10191** |
> | **Reward = 0.75** Time (s) | 2616.54 | 971.55 | 2009.59 | - | **358.17** |
> | **Reward = 0.75** TFLOPs | 101260 | 37599 | 77771 | - | **13861** |
> | **Reward = 0.80** Time (s) | - | - | - | - | **452.53** |
> | **Reward = 0.80** TFLOPs | - | - | - | - | **17513** |
> | **Reward = 0.85** Time (s) | - | - | - | - | **1025.17** |
> | **Reward = 0.85** TFLOPs | - | - | - | - | **39674** |
>
> ---
>
> We sincerely thank the reviewer for this valuable suggestion. Following your advice, we have updated the corresponding results in **Appendix A.9** of the revised manuscript.

---

> ### Author Response · Authors · 2025-11-25
> **Response to Weaknesses**
>
> > **[Weaknesses] I am a bit worried about the technical novelty (but it seems ok: The fine-tuning input x_t can be obtained by denoising sampling - seems new). But Theorem 1 is a direct chain‑rule decomposition (can it be called a Theorem?); Theorem 2 is the local derivative after inserting sg(·). Both are too straightforward to be called Theorem... Besides, SPL improvements are modest... On HumanML3D retrieval, SPL improves ReAlign by R@1 +2.5% / R@3 +1.4%; on KIT‑ML, R@5 +2.2%. In the Limitation section, the authors note SPL’s reliance on retrieval mining may introduce noisy/ambiguous pairs.**
>
> **Response**: We thank the reviewer for the detailed feedback and constructive suggestions.
>
> ---
>
> > **[A-1] Regarding "Theorem"**
>
> We appreciate the reviewer's feedback on the naming. We agree that "Corollary" is an imprecise term for these results. Accordingly, we have revised the manuscript to rename **Theorem 1 and Theorem 2** as **Corollariy 1 and Corollariy 2** to better reflect their nature as direct mathematical derivations.
>
> > **[A-2] Regarding the significance of SPL improvements**
>
> We understand the reviewer's concern about the magnitude of the improvements. While individual metric gains may seem modest, we believe the **average improvement is consistent and meaningful**, especially when compared to prior work.
>
> On HumanML3D, our Self-Preference Learning (SPL) improves upon the strong ReAlign baseline by **+1.15%** for Text-to-Motion (T2M) retrieval and **+0.80%** for Motion-to-Text (M2T) retrieval on average (across R@1 to R@10). For context, this gain is more substantial than that of LaMP (ICLR 2025) over its TMR (ICCV 2023) baseline, which was +0.35 (T2M) and +0.84 (M2T).
>
> | Method | Avg R@1-@10 (T2M) | Avg R@1-@10 (M2T) |
> | :--- | :--- | :--- |
> | TEMOS | 62.05 | 62.71 |
> | T2M | 78.08 | 78.19 |
> | TMR | 84.42 | 84.50 |
> | LaMP | 84.77 | 85.34 |
> | ReAlign (Baseline) | 85.10 | 85.69 |
> | **w/ SPL (Ours)** | **86.25** | **86.49** |
>
> > **[A-3] Regarding the reliance on noisy/ambiguous mined pairs**
>
> We agree that the reliability of mined preference pairs and the dependency on the retrieval model are crucial points. We have conducted extensive analysis to address these concerns, with full details in **Appendix A.10.1, A.10.2, and A.10.3**. Our response covers three key areas:
>
> **1. Independence from a Specific High-Performing Model:** To test SPL's dependency on the retrieval model's strength, we conducted a sensitivity analysis using TMR, a weaker model. As shown in **Appendix A.10.1**, SPL consistently improves TMR's performance (e.g., T2M R@1 improves from 67.16 to 68.76). This demonstrates that SPL is not restricted to a single strong model and can benefit weaker ones.
>
> | Method | R-Precision Top 1 | Top 2 | Top 3 | FID ↓ | MM Dist ↓ | Diversity → |
> | --- | --- | --- | --- | --- | --- | --- |
> | Real | 0.511 | 0.703 | 0.797 | 0.002 | 2.974 | 9.503 |
> | MLD (Baseline) | 0.481 | 0.673 | 0.772 | 0.473 | 3.196 | 9.724 |
> | +TMR (w/ SPL, Step) | 0.573 | 0.760 | 0.843 | 0.173 | 2.682 | 9.942 |
> | +TMR (w/ SPL, Chain) | 0.567 | 0.753 | 0.836 | 0.158 | 2.698 | 9.874 |
>
>
> **2. Robustness to Input Noise:** To directly assess the impact of noise, we evaluated SPL on noisy motion inputs. As detailed in **Appendix A.10.3** and summarized below, SPL maintains its performance advantage even when trained with noisy data. For instance, SPL trained on noisy inputs (T2M R@1: 69.36) still outperforms the strong baseline trained on clean inputs (ReAlign T2M R@1: 67.59).
>
> | Method | Input | T2M R@1 | T2M R@2 | T2M R@3 | T2M R@5 | T2M R@10 | M2T R@1 | M2T R@2 | M2T R@3 | M2T R@5 | M2T R@10 |
> | --- | --- | --- | --- | --- | --- | --- | --- | --- | --- | --- | --- |
> | ReAlign | Clean | 67.59 | 82.24 | 87.44 | 91.97 | 96.28 | 68.94 | 82.86 | 87.95 | 92.44 | 96.28 |
> | ReAlign | Noisy | 67.20 | 81.46 | 87.11 | 91.39 | 95.67 | 68.02 | 81.84 | 87.56 | 91.39 | 95.69 |
> | ReAlign | Pred. Clean | 67.80 | 82.38 | 87.86 | 92.27 | 96.61 | 68.04 | 82.47 | 87.97 | 92.10 | 96.34 |
> | SPL | Clean | 69.31 | 83.71 | 88.66 | 92.81 | 96.75 | 70.23 | 83.41 | 88.72 | 93.07 | 97.04 |
> | SPL | Noisy | 69.36 | 83.63 | 88.53 | 92.83 | 96.76 | 70.34 | 83.41 | 88.66 | 93.04 | 96.93 |
> | SPL | Pred. Clean | 68.39 | 83.31 | 88.59 | 93.11 | 96.73 | 68.60 | 82.35 | 88.06 | 92.79 | 96.53 |
> | TMR | Clean | 67.16 | 81.32 | 86.81 | 91.43 | 95.36 | 67.97 | 81.20 | 86.35 | 91.70 | 95.27 |
> | TMR | Pred. Clean | 66.98 | 81.04 | 87.09 | 92.11 | 95.74 | 68.32 | 80.69 | 86.43 | 92.13 | 95.84 |
>
> These combined results demonstrate that our approach is robust, not overly reliant on a specific model, and contains mechanisms to ensure the reliability of mined pairs.
>
> ---
>
> We thank the reviewer for prompting this analysis, which we believe strengthens our paper.

---

> > ### Comment · Reviewer_pyGX · 2025-11-28
> >
> > Thank you for the thorough response to my concerns. The extensive new experiments are somewhat beyond what I expected and clearly demonstrate the effectiveness of the proposed method. I am in favor of this paper and will maintain my current score.

---

> > > ### Author Response · Authors · 2025-12-01
> > >
> > > Thank you very much for your positive evaluation. We greatly appreciate your time and thoughtful review.

---

### Official Review · Reviewer_v4NY · 2025-10-28

**Soundness:** 4
**Presentation:** 3
**Contribution:** 4
**Rating:** 10
**Confidence:** 3

**Summary:**

The paper addresses the challenge of efficiently fine-tuning diffusion-based motion generation models using differentiable rewards, which suffer from coarse-grained optimization and extremely high memory usage due to recursive gradient dependencies across denoising steps. The paper propose EasyTune, a step-aware fine-tuning framework that optimizes the diffusion model at each denoising step independently instead of over the full trajectory. In EasyTune, the model computes the reward of the current noised motion, applies a stop-gradient operation to decouple dependencies, updates parameters locally, and clears the computation graph before moving to the next step—achieving fine-grained supervision and constant memory cost. To overcome the lack of human preference data, the paper further introduces Self-refinement Preference Learning (SPL), which adapts a pre-trained text-motion retrieval model into a reward model by automatically mining motion preference pairs from retrieval errors and fine-tuning them using a KL-divergence objective. Experiments on HumanML3D and KIT-ML with six pre-trained motion diffusion models show that EasyTune significantly outperforms prior fine-tuning methods such as DRaFT and DRTune.

**Strengths:**

- The paper addresses an important and timely problem in diffusion model fine-tuning, providing a thorough analysis of the limitations in existing differentiable reward methods.
- The proposed EasyTune framework is conceptually clear and mathematically well-formulated, offering a principled solution to recursive gradient and memory inefficiency issues.
- The authors conduct extensive experiments and comparisons across multiple datasets and backbone models, demonstrating consistent and substantial performance gains with strong quantitative and qualitative evidence.

**Weaknesses:**

The paper’s presentation could be improved — some sections are densely formatted and may benefit from clearer visual structure (e.g., spacing, figure placement, and paragraph organization) to enhance overall readability.

**Questions:**

Can the proposed method be applied to other domains such as image generation?

---

> ### Author Response · Authors · 2025-11-25
> **Response to Q1**
>
> > **[Q1] Can the proposed method be applied to other domains such as image generation?**
>
> We thank the reviewer for this insightful question regarding the generalizability of our method.
>
> To assess the generalizability of EasyTune, we analyze a key determinant: **the perceptual fidelity of intermediate noisy states across different data domains.** Our investigation into the similarity between these intermediate states and their final, clean counterparts reveals that **noisy motion states are significantly more perceptible to discriminative models than noisy image states.** For instance, during a 16-step denoising process, the cosine similarity $\cos(x_0^m, x_8^m)$ between an intermediate motion $x_8^m$ and the clean motion $x_0^m$ can surpass 0.85, while the equivalent similarity for images, $\cos(x_0^i, x_8^i)$, remains below 0.7. We posit that this perceptual disparity may account for the prevalent reliance on outcome-level rewards in the image domain, as opposed to the step-aware signals utilized by our approach. Consequently, this makes the motion domain a particularly amenable setting for the step-aware optimization of EasyTune.
>
> To substantiate this analysis, we conducted a unified noise-perception study across both image and motion domains. For the image domain, we employed FLUX.1-dev as a representative model, and for motion, we utilized MLD with a TMR feature extractor. In each domain, we generated full denoising trajectories and computed cosine similarities at each step between the noisy states and the final clean samples, along with those of their ODE-based predictions and the final clean samples. Our findings, detailed in **Appendix A.6**, indicate that **motion models exhibit a markedly stronger alignment between intermediate noisy states and the final clean motions compared to image models. Notably, however, both domains show comparable performance when evaluated on their ODE-based predictions.** We refer the reviewer to **Appendix A.6** for the complete experimental settings, quantitative plots (**Fig. S4**), and qualitative visualizations (**Fig. S5 & S6**), as space constraints preclude their inclusion in the main text.
>
> ---
>
> We sincerely thank the reviewer for this suggestion, which has strengthened the motivation for our work. We have incorporated this discussion into the Motivation section (**Lines 231–236**) and provided the detailed qualitative and quantitative results in **Appendix A.6**.

---

> ### Author Response · Authors · 2025-11-25
> **Response to W1**
>
> > **[w1] The paper’s presentation could be improved — some sections are densely formatted and may benefit from clearer visual structure (e.g., spacing, figure placement, and paragraph organization) to enhance overall readability.**
>
> **Response**: We sincerely thank the reviewer for this helpful suggestion. We have revised the manuscript by adding more **space** and have **reorganized Figure 2** to improve readability. **We will ensure the final layout is further polished for the next version.**

---

### Official Review · Reviewer_yMBz · 2025-10-31

**Soundness:** 3
**Presentation:** 4
**Contribution:** 3
**Rating:** 6
**Confidence:** 4

**Summary:**

This paper presents EasyTune, a step-aware fine-tuning method for diffusion-based motion generation. The approach aims to improve the efficiency of differentiable reward optimization by decoupling recursive dependencies across denoising steps. A Self-refinement Preference Learning (SPL) mechanism is also introduced to train reward models without human annotations. Experimental results on standard benchmarks indicate performance and memory advantages over existing methods, suggesting that the proposed framework is effective and practically relevant.

**Strengths:**

1. The paper clearly explains the main problem in existing diffusion fine-tuning methods and gives a straightforward way to make the optimization more efficient.

2. The experiments are thorough and show steady improvements on several benchmarks and models.

3. The method can be applied in practice since it does not rely on human-labeled data for reward learning.

4. The writing and presentation are clear and easy to follow.

5. The video results presented in the supplementary materials look decent.

**Weaknesses:**

### Major Concerns
1. In the proposed Self-refinement Preference Learning (SPL), a pre-trained text-to-motion retrieval model is used for preference evaluation. It would be important to clarify how critical the choice of this model is. For example, if a weaker retrieval model were used, would the overall performance of EasyTune be significantly affected? Some discussion or analysis on this point would strengthen the argument, especially since the retrieval model is further fine-tuned.

2. While the paper emphasizes the large reduction in GPU memory, the efficiency analysis could be expanded. It would be helpful to report additional metrics such as GFLOPs, inference/training time, and the GPU type used, to give a more complete picture of the claimed efficiency gains.

3. The theoretical analysis, although helpful for understanding the recursive dependency issue, remains somewhat limited. A more detailed examination of convergence behavior or optimization stability under step-wise updates would improve the technical depth.

4. The SPL module depends on automatically mined preference pairs, which could introduce noise or bias. Some quantitative analysis on the reliability or noise level of these pseudo pairs would make the approach more convincing.

### Minor Concerns
- In Line 78, the symbol “θ” appears too close to the “p” in “depended”; it would be better to slightly adjust the spacing for readability.

**Questions:**

1. Could the authors provide more details on how the step-wise optimization interacts with the diffusion time schedule? For instance, does the reward weighting or learning rate vary across steps, and how sensitive is the method to this design?

2. In Eq. (7), the stop-gradient operation ensures local optimization per step. Does this affect the global consistency of the final denoised result? Have the authors observed any degradation in smoothness or temporal coherence of the generated motions?

3. Regarding the SPL mechanism, how often are new preference pairs mined during training? Is the mining process iterative (i.e., reward model updated → new pairs mined), or is it performed only once before fine-tuning?

4. The proposed method is demonstrated on motion generation tasks. Do the authors believe EasyTune can generalize to other diffusion domains (e.g., image or video generation), or are there task-specific assumptions that limit its applicability?

Looking forward to hearing from the authors for more explanation and discussion.

---

> ### Author Response · Authors · 2025-11-25
> **Response to Q1(1/2)**
>
> > **[Q1] Could the authors provide more details on how the step-wise optimization interacts with the diffusion time schedule? For instance, does the reward weighting or learning rate vary across steps, and how sensitive is the method to this design?**
>
> **Response**: We thank the reviewer for this helpful suggestion and agree that more details are needed on how step-wise optimization interacts with the diffusion time schedule.
>
> To address this, we provide additional experiments on step-level reward reweighting in **Appendix A.8**, and we further analyze how the reward model responds to different diffusion steps in Appendix **A.6 and A.10.3.**
>
> ---
>
> > **[A-1] Results on Reward Weighting Experiment**
>
> To clarify how step-wise optimization interacts with the diffusion time schedule, we conduct an ablation on step-level reward reweighting and evaluate four alternative strategies, summarized in **Appendix A.8**. Specifically, we compare: (1) optimizing only the **final 20 steps**; (2) optimizing only the **initial 20 steps**; (3) linearly increasing weights that **emphasize later steps**; and (4) linearly decreasing weights that **emphasize earlier steps**.
>
> | Method | R@1 | R@2 | R@3 | FID ↓ | MM Dist ↓ | Diversity → | Memory (GB) ↓ |
> |-|-|-|-|-|-|-|-|
> | Real | 0.511 | 0.703 | 0.797 | 0.002 | 2.974 | 9.503 | - |
> | MLD (Baseline) | 0.481 | 0.673 | 0.772 | 0.473 | 3.196 | 9.724 | 15.21 |
> | w/ DRaFT-50 | 0.528 | 0.724 | 0.819 | 0.197 | 2.872 | 9.641 | 37.32 |
> | w/ EasyTune (Full) | 0.581 | 0.769 | 0.855 | 0.132 | 2.637 | 9.465 | 22.10 |
> | EasyTune + (1) | 0.546 | 0.735 | 0.804 | 0.184 | 2.815 | 9.682 | 22.10 |
> | EasyTune + (2) | 0.567 | 0.759 | 0.842 | 0.158 | 2.673 | 9.430 | 22.10 |
> | EasyTune + (3) | 0.556 | 0.748 | 0.838 | 0.147 | 2.652 | 9.421 | 22.10 |
> | EasyTune + (4) | 0.584 | 0.773 | 0.859 | 0.136 | 2.631 | 9.521 | 22.10 |
>
> Overall, strategies (1) and (2) are clearly suboptimal, whereas both reweighted variants are beneficial, with the linearly decreasing scheme (strategy (4)) performing best and even slightly outperforming EasyTune (Full).

---

> ### Author Response · Authors · 2025-11-25
> **Response to Q1(2/2)**
>
> > **[A-2] Discussion on Dynamic Learning Rate**
>
> A simple way to emphasize earlier denoising steps is to use a time-dependent learning rate (larger LR for smaller t). In our setting, this is mathematically equivalent to step-wise reward reweighting, since only the reward term depends on the trajectory. Our objective is:
> $$\mathcal{L}\_{\mathrm{EasyTune}}(\theta) = -\mathbb{E}\_{c\sim \mathcal{D}\_\mathrm{T}, \mathbf{x}^\theta_t \sim \pi_\theta(\cdot|c), t\sim \mathcal{U}(0, T)} \left[ \mathcal{R}_{\phi} (\mathbf{x}^\theta_t, t, c) \right].$$
>
> When we use a time-dependent learning rate, the parameter update becomes:
> $$\theta = \theta - l_r(t)\cdot \nabla\_\theta \mathcal{L}\_{\mathrm{EasyTune}}(\theta), $$
> where $l_r(t)$ denotes the learning rate at time step $t$.
>
> This update can be rewritten as
> $$\begin{cases}
> \theta = \theta - l\_r(t) \cdot \nabla_\theta \mathcal{L}\_{\mathrm{EasyTune}}(\theta) \\\
> \mathcal{L}\_{\mathrm{EasyTune}}(\theta) = -\mathbb{E} \left[ \mathcal{R}\_{\phi} (\mathbf{x}^\theta_t, t, c) \right]
> \end{cases} \iff \begin{cases}
> \theta = \theta -  \nabla_\theta \mathcal{L}\_{\mathrm{EasyTune}}(\theta) \\\\
> \mathcal{L}\_{\mathrm{EasyTune}}(\theta) = -\mathbb{E}   \left[ l_r(t) \cdot \mathcal{R}_{\phi} (\mathbf{x}^\theta_t, t, c) \right]
> \end{cases}$$
> which shows that using a dynamic learning rate is effectively equivalent to applying step-dependent weights $l_r(t)$ to the reward signal.
>
> Therefore, from a theoretical perspective, the effects of the dynamic learning rate should be consistent with those of step-wise reward reweighting.
>
> > **[A-3] Analysis of reward perception across diffusion steps**
>
> To further analyse the reliability of our method in noisy states, we also analyze the reward perception across different diffusion steps (see **Appendix A.6** and **A.10.3**). These results show that the reward model reliably perceives per-step noise levels and provides effective supervision during optimization.
>
> | Methods | Input Data | R@1 (T→M) | R@2 (T→M) | R@3 (T→M) | R@5 (T→M) | R@10 (T→M) | R@1 (M→T) | R@2 (M→T) | R@3 (M→T) | R@5 (M→T) | R@10 (M→T) |
> |-|-|-|-|-|-|-|-|-|-|-|-|
> | ReAlign | Clean Data | 67.59 | 82.24 | 87.44 | 91.97 | 96.28 | 68.94 | 82.86 | 87.95 | 92.44 | 96.28 |
> | ReAlign | Noisy Data | 67.20 | 81.46 | 87.11 | 91.39 | 95.67 | 68.02 | 81.84 | 87.56 | 91.39 | 95.69 |
> | ReAlign | Predicted Clean Data | 67.80 | 82.38 | 87.86 | 92.27 | 96.61 | 68.04 | 82.47 | 87.97 | 92.10 | 96.34 |
> | SPL | Clean Data | 69.31 | 83.71 | 88.66 | 92.81 | 96.75 | 70.23 | 83.41 | 88.72 | 93.07 | 97.04 |
> | SPL | Noisy Data | 69.36 | 83.63 | 88.53 | 92.83 | 96.76 | 70.34 | 83.41 | 88.66 | 93.04 | 96.93 |
> | SPL | Predicted Clean Data | 68.39 | 83.31 | 88.59 | 93.11 | 96.73 | 68.60 | 82.35 | 88.06 | 92.79 | 96.53 |
> | TMR | Clean Data | 67.16 | 81.32 | 86.81 | 91.43 | 95.36 | 67.97 | 81.20 | 86.35 | 91.70 | 95.27 |
> | TMR | Predicted Clean Data | 66.98 | 81.04 | 87.09 | 92.11 | 95.74 | 68.32 | 80.69 | 86.43 | 92.13 | 95.84 |
>
> ---
>
> We sincerely thank the reviewer for these valuable comments, which help clarify and strengthen the effectiveness of our step-wise optimization. Following the reviewer’s suggestion, we have added the corresponding experiments and discussions in Appendix **A.6, A.8, and A.10.3**.

---

> ### Author Response · Authors · 2025-11-25
> **Response to Q2 (1/2)**
>
> > **[Q2] In Eq. (7), the stop-gradient operation ensures local optimization per step. Does this affect the global consistency of the final denoised result? Have the authors observed any degradation in smoothness or temporal coherence of the generated motions?**
>
> **Response**: We thank the reviewer for the helpful suggestion and fully understand the concern that the stop-gradient operation might undermine the global consistency of the final denoised results. This issue has been carefully considered in our design:
>   1) **Appendix B.2** provides a **theoretical analysis** demonstrating that the bias introduced by step-wise optimization is provably bounded.
>   2) **Section 4.1 (Fig. 5)** introduces **Chain Optimization**, a practical strategy that mitigates this bias while preserving the chain property.
>   3) **Appendix B.1** offers a **policy-gradient perspective** that supports the theoretical soundness of our optimization framework.
>
> ---
>
> > **[A2-1] Theoretical analysis of the bounded bias**
>
> From a theoretical perspective, **Appendix B.2** quantifies **the bias introduced by Step Optimization relative to reward-guided sampling**. In particular, the reward-guided sampling update can be written as
> $$
> \mathbf{x}\_{t-1}^g = \mathbf{x}\_{t-1} + \nabla\_\theta \mathcal{R}(\mathbf{x}\_{t-1}), \\; \text{(Reward-Guided Sampling)}
> $$
> and Corollary B.1 shows that the optimization of our method is equivalent to reward-guided sampling up to a bounded bias:
> $$
> \mathbf{x}\_{t-1} = \mathbf{x}\_{t-1}^g + \mathcal{O}(\|\nabla_\theta \mathcal{R}\|^2),
> $$
> where $\mathbf{x}\_{t-1}$ is the sample obtained with step-wise parameter updates and $\mathbf{x}_{t-1}^g$ is the corresponding reward-guided sample. Thus, the discrepancy between the two trajectories is only second-order in the gradient norm and is negligible under the small learning rates used in practice.
>
> Therefore, our method is **theoretically close to reward‑guided sampling**, which has already been **preliminarily explored in prior work** [1] and likewise has not been observed to compromise global consistency.
>
> [1] ReAlign: Text-to-Motion Generation via Step-Aware Reward-Guided Alignment, AAAI 2026

---

> ### Author Response · Authors · 2025-11-25
> **Response to Q2 (2/2)**
>
> > **[A2-2] Discussion & Results on Chain Optimization**
>
> We now briefly revisit the design of Chain Optimization in **Fig. 5**. To further mitigate this bias in practice by decoupling sampling and learning, we maintain a separate, frozen copy of the model for denoising, generate a complete denoising trajectory with this frozen model, and update the trainable parameters only once after the whole trajectory is obtained. In this way, **all samples in a trajectory are produced by a fixed model rather than a model whose parameters are being updated online**, which effectively avoids the step-optimization bias. The corresponding comparison between Step Optimization and Chain Optimization is reported in **Table 1.**
>
>
> > **[A2-3] Discussion on Step Optimization and Policy Gradient Methods**
>
> We further relate our step-wise objective to standard policy-gradient methods, another RL-based approach for diffusion optimization, in **Appendix B.1**. Compared with other differentiable-reward methods, **both our approach and these policy-gradient methods truncate the gradient along the diffusion chain, and in practice, they have not been observed to suffer from global inconsistency issues.**
>
>
> Classical methods such as GRPO and DDPO treat the diffusion model as a policy $p\_\theta(\mathbf{x}\_{0:T} \mid c) = \prod\_{t=0}^{T-1} p\_\theta(\mathbf{x}\_t \mid \mathbf{x}\_{t+1}, c, t)$ and optimize a trajectory-level reward. For example, DDPO maximizes
> $$
> \nabla\_\theta \mathcal{J}\_{\mathrm{DDPO}}(\theta) = \mathbb{E} \Big[ r(\mathbf{x}\_0,c) \sum\_{t=0}^{T-1} \nabla_\theta \log p\_\theta(\mathbf{x}\_t \mid \mathbf{x}\_{t+1}, c, t) \Big].
> $$
> GRPO adopts a similar structure but replaces the scalar reward $r(\mathbf{x}\_0,c)$ with a group-wise normalized advantage $\mathcal{A}^k$ and applies PPO-style clipping. Concretely, its objective can be written as
> $$
> \mathcal{J}\_{\mathrm{GRPO}}(\theta, \mathcal{G})
> = \mathbb{E}\_{\substack{c\sim\mathcal{D},\, \mathbf{x}^{1:K}, t\sim\mathcal{U}(0,T),\, k\sim\mathcal{U}(1,K)}} \Big[
> \min\big( \rho\_{\theta}^{k,t}\,\mathcal{A}^{k},\;
> \mathrm{clip}(\rho\_{\theta}^{k,t},\,1-\varepsilon,\,1+\varepsilon)\,\mathcal{A}^{k} \big)
> \Big],
> $$
> where
> $$
> \mathcal{A}^{k} = \frac{r(\mathbf{x}^{k}\_0,c)-\mu(r(\mathbf{x}^{1:K},c))}{\sigma(r(\mathbf{x}^{1:K},c))},\quad
> \rho\_{\theta}^{k,t} =
> \frac{p\_{\theta}(\mathbf{x}^{k}\_t \mid \mathbf{x}^{k}\_{t-1}, c)}
>      {p\_{\theta\_{\mathrm{old}}}(\mathbf{x}^{k}\_t \mid \mathbf{x}^{k}\_{t-1}, c)}.
> $$
>
> Intuitively, these policy-gradient methods also avoid backpropagating rewards through the entire diffusion trajectory; instead, they update the model only via local, step-wise log-probability terms (see $\rho_{\theta}^{k,t}$), which effectively truncates gradient propagation along the chain.
>
> Similarly, EasyTune directly optimizes a step-wise objective
> $$
> \mathcal{L}\_{\mathrm{EasyTune}}(\theta) = -\mathbb{E}\_{c \sim \mathcal{D}\_\mathrm{T},\; t \sim \mathcal{U}(0,T),\; \mathbf{x}\_t^\theta \sim \pi_\theta(\cdot \mid c)} \big[
> \mathcal{R}\_\phi(\mathbf{x}\_t^\theta, t, c)
> \big],
> $$
> where $\mathcal{R}_\phi(\mathbf{x}_t^\theta, t, c)$ is a reward assigned directly to the intermediate sample at step $t$.
>
> In summary, EasyTune, like GRPO and DDPO, optimizes the diffusion model via **step-wise**, reward-weighted updates rather than a single end-to-end backpropagation. Consequently, in practice, both our approach and these policy-gradient methods have not been observed to suffer from **global inconsistency issues**.
>
> ---
>
> We sincerely thank the reviewer for this question, which has helped us deepen the discussion in our paper. We hope that our response has adequately addressed your concerns.

---

> ### Author Response · Authors · 2025-11-25
> **Response to Q3&W1&W4 (1/2)**
>
> > **[Q3] Regarding the SPL mechanism, how often are new preference pairs mined during training? Is the mining process iterative (i.e., reward model updated → new pairs mined), or is it performed only once before fine-tuning?**
>
> > **[W1] In the proposed Self-refinement Preference Learning (SPL), a pre-trained text-to-motion retrieval model is used for preference evaluation. It would be important to clarify how critical the choice of this model is. For example, if a weaker retrieval model were used, would the overall performance of EasyTune be significantly affected? Some discussion or analysis on this point would strengthen the argument, especially since the retrieval model is further fine-tuned.**
>
> > **[W4] The SPL module depends on automatically mined preference pairs, which could introduce noise or bias. Some quantitative analysis on the reliability or noise level of these pseudo pairs would make the approach more convincing.**
>
>
> **Response:** We thank the reviewer for the helpful suggestions and agree that clarifying the SPL configuration and providing more detailed analysis will strengthen the paper.
>
> In response, we first **clarify the setup of SPL**. We then study its performance under **a weaker retrieval model**, under **different choices of the candidate number $K$ and retrieval pool**, and finally examine its **robustness** to noise. The corresponding discussions and results are provided in **Appendix A.10.1, A.10.2, and A.10.3** of the revised manuscript.
>
>
> ---
>
> > **[A-Q1] SPL Configuration**
>
> Our method **dynamically mines preference pairs online** during each optimization iteration (i.e., **reward model updated → new pairs mined**). The rationale behind this iterative approach is that we aim to mine preference pairs that target the mistake of the current model state, rather than the pretrained one. This online setting enables the model to perform targeted self-correction, specifically for its own retrieval errors, leading to more effective self-improvement. We also clarify this issue in **Appendix B.4** to avoid potential misunderstandings for readers.
>
>
> > **[A-W1-1] Experimental results on the weaker retrieval model**
>
> To directly address the reviewer’s concern about the dependence on the retrieval model, we conduct a sensitivity analysis using TMR, **a weaker retrieval model** than the one used in our main experiments. In text-motion retrieval experiments, SPL still consistently improves TMR, as summarized below:
>
> | Method | T2M R@1 | T2M R@2 | T2M R@3 | T2M R@5 | T2M R@10 | M2T R@1 | M2T R@2 | M2T R@3 | M2T R@5 | M2T R@10 |
> | --- | --- | --- | --- | --- | --- | --- | --- | --- | --- | --- |
> | TMR | 67.16 | 81.32 | 86.81 | 91.43 | 95.36 | 67.97 | 81.20 | 86.35 | 91.70 | 95.27 |
> | TMR + SPL | 68.76 | 82.36 | 87.99 | 92.06 | 96.47 | 69.03 | 82.87 | 87.84 | 92.56 | 96.45 |
>
> These results show that SPL is not restricted to strong retrieval models and can also benefit weaker ones, detailed in **Appendix A.10.1**.
>
>
>
> > **[A-W1-2] Experimental results using a weaker reward model for generation**
>
> In **Appendix A.10.1**, we further evaluate whether **generative models** with step-aware optimization can still benefit from **a weaker reward model**. Using TMR as the reward model for MLD, both the step and chain variants consistently improve over the baseline across retrieval and distributional metrics, as summarized below:
>
> | Method | R-Precision Top 1 | Top 2 | Top 3 | FID ↓ | MM Dist ↓ | Diversity → |
> | --- | --- | --- | --- | --- | --- | --- |
> | Real | 0.511 | 0.703 | 0.797 | 0.002 | 2.974 | 9.503 |
> | MLD (Baseline) | 0.481 | 0.673 | 0.772 | 0.473 | 3.196 | 9.724 |
> | +TMR (w/ SPL, Step) | 0.573 | 0.760 | 0.843 | 0.173 | 2.682 | 9.942 |
> | +TMR (w/ SPL, Chain) | 0.567 | 0.753 | 0.836 | 0.158 | 2.698 | 9.874 |
>
> These results confirm that even a weaker but reasonably discriminative retrieval model can serve as an effective reward model for guiding the generative model.

---

> ### Author Response · Authors · 2025-11-25
> **Response to Q3&W1&W4 (2/2)**
>
> > **[A-W1-3] Additional analysis on candidate size $K$ and retrieval pools**
>
> To further address the reviewer’s concerns, we study how the **candidate number (K)** and the **retrieval pool** influence the behavior and stability of SPL. As detailed in **Appendix A.10.2**, SPL mines preference pairs online, so it is important that successful retrievals substantially outnumber failures; this is mainly controlled by (K) and the pool design. Concretely, following TMR we consider four pools: (a) All, using the full test set; (b) All with a high similarity threshold (>0.95); (c) a dissimilar subset of 100 motion–text pairs; and (d) randomly sampled mini-batches of 32 pairs. Because (a) and (b) have low success rates and are computationally expensive, we focus on (c) and (d) and adopt (d) in our main experiments: with (K=10), (d) yields a failure ratio of about 1:20 and stable training, whereas (c) gives roughly 1:6 and is less stable. We further perform a sensitivity study over $(K \in \\{10,15,20\\})$ under (c) and (d), with the quantitative results and discussion provided in **Appendix A.10.2**.
>
>
> | Method | R-Prec@1 | R-Prec@2 | R-Prec@3 | FID ↓ | MM Dist ↓ | Diversity → |
> | --- | --- | --- | --- | --- | --- | --- |
> | Real | 0.511 | 0.703 | 0.797 | 0.002 | 2.974 | 9.503 |
> | MLD (Baseline) | 0.481 | 0.673 | 0.772 | 0.473 | 3.196 | 9.724 |
> | $K=10$ + (d) | 0.581 | 0.769 | 0.855 | 0.132 | 2.637 | 9.465 |
> | $K=15$ + (d) | 0.571 | 0.758 | 0.843 | 0.142 | 2.668 | 9.486 |
> | $K=20$ + (d) | 0.564 | 0.747 | 0.830 | 0.184 | 2.704 | 9.629 |
> | $K=10$ + (c) | - | - | - | - | - | - |
> | $K=15$ + (c) | 0.585 | 0.773 | 0.859 | 0.155 | 2.626 | 9.428 |
> | $K=20$ + (c) | 0.574 | 0.759 | 0.844 | 0.149 | 2.653 | 9.495 |
>
> Overall, SPL consistently improves over the MLD baseline across all valid settings. Further details are provided in **Appendix A.10.2**.
>
>
> > **[A-W4] Robustness of SPL to noisy pseudo preference pairs**
>
> **Noisy data** can affect the retrieval model and thereby impact the performance of the generative model. To assess the reliability of automatically mined preference pairs under such conditions, we evaluate ReAlign, SPL, and TMR on HumanML3D with clean, noisy, and ODE-denoised motion inputs, in **Appendix A.10.3**.
>
>
> | Method | Input | T2M R@1 | T2M R@2 | T2M R@3 | T2M R@5 | T2M R@10 | M2T R@1 | M2T R@2 | M2T R@3 | M2T R@5 | M2T R@10 |
> | --- | --- | --- | --- | --- | --- | --- | --- | --- | --- | --- | --- |
> | ReAlign | Clean | 67.59 | 82.24 | 87.44 | 91.97 | 96.28 | 68.94 | 82.86 | 87.95 | 92.44 | 96.28 |
> | ReAlign | Noisy | 67.20 | 81.46 | 87.11 | 91.39 | 95.67 | 68.02 | 81.84 | 87.56 | 91.39 | 95.69 |
> | ReAlign | Pred. Clean | 67.80 | 82.38 | 87.86 | 92.27 | 96.61 | 68.04 | 82.47 | 87.97 | 92.10 | 96.34 |
> | SPL | Clean | 69.31 | 83.71 | 88.66 | 92.81 | 96.75 | 70.23 | 83.41 | 88.72 | 93.07 | 97.04 |
> | SPL | Noisy | 69.36 | 83.63 | 88.53 | 92.83 | 96.76 | 70.34 | 83.41 | 88.66 | 93.04 | 96.93 |
> | SPL | Pred. Clean | 68.39 | 83.31 | 88.59 | 93.11 | 96.73 | 68.60 | 82.35 | 88.06 | 92.79 | 96.53 |
> | TMR | Clean | 67.16 | 81.32 | 86.81 | 91.43 | 95.36 | 67.97 | 81.20 | 86.35 | 91.70 | 95.27 |
> | TMR | Pred. Clean | 66.98 | 81.04 | 87.09 | 92.11 | 95.74 | 68.32 | 80.69 | 86.43 | 92.13 | 95.84 |
>
> These results indicate that our reward models are **robust to noise in intermediate diffusion states** and that the automatically mined preference pairs are reliable in practice.
>
> ---
>
> We thank the reviewer for these insightful comments, which have led us to clarify the SPL configuration and add the above analyses and results to the revised manuscript.

---

> ### Author Response · Authors · 2025-11-25
> **Response to Q4**
>
> > **[Q4] The proposed method is demonstrated on motion generation tasks. Do the authors believe EasyTune can generalize to other diffusion domains (e.g., image or video generation), or are there task-specific assumptions that limit its applicability?**
>
>
> We thank the reviewer for this insightful question. This is indeed a valuable and worthwhile question to explore, detailed in **Appendix A.6**.
>
> To discuss the generalization of EasyTune, we first analyze a key factor: the difference in **perception of intermediate** noisy states between **motion** diffusion models versus **other domains**. Building upon this insight, we study the similarity between intermediate states and the final clean samples, as well as their correlation with reward signals. Our analysis shows that, compared with the **image domain**, **noisy motion states** are **much more easily perceived** by motion discriminators. For example, in a 16-step denoising process, the cosine similarity $\cos(x_0^m, x_8^m)$ between an intermediate motion $x_8^m$ and its clean counterpart ($x_0^m$), can exceed **0.85**, whereas the corresponding image similarity $\cos(x_0^i, x_8^i)$ is below **0.7**. This gap in the perceptibility of noisy states partly explains why image-domain methods predominantly rely on outcome-level rewards rather than step-aware rewards. Consequently, the **motion domain** is particularly **well-suited for step-aware optimization** as adopted in EasyTune.
>
> Specifically, we conduct a unified noise-perception study on both image and motion diffusion models. On the image setting, we use **FLUX.1-dev** as a representative text-to-image model; on the motion setting, we adopt **MLD** as the base generator and TMR as the feature extractor. For each domain, we generate full denoising trajectories and, at every step, compute cosine similarities between noisy states and the clean samples, along with those between their ODE-based predictions and the clean samples. As detailed in **Appendix A.6**, motion models maintain much **stronger alignment** between **intermediate noisy states and clean motions** than image models, whereas both domains exhibit similar behavior when evaluated on ODE-based predictions. Due to space constraints, we cannot include the full quantitative curves and visual examples here, and we kindly refer the reviewer to **Appendix A.6** for complete experimental settings, plots (**Fig. S4**), and qualitative visualizations (**Fig. S5 & S6**).
>
> ---
>
> We sincerely thank the reviewer for this suggestion, which has further strengthened our motivation. We have added the corresponding discussion to the Motivation section (**Lines 231–236**) and provided detailed qualitative and quantitative results in **Appendix A.6**.

---

> ### Author Response · Authors · 2025-11-25
> **Response to W2**
>
> > **[w2] While the paper emphasizes the large reduction in GPU memory, the efficiency analysis could be expanded. It would be helpful to report additional metrics such as TFLOPs, inference/training time, and the GPU type used, to give a more complete picture of the claimed efficiency gains.**
>
> **Response**: We thank the reviewer for this helpful suggestion. A thorough analysis is indeed crucial for highlighting and strengthening the contribution of our method.
>
> We provide a **detailed overhead analysis** in **Appendix A.9.2**. Beyond computational overhead, we also report more **fine-grained memory statistics**. Here, we summarize the computational costs in tabular form, while the detailed memory usage and trajectories are visualized in **Appendix A.9.1**.
>
>
> ---
>
> > **[A-1] Computational overhead (training time and TFLOPs)**
>
> **Appendix A.9.2** reports both the per-step optimization time and the **total training time and TFLOPs** needed to reach different reward scores. All methods share the same diffusion sampling procedure and are run on *a single NVIDIA RTX **A6000 GPU** with an Intel(R) Xeon(R) Silver 4316 CPU @ 2.30GHz*. There is no additional inference-time overhead. The key results are summarized in the table below (where “–” indicates that the corresponding method **fails** to reach the **given reward score** within the predefined training budget).
>
> |  | DRaFT | AlignProp | DRTune | ReFL | EasyTune (Ours) |
> | --- | --- | --- | --- | --- | --- |
> | Time per Opt. (s) | 5.61 | 5.17 | 4.90 | 4.72 | **1.47** |
> | **Reward = 0.70** Time (s) | 466.27 | 271.99 | 554.77 | 820.29 | **263.36** |
> | **Reward = 0.70** TFLOPs | 18044 | 10526 | 21469 | 31745 | **10191** |
> | **Reward = 0.75** Time (s) | 2616.54 | 971.55 | 2009.59 | - | **358.17** |
> | **Reward = 0.75** TFLOPs | 101260 | 37599 | 77771 | - | **13861** |
> | **Reward = 0.80** Time (s) | - | - | - | - | **452.53** |
> | **Reward = 0.80** TFLOPs | - | - | - | - | **17513** |
> | **Reward = 0.85** Time (s) | - | - | - | - | **1025.17** |
> | **Reward = 0.85** TFLOPs | - | - | - | - | **39674** |
>
> Overall, EasyTune requires far **less time and TFLOPs** than baselines (e.g., ~**7.3×** faster than DRaFT at reward **0.75**) and is the only method that reaches **0.80** and **0.85** within the given budget.
>
> > **[A-2] Comprehensive Memory overhead**
>
> In Appendix **A.9.1**, we compare the **GPU memory usage** of EasyTune and existing fine-tuning methods, and visualize both **per-stage usage** and the **full memory trajectory.**
>
> ---
>
> We sincerely thank the reviewer for this valuable suggestion. Following your advice, we have updated the corresponding results in **Appendix A.9** of the revised manuscript.

---

> ### Author Response · Authors · 2025-11-25
> **Response to W3**
>
> > **[W3] The theoretical analysis, although helpful for understanding the recursive dependency issue, remains somewhat limited. A more detailed examination of convergence behavior or optimization stability under step-wise updates would improve the technical depth.**
>
> **Response**: We thank the reviewer for this insightful comment and agree that explicitly characterizing **convergence** and optimization stability under step-wise updates improves the technical depth of our method. In the revised manuscript, **Appendix C.2** provides a **convergence analysis**. we briefly summarize it below.
>
> Under standard assumptions for non-convex stochastic gradient methods (i) the expected objective $\mathcal{L}(\theta)$ is lower bounded and $L$-smooth, (ii) the stochastic gradient has bounded second moment, and (iii) the bias introduced by the stop-gradient operation is uniformly bounded and proportional to the step size **Appendix C.2** proves that
> $$
> \liminf_{k\to\infty} \mathbb{E}\big[\|\nabla\mathcal{L}(\theta_k)\|^2\big] = 0.
> $$
> This result ensures stable step-wise optimization. Detailed proof is in **Appendix C.2**.
>
> ---
>
> We thank the reviewer for this suggestion, which has helped improve the theoretical depth and completeness of our method.

---

> ### Author Response · Authors · 2025-11-25
> **Response to Minor Concern**
>
> > **[W5] In Line 78, the symbol “θ” appears too close to the “p” in “depended”; it would be better to slightly adjust the spacing for readability.**
>
> **Response**: We thank the reviewer for carefully pointing out this typographical issue. In the revised manuscript, we have adjusted the spacing between the letter “p” and the symbol θ in “depended” to improve clarity and readability.
>
> ---
>
> We thank the reviewer for their careful reading, which helps improve the overall presentation quality of the paper.

---

### Author Response · Authors · 2025-12-01
**Global Response**

We thank all reviewers for their constructive and insightful feedback and for the overall positive assessment of our work. Overall, reviewers **pyGX** and **UAQn** recommend **borderline acceptance**, reviewer **v4NY** recommends **strong acceptance**, and reviewer **yMBz**, despite an initial borderline‑reject rating, finds that our revisions **address the main concerns** and is willing to **raise the score toward acceptance**. We sincerely appreciate the reviewers' positive feedback.

To address the main issues raised by the reviewers, we made the following key changes:

- We strengthened the evidence for the robustness and transferability of SPL and our reward models by adding **a unified noise‑perception study** (**Appendix A.6**), **robustness experiments on noisy and ODE‑denoised inputs** (**Appendix A.10.3**), **experiments with a weaker retrieval/reward model TMR** (**Appendix A.10.1**), **a sensitivity analysis of candidate number and retrieval pools** with practical guidelines (**Appendix A.10.2, Appendix B.4**), and **a physical‑perception analysis of the reward model** (**Appendix A.11**).

- We expanded the efficiency, overhead, and reproducibility analysis by reporting **per‑step optimization time, total training time, and TFLOPs** on a single RTX A6000 (**Appendix A.9.2**) and **detailed GPU memory trajectories** (**Appendix A.9.1**), showing that EasyTune achieves significantly lower training cost and memory usage while keeping inference‑time overhead unchanged. We also added **a failure‑case analysis of reward hacking** with mitigation strategies such as KL regularization and early stopping (**Appendix A.7, Appendix A.3**), and improved reproducibility by **correcting the learning‑rate configuration and adding a learning‑rate ablation** (**Sec. 5.1, Appendix A.10.4**).

- We clarified the EasyTune framework and its theoretical foundations by providing **a bounded‑bias analysis that connects our step‑wise updates to reward‑guided sampling** (**Appendix B.2**), introducing **Chain Optimization** and its relation to **policy‑gradient‑based methods** (**Section 4.1, Appendix B.1**), adding **step‑level reward‑reweighting ablations over the diffusion time schedule** (**Appendix A.8**), and giving **explicit guidance on noise‑aware vs. one‑step (ODE) rewards** (**Appendix A.6, Appendix A.10.3, Appendix B.5**), together with **a convergence analysis of step‑wise optimization** (**Appendix C.2**). We further expanded the discussion of **differentiable vs. non‑differentiable rewards** and the applicability of EasyTune in text‑to‑motion and physical‑aware settings (**Related Work, Limitation**), and **improved the manuscript layout and figure organization** to enhance readability.

We are very grateful to reviewers yMBz, v4NY, pyGX, and UAQn for their thoughtful comments, which helped us substantially improve the paper's clarity, rigor, and significance. We sincerely thank all reviewers and area chairs for their time and careful consideration.

---

### Meta-Review · Area_Chair_tLcn · 2025-12-24

**Summary:**

This paper proposes a EasyTune framework for diffusion-based motion generation to enables dense and effective optimization using differentiable rewards.
The strength of the paper is the clear motivation and conceptual formulation of the proposed EasyTune framework.

The reviewers raised the following concerns:
1. limited theoretical analysis on convergence;
2. the robustness of the retrieval model;
3. detailed computational overhead metrics
4. the applicability to physical/non-differentiable rewards.
After rebuttal, most concerns are well solved, and the consensus is positive.

In summary, this paper was reviewed by four experts in the field. The recommendations are 6, 10, 6, 4. The reviewers like the clear motivation and effectiveness of the proposed step-aware fine-tuning framework (EasyTune) in resolving memory bottlenecks (achieving O(1) memory cost), the novel Self-refinement Preference Learning (SPL) mechanism for label-free optimization, and the strong empirical improvements in both efficiency and generation quality. And concerns on theoretical convergence analysis, detailed computational overhead metrics, robustness to noise or weaker retrieval models, and applicability to non-differentiable rewards are well solved.

**Reviewer Concerns:**

**Well addressed:**

- Computational Efficiency Analysis (yMBz): The reviewer requested more detailed metrics. The authors provided a comprehensive breakdown of TFLOPs, total training time, and detailed GPU memory trajectories, showing that EasyTune is significantly faster and more memory-efficient.
- Theoretical Convergence (yMBz, pyGX): The authors added a convergence analysis for the step-wise optimization and clarified the bias bounds (Corollaries 1 & 2), satisfying the reviewers' request for theoretical depth.
- Robustness of SPL (yMBz, UAQn): Concerns about reliance on specific retrieval models were addressed via sensitivity analyses using weaker models (TMR) and noisy inputs, demonstrating the method is robust.
- Physical/Non-Differentiable Rewards & Metric Chasing (UAQn): The authors clarified that their method aligns with physical constraints implicitly and explained the trade-offs, which the reviewer found convincing.
- Generalization to other domains (pyGX): The authors explained the noise perception gap between image and motion domains.

**Partly addressed:**

None.

**Unsolved:**

None.

**Reviewer Scores:**

**yMBz (6):**

yMBz explicitly stated in the discussion that they are ``willing to raise the score toward acceptance`` after the revisions.
yMBz would keep or raise the score.

**v4NY (10):**

v4NY recommends strong acceptance and had no major remaining issues, and would likely keep the scores.

**pyGX (6):**

pyGX stated that ``I am in favor of this paper and will maintain my current score``, and would likely keep the scores.


**UAQn (4):**

UAQn was initial negative but explicitly commented after the rebuttal ``Well rebuttal! I will raise my score ASAP!``
UAQn would likely raise the score.

---

### Decision · Program_Chairs · 2026-01-26

Accept (Poster)